# Variance Reduced ProxSkip: Algorithm, Theory and Application to Federated Learning

**Grigory Malinovsky**
KAUST
grigory.malinovsky@kaust.edu.sa

**Kai Yi**
KAUST
kai.yi@kaust.edu.sa

**Peter Richtárik**
KAUST
peter.richtarik@kaust.edu.sa

## Abstract

We study distributed optimization methods based on the *local training (LT)* paradigm: achieving communication efficiency by performing richer local gradient-based training on the clients before parameter averaging. Looking back at the progress of the field, we *identify 5 generations of LT methods*: 1) heuristic, 2) homogeneous, 3) sublinear, 4) linear, and 5) accelerated. The 5[th] generation, initiated by the ProxSkip method of Mishchenko et al. [2022] and its analysis, is characterized by the first theoretical confirmation that LT is a communication acceleration mechanism. Inspired by this recent progress, we contribute to the 5[th] generation of LT methods by showing that it is possible to enhance them further using *variance reduction*. While all previous theoretical results for LT methods ignore the cost of local work altogether, and are framed purely in terms of the number of communication rounds, we show that our methods can be substantially faster in terms of the *total training cost* than the state-of-the-art method ProxSkip in theory and practice in the regime when local computation is sufficiently expensive. We characterize this threshold theoretically, and confirm our theoretical predictions with empirical results.

## 1 Introduction

Announced in April 2017 in a Google AI blog [McMahan and Ramage, 2017], and citing four foundational papers [Konečný et al., 2016b,a, McMahan et al., 2017, Bonawitz et al., 2017] of what was to become a new and rapidly growing interdisciplinary field, *federated learning* (FL) constitutes a novel paradigm for training supervised machine learning models. The key idea is the acknowledgement that increasing amounts of data are being captured and stored on edge devices, such as mobile phones, sensors and hospital workstations, and that moving the data to a datacenter for centralized processing may be infeasible or undesirable for various reasons, including high energy costs and data privacy concerns [Kairouz et al., 2019, Li et al., 2020a]. FL faces a multitude of challenges which are being actively addressed by the research community.

### 1.1 Formalism

We study the standard optimization formulation of federated learning [Konečný et al., 2016a, McMahan et al., 2017, Kairouz et al., 2019, Wang et al., 2021] given by

$$\min_{x \in \mathbb{R}^{d'}} \phi(x), \qquad \phi(x) := \sum_{i=1}^{M} \frac{n_i}{n} \phi_i(x), \qquad \phi_i(x) := \frac{1}{n_i} \sum_{j=1}^{n_i} \phi_{ij}(x), \tag{1}$$

36th Conference on Neural Information Processing Systems (NeurIPS 2022).

where $M$ is the number of clients (devices, machines, workers), $n_i$ is the number of training data points on client $i \in \{1, 2, \ldots, M\}$, and $n := \sum_{i=1}^{M} n_i$ is the total number of training data points collectively owned by this federation of $M$ clients. Note that $\phi$ is the empirical risk over the federated dataset. Perhaps conceptually the simplest method for solving (1) is *gradient descent (*GD*)*,

$$x_{t+1} = x_t - \gamma \nabla \phi(x_t) = x_t - \gamma \sum_{i=1}^{M} \frac{n_i}{n} \nabla \phi_i(x_t) = \sum_{i=1}^{M} \frac{n_i}{n} \left( x_t - \gamma \nabla \phi_i(x_t) \right), \qquad (2)$$

where $\gamma > 0$ is the stepsize. It will be useful to describe how GD would be implemented in a federated environment. First, all clients $i \in \{1, \ldots, M\}$ in parallel perform a single local gradient step starting from the current global model $x_t$, arriving at the local models $x_{it} := x_t - \gamma \nabla \phi_i(x_t)$, $i \in \{1, \ldots, M\}$. These local models are then communicated to the *orchestrating server*, which aggregates them via weighted averaging, arriving at the new global model $x_{t+1} = \sum_{i=1}^{M} \frac{n_i}{n} x_{it}$. This new model is then broadcast back to all clients, and the process is repeated until a model of sufficient quality is found.

## 1.2 Federated averaging

Proposed by Povey et al. [2015], Moritz et al. [2016], McMahan et al. [2017], federated averaging (FedAvg) is arguably the most popular method for solving the standard FL formulation (1). Motivated by the specific constraints of federated environments, FedAvg can be seen as a practical enhancement of GD via the simultaneous application of three techniques: a) data sampling (DS), b) client sampling (CS), and c) local training (LT). That is,

FedAvg = GD + (DS + CS + LT).

We will now briefly describe each of these three GD-enhancing techniques separately.

(a) **GD + Data Sampling.** In situations when the local datasets are so large that the computation of the exact local gradients becomes a bottleneck, it makes sense to approximate them via data sampling. That is, instead of passing through all local data to compute the local gradient $\nabla \phi_i(x_t)$, each client $i$ computes the gradients $\nabla \phi_{ij}(x_t)$ for $j \in \mathcal{D}_{it}$ only, where $\mathcal{D}_{it}$ is a suitably chosen small-enough subset of the local dataset $\{1, \ldots, n_i\}$. These gradients are then used to form gradient estimators $g_i(x_t) \approx \nabla \phi_i(x_t)$ which are used to perform a local SGD step on all clients. The rest of the procedure is the same as in the case of GD. That is, the local models obtained in this way are sent to the orchestrating server, the server aggregates them via weighted averaging and broadcasts the resulting model back to all clients. Combination of GD and DS can be seen as a particular version of SGD, where the stochastic gradient estimator is formed from the gradients $\nabla \phi_{ij}(x_t)$ associated with the datapoints $(i, j)$ where $j \in \cup_{i=1}^{M} \mathcal{D}_{it}$. While DS is still an active area of research, it has been studied for a long time, and is in general well understood [Takáč et al., 2013, Li et al., 2014, Csiba and Richtárik, 2018, Gower et al., 2019a, Horváth and Richtárik, 2019, Khaled and Richtárik, 2020].

(b) **GD + Client Sampling.** In practical federated environments, and especially in cross-device FL [Kairouz et al., 2019], the number of clients is enormous, they are not all available at all times, and the orchestrating server has limited compute and memory capacity. For these and other reasons, practical FL methods need to work in an environment in which a small subset $\mathcal{S}_t \subseteq \{1, \ldots, M\}$ of the clients is sampled ("participates") in each communication/aggregation/training round only. Since only the participating clients $i \in \mathcal{S}_t$ perform a local GD step and communicate the resulting local model to the orchestrating server for aggregation, this induces an error compared to GD, which has an adverse effect on the convergence rate. Combination of GD and CS can be seen as a particular version of SGD, where the stochastic gradient estimator is formed from the gradients $\nabla \phi_{ij}(x_t)$ associated with the datapoints $(i, j)$ where $j \in \cup_{i=1}^{\mathcal{S}_t} \{1, \ldots, n_i\}$. While CS is still an active area of research, since CS is a special type of DS, much was known about CS long before FedAvg was proposed [Gower et al., 2019a, Horváth and Richtárik, 2019]. Still, CS poses new challenges tackled by the community Eichner et al. [2019], Chen et al. [2020], Gower et al. [2019a], Cho et al. [2020], Charles et al. [2021].

(c) **GD + Local Training.** In federated learning, the cost of communication between the clients and the orchestrating server forms the key bottleneck. Indeed, in their FedAvg paper, which introduced LT to the world of federated learning, McMahan et al. [2017] wrote:

Table 1: Five generations of local training (LT) methods summarizing the progress made by the ML/FL community over the span of 7+ years in the understanding of the *communication acceleration properties of LT*.

| Generation[a] | Theory | Assumptions | Comm. Complexity[b] | Selected Key References |
|---|---|---|---|---|
| 1. Heuristic | ✗ | — | empirical results only | LocalSGD [Povey et al., 2015] |
| | ✗ | — | empirical results only | SparkNet [Moritz et al., 2016] |
| | ✗ | — | empirical results only | FedAvg [McMahan et al., 2017] |
| 2. Homogeneous | ✓ | bounded gradients | sublinear | FedAvg [Li et al., 2020b] |
| | ✓ | bounded grad. diversity[c] | linear but worse than GD | LFGD [Haddadpour and Mahdavi, 2019] |
| 3. Sublinear | ✓ | standard[d] | sublinear | LGD [Khaled et al., 2019] |
| | ✓ | standard | sublinear | LSGD [Khaled et al., 2020] |
| 4. Linear | ✓ | standard | linear but worse than GD | Scaffold [Karimireddy et al., 2020] |
| | ✓ | standard | linear but worse than GD | S-Local-GD [Gorbunov et al., 2020a] |
| | ✓ | standard | linear but worse than GD | FedLin [Mitra et al., 2021] |
| 5. Accelerated | ✓ | standard | linear & better than GD | ProxSkip/Scaffnew [Mishchenko et al., 2022] |
| | ✓ | standard | linear & better than GD | ProxSkip-VR [**THIS WORK**] |

[a] Since client sampling (CS) and data sampling (DS) can only *worsen* theoretical communication complexity, our historical breakdown of the literature into 5 generations of LT methods focuses on the full client participation (i.e., no CS) and exact local gradient (i.e., no DS) setting. While some of the referenced methods incorporate CS and DS techniques, these are irrelevant for our purposes. Indeed, from the viewpoint of communication complexity, all these algorithms enjoy best theoretical performance in the no-CS and no-DS regime.

[b] For the purposes of this table, we consider problem (1) in the *smooth* and *strongly convex* regime only. This is because the literature on LT methods struggles to understand even in this simplest (from the point of view of optimization) regime.

[c] *Bounded gradient diversity* is a uniform bound on a specific notion of gradient variance depending on client sampling probabilities. However, this assumption (as all homogeneity assumptions) is very restrictive. For example, it is not satisfied the standard class of smooth and strongly convex functions.

[d] The notorious FL challenge of handling non-i.i.d. data by LT methods was solved by Khaled et al. [2019] (from the viewpoint of *optimization*). From generation 3 onwards, there was no need to invoke any data/gradient homogeneity assumptions. Handling non-i.i.d. data remains a challenge from the point of view of *generalization*, typically by considering *personalized* FL models.

> *"In contrast[1], in federated optimization communication costs dominate".*

LT is a conceptually simple and surprisingly powerful communication-acceleration technique. The basic idea behind LT is for the clients to perform *multiple* local GD steps instead of a single step (which is how GD operates) before communication and aggregation takes place. The intuitive reasoning used in virtually all papers on this topic is: performing multiple local GD steps results in "richer" and ultimately more useful local training in the sense that fewer communication rounds will *hopefully* suffice to finish the training. McMahan et al. [2017] supported this intuition with ample empirical evidence, and credited LT as the critical component behind the success of FedAvg:

> *"Thus, our goal is to use additional computation in order to decrease the number of rounds of communication needed to train a model. . ."* *"Communication costs are the principal constraint, and we show a reduction in required communication rounds by 10–100× as compared to synchronized stochastic gradient descent."* *". . . the speedups we achieve are due primarily to adding more computation on each client".*

## 2  Five Generations of Local Training Methods

We now offer several historical comments on the most important developments related to the *theoretical* understanding of LT. To this end, we have identified 5 distinct generations of LT methods, each with its unique challenges and characteristics. To make the narrative simple, and since we focus on this regime in our paper, we limit our overview to loss functions $\phi_i$ that are $\mu$-strongly convex and $L$-smooth. This is arguably the most studied class of functions in continuous optimization [Nesterov, 2004], and for this reason, it presents a valuable litmus test for any theory of LT.

### 2.1  Generation 1: Heuristic Age

While LT ideas were used in several machine learning domains before [Povey et al., 2015, Moritz et al., 2016], LT truly rose to prominence as a practically potent communication acceleration technique due to the seminal paper of McMahan et al. [2017] which introduced the FedAvg algorithm. However, no theory was provided in their work, nor in any prior work. LT-based heuristics, i.e., methods without any theoretical guarantees, dominated the initial development of the field up to, and including, the FedAvg paper.

---

[1]to datacenter optimization

## 2.2 Generation 2: Homogeneous Age

The first theoretical results for LT methods offering explicit convergence rates relied on various data/gradient *homogeneity*[2] assumptions. The intuitive rationale behind such assumptions comes from the following thought process. In the extreme case when all the local functions $\phi_i$ are *identical* (this is often referred to as the *homogeneous* or *i.i.d. data* regime), there is a very simple approach to making GD communication-efficient: push the idea of LT to its extreme by running GD on all clients, independently and in parallel, without any communication/synchronization/averaging whatsoever. Extrapolating from this, it is reasonable to assume that as we increase heterogeneity, taking multiple local steps should still be beneficial as long as we do not take too many steps. Several authors analyzed various LT methods under such assumptions, and obtained rates [Haddadpour and Mahdavi, 2019, Yu et al., 2019, Li et al., 2019, 2020b]. However, bounded dissimilarity assumptions are highly problematic. First, they do not seem to be satisfied even for some of the simplest function classes, such as strongly convex quadratics [Khaled et al., 2019, 2020], and moreover, it is well known that practical FL datasets are highly heterogeneous/non-i.i.d. McMahan et al. [2017], Kairouz et al. [2019]. So, analyses relying on such strong assumptions are both mathematically questionable, and practically irrelevant.

## 2.3 Generation 3: Sublinear Age

The third generation of LT methods is characterized by the successful removal of the bounded dissimilarity assumptions from the convergence theory. Khaled et al. [2019] first achieved this breakthrough by studying the simplest LT method: local gradient descent (LGD) (i.e., a simple combination of GD and LT). While works belonging to this generation elevated LT to the same theoretical footing as GD in terms of the assumptions, which marked an important milestone in our understanding of LT, unfortunately, the obtained communication complexity theory of LGD is pessimistic when compared to vanilla GD. Indeed, the inclusion of LT did *not* lead to an improvement upon the communication complexity of vanilla GD. Moreover, while GD enjoys a linear communication complexity (in the smooth and strongly convex regime), the communication complexity of LGD is *sublinear*. In a follow-up work, Khaled et al. [2020] later analyzed LGD in combination with DS as well. Woodworth et al. [2020] and Glasgow et al. [2022] provided lower bounds for LGD with DS showing that it is not better than minibatch SGD in heterogeneous setting. See the work of Malinovsky et al. [2020] for a fixed-point theory viewpoint.

## 2.4 Generation 4: Linear Age

The fourth generation of LT methods is characterized by the effort to design *linearly* converging variants of LT algorithms. In order to achieve this, it was important to tame the adverse effect of the so-called *client drift* [Karimireddy et al., 2020], which was identified as the culprit of the worse-than-GD theoretical performance of the previous generation of LT methods. The first LT-based method that successfully tamed client drift, and as a result obtained a linear convergence rate, was Scaffold [Karimireddy et al., 2020]. Several alternative approaches to obtaining the same effect were later proposed by Gorbunov et al. [2020a] and Mitra et al. [2021] . While obtaining a linear rate for LT methods under standard assumptions was a major achievement, the communication complexity of these methods is still somewhat worse[3] than that of vanilla GD, and is at best equal to that of GD.

## 2.5 Generation 5: Accelerated Age

Finally, the fifth generation of LT methods was initiated recently by Mishchenko et al. [2022] with their ProxSkip method which enjoys *accelerated communication complexity*. Acceleration comes from the LT steps coupled with a new client drift reduction technique and a probabilistic approach to deciding whether communication takes place or not. Mishchenko et al. [2022] first reformulate (1)

---

[2]We use the term *homogeneity* to refer to various related assumptions used in the literature, including *bounded gradient norms* [Li et al., 2020b], *bounded gradient variance* [Li et al., 2019, Yu et al., 2019] and *bounded gradient diversity* [Haddadpour and Mahdavi, 2019].

[3]Both GD, and LT methods such as Scaffold [Karimireddy et al., 2020], S-Local-GD [Gorbunov et al., 2020a] and FedLin [Mitra et al., 2021] enjoy the linear rate $\mathcal{O}(\kappa \log \frac{1}{\varepsilon})$, where $\kappa$ is a condition number. However, this condition number is in general slightly worse for the LT methods.

---

**Algorithm 1** ProxSkip-VR

---

1: **Parameters:** stepsize $\gamma > 0$, probability $p \in (0, 1]$, initial iterate $x_0 \in \mathbb{R}^d$, initial control vector $y_0 \in \mathbb{R}^d$, initial gradient shift $h_0 \in \mathbb{R}^d$, number of iterations $T \geq 1$
2: **for** $t = 0, 1, \ldots, T-1$ **do**
3:      $g_t = g(x_t, y_t, \xi_t)$          $\diamond$ Sample $\xi_t$ and construct an unbiased estimator of $\nabla f(x_t)$
4:      $\hat{x}_{t+1} = x_t - \gamma(g_t - h_t)$          $\diamond$ Take a gradient-type step adjusted via the shift $h_t$
5:      Construct new control vector $y_{t+1}$
6:      Flip a coin $\theta_t \in \{0, 1\}$ where $\mathrm{Prob}(\theta_t = 1) = p$    $\diamond$ Decides whether to skip the prox or not
7:      **if** $\theta_t = 1$ **then**
8:          $x_{t+1} = \mathrm{prox}_{\frac{\gamma}{p} r}\left(\hat{x}_{t+1} - \frac{\gamma}{p} h_t\right)$          $\diamond$ Apply prox, but only with probability $p$
9:      **else**
10:          $x_{t+1} = \hat{x}_{t+1}$          $\diamond$ Skip the prox!
11:      **end if**
12:      $h_{t+1} = h_t + \frac{p}{\gamma}(x_{t+1} - \hat{x}_{t+1})$          $\diamond$ Update the shift $h_t$
13: **end for**

---

into the equivalent consensus form

$$\min_{x \in \mathbb{R}^d} f(x) + r(x), \tag{3}$$

where $d = Md'$, $x = (x_1, \ldots, x_M) \in \mathbb{R}^d$, and

$$f(x) := \sum_{i=1}^{M} \frac{n}{n_i} \phi_i(x_i), \qquad r(x) = \begin{cases} 0 & \text{if } x_1 = \cdots = x_M, \\ +\infty & \text{otherwise.} \end{cases} \tag{4}$$

The ProxSkip method is a randomized variant of proximal gradient descent (ProxGD) [Nesterov, 2013, Beck, 2017] for solving (3), with the proximity operator of $r$, given by

$$\mathrm{prox}_r(x) := \arg\min_y \left( r(y) + \frac{1}{2}\|y - x\|^2 \right),$$

being evaluated in each iteration with probability $p \in (0, 1]$ only. Remarkably, Mishchenko et al. [2022] showed that it is possible to choose $p$ as low as $1/\sqrt{\kappa}$, where $\kappa$ is the condition number of $f$, without this worsening the rate of its parent method ProxGD. In summary, ProxSkip lets the $M$ clients perform $\sqrt{\kappa}$ local gradient steps in expectation, followed by the evaluation of the prox of $r$, which in the case of the consensus reformulation of (1) means averaging across all $M$ nodes.

## 3 ProxSkip-VR: A General Variance Reduction Framework for ProxSkip

In this work we contribute to the fifth generation of LT methods by extending the work of Mishchenko et al. [2022] to allow for a very large family of gradient estimators, including variance reduced (VR) ones [Johnson and Zhang, 2013, Defazio et al., 2014, Kovalev et al., 2020a, Mishchenko et al., 2019].

Like ProxSkip, our method ProxSkip-VR (Algorithm 1) is aimed to solve the composite problem (3) in a more general setting (see Assumptions 1–3), with the special structure (4) coming from the consensus reformulation being a special case only. Our method differs from ProxSkip in that we replace the gradient $\nabla f(x_t)$ by an unbiased estimator $g_t = g(x_t, y_t, \xi_t)$, where $\xi_t$ is the source of randomness controlling unbiasedness and $y_t$ is a control vector whose role is to progressively reduce the variance of the estimator, so that

$$\mathbb{E}\left[g_t \mid x_t, y_t\right] = \nabla f(x_t).$$

There are several motivations behind this endeavor. First, it is a-priori not clear whether the novel proof technique employed by Mishchenko et al. [2022] can be combined with the proof techniques used in the analysis of VR methods, and hence it is scientifically significant to investigate the possibility of such a merger of two strands of the literature. We show that this is possible. Second, marrying VR estimators with ProxSkip can lead to novel system architectures which are more elaborate than the simplistic client-server architecture (see Section 4). Lastly, while researchers contributing to generations 1–4 of LT methods were preoccupied with trying to close the gap on GD

in terms of communication efficiency, they *ignored* the number of the local steps appearing in their algorithms, and reported their bounds primarily in terms of the number of communication rounds. Bounds reported this way make complete sense in the scenario when the cost of local work (e.g., one SGD step w.r.t. a single data point), say $\delta$, is negligible compared to the cost of communication, which can w.l.o.g. assume to be 1, and when the number of local steps is small. With the advent of the fifth generation of LT methods, we can (to a large degree) stop worrying about communication efficiency, and can now ask more refined questions, such as:

> *Are there gradient estimators which, when combined with* ProxSkip, *lead to faster algorithms in terms of the total cost, which includes the communication cost as well as the cost of local training?*

We give an affirmative answer to the question in Sections 4 and 5.

## 3.1 Standard assumptions

We assume throughout that $f$ is differentiable, and let $D_f(x, y) := f(x) - f(y) - \langle \nabla f(y), x - y \rangle$ denote the Bregman divergence of $f$. Throughout the work we make the following assumptions:

**Assumption 1** (*L*-smoothness). *$\exists L > 0$ such that $2D_f(x, y) \leq L \|x - y\|^2$ for all $x, y \in \mathbb{R}^d$.*

**Assumption 2** (*$\mu$-convexity*). *$\exists \mu > 0$ such that $\mu \|x - y\|^2 \leq 2D_f(x, y)$ for all $x, y \in \mathbb{R}^d$.*

**Assumption 3.** *The regularizer $r : \mathbb{R}^d \to \mathbb{R} \cup \{+\infty\}$ is proper, closed and convex.*

Under the above assumptions, (3) has a unique minimizer $x_\star$. Let $h_\star := \nabla f(x_\star)$.

## 3.2 Modelling variance reduced gradient estimators

Our next assumption, initially introduced by Gorbunov et al. [2020b], postulates several parametric inequalities characterizing the behavior and ultimately the quality of a gradient estimator. Similar assumptions appeared later in [Gorbunov et al., 2020a,c].

**Assumption 4.** *Let $\{x_t\}$ be iterates produced by* ProxSkip-VR. *First, we assume that the stochastic gradients $g_t = g(x_t, y_t, \xi_t)$ are unbiased for all $t \geq 0$, namely*

$$\mathbb{E}[g_t \mid x_t, y_t] = \nabla f(x_t). \tag{5}$$

*Second, we assume that there exist non-negative constants $A, B, C, \tilde{A}, \tilde{B}, \tilde{C}$, with $\tilde{B} < 1$, and a nonnegative mapping $y_t \mapsto \sigma(y_t) := \sigma_t$ such that the following two relations hold for all $t \geq 0$,*

$$\mathbb{E}\left[\|g_t - \nabla f(x_*)\|^2 \mid x_t, y_t\right] \leq 2AD_f(x_t, x_*) + B\sigma_t + C, \tag{6}$$

$$\mathbb{E}[\sigma_{t+1} \mid x_t, y_t] \leq 2\tilde{A}D_f(x_t, x_*) + \tilde{B}\sigma_t + \tilde{C}. \tag{7}$$

Assumption 4 covers a very large collection of gradient estimators, including an infinite variety of subsampling/minibatch estimators, gradient sparsification and quantization estimators, and their combinations; see [Gorbunov et al., 2020b] for examples. VR estimators are characterized by $C = \tilde{C} = 0$; most non-VR estimators by $\tilde{A} = \tilde{B} = \tilde{C} = B = 0$ and $C > 0$ [Gower et al., 2019b].

## 3.3 Main result

We are now ready to formulate our main result.

**Theorem 5.** *Let Assumptions 2 and 3 hold, and let $g_t$ be a gradient estimator satisfying Assumption 4. If $B > 0$, choose any $W > B/(1-\tilde{B})$ and $\beta = (B+W\tilde{B})/W$. If $B = 0$, let $W = 0$ and $\beta = \tilde{B}$. Choose stepsize $0 < \gamma \leq \min\{1/\mu, 1/(A+W\tilde{A})\}$. Then the iterates of* ProxSkip-VR *for any $p \in (0, 1]$ satisfy*

$$\mathbb{E}[\Psi_T] \leq \max\left\{(1-\gamma\mu)^T, \beta^T, (1-p^2)^T\right\}\Psi_0 + \frac{\left(C + W\tilde{C}\right)\gamma^2}{\min\{\gamma\mu, p^2, 1-\beta\}}, \tag{8}$$

*where the Lyapunov function is defined by $\Psi_t := \|x_t - x_\star\|^2 + \frac{\gamma^2}{p^2}\|h_t - h_\star\|^2 + \gamma^2 W\sigma_t$.*

Table 2: Special cases of ProxSkip-VR, depending on the choice of the gradient estimator $g_t$.

| Estimator of $\nabla f$ | Communication Complexity of ProxSkip-VR | Iteration Complexity of ProxSkip-VR | Corollaries of Theorem 5 |
|---|---|---|---|
| GD [b] | $\mathcal{O}\left(\sqrt{L/\mu}\log 1/\varepsilon\right)$ | $\mathcal{O}\left(L/\mu\log 1/\varepsilon\right)$ | Theorem 6 |
| SGD [c] | $\mathcal{O}\left(\left(\sqrt{A/\mu}+\sqrt{2C/\varepsilon\mu^2}\right)\log 1/\varepsilon\right)$ | $\mathcal{O}\left(\left(A/\mu+2C/\varepsilon\mu^2\right)\log 1/\varepsilon\right)$ | Theorem 8 |
| HUB [NEW] | $\mathcal{O}\left(\sqrt{L_{\max}/\mu\left(1+\omega/\tau\right)}\log 1/\varepsilon\right)$ | $\mathcal{O}\left(L_{\max}/\mu\left(1+\omega/\tau\right)\log 1/\varepsilon\right)$ | Theorem 9 |
| LSVRG [NEW] | $\mathcal{O}\left(\sqrt{L(\tau)/\mu}\log 1/\varepsilon\right)$ | $\mathcal{O}\left(L(\tau)/\mu\log 1/\varepsilon\right)$ | Corollary 1 |
| Q [NEW] | $\mathcal{O}\left(\sqrt{L_{\max}/\mu\left(1+\omega/M\right)}\log 1/\varepsilon\right)$ | $\mathcal{O}\left(L_{\max}/\mu\left(1+\omega/M\right)\log 1/\varepsilon\right)$ | Corollary 2 |

[a] Any estimator satisfying Assumption 4
[b] ProxSkip-VR with the GD estimator reduces to the ProxSkip method of Mishchenko et al. [2022]
[c] ProxSkip-VR with the SGD estimator satisfying Assumption 7 reduces to the SProxSkip method of Mishchenko et al. [2022]
[d] $L(\tau) := \frac{m-\tau}{\tau(m-1)}L_{\max} + \frac{m(\tau-1)}{\tau(m-1)}L$, where $\tau$ is the mini-batch size and $m$ is the number of clients belonging to one hub

## 3.4 Two examples of gradient estimators

Here we give two illustrating examples of estimators satisfying Assumption 4.

**Theorem 6** (GD estimator). *Let Assumption 1, 2 and 3 hold. Then for the trivial estimator $g_t = \nabla f(x_t)$, Assumption 4 holds with the following parameters:*

$$A = L, \quad B = 0, \quad C = 0, \quad \tilde{A} = 0, \quad \tilde{B} = 0, \quad \tilde{C} = 0, \quad \sigma_t \equiv 0.$$

*Choose a stepsize satisfying $0 < \gamma \leq 1/L$. Then the iterates of ProxSkip-VR for any $p \in (0, 1]$ satisfy*

$$\mathbb{E}\left[\Psi_T\right] \leq \max\left\{(1-\gamma\mu)^T, (1-p^2)^T\right\}\Psi_0, \tag{9}$$

*where $\Psi_t := \|x_t - x_\star\|^2 + \frac{\gamma^2}{p^2}\|h_t - h_\star\|^2$. Let $\gamma = 1/L$ and $p = \sqrt{\mu/L}$ then the communication and iteration complexities of ProxSkip-VR are $\#comms = \mathcal{O}\left(\sqrt{\frac{L}{\mu}}\log\frac{1}{\varepsilon}\right), \#iters = \mathcal{O}\left(\frac{L}{\mu}\log\frac{1}{\varepsilon}\right).$*

This recovers the result obtained by Mishchenko et al. [2022] for their ProxSkip method. The next assumption holds for virtually all (non-VR) estimators based on subsampling [Gower et al., 2019b].

**Assumption 7** (Expected smoothness). *We say that an unbiased estimator $g(x; \xi) : \mathbb{R}^d \to \mathbb{R}^d$ of the gradient $\nabla f(x)$ satisfies the expected smoothness inequality if there exists $A'' > 0$ such that*

$$\mathbb{E}\left[\|g(x;\xi) - g(x_\star;\xi)\|^2\right] \leq 2A''D_f(x, x_\star), \quad \forall x \in \mathbb{R}^d.$$

**Theorem 8** (SGD estimator). *Let $g(x, \xi)$ satisfy Assumption 7 and define $g_t := g(x_t, \xi_t)$, where $\xi_t$ is chosen independently at time $t$. Then Assumption 4 holds with the following parameters:*

$$A = 2A'', \quad B = 0, \quad C = 2\mathrm{Var}(g(x_\star, \xi)), \quad \tilde{A} = 0, \quad \tilde{B} = 0, \quad \tilde{C} = 0, \quad \sigma_t \equiv 0.$$

*Moreover, assume that Assumption 2 holds. Choose stepsize $0 < \gamma \leq \min\{1/\mu, 1/A\}$. Then the iterates of ProxSkip-VR for any probability $p \in (0, 1]$ satisfy*

$$\mathbb{E}\left[\Psi_T\right] \leq \max\left\{(1-\gamma\mu)^T, (1-p^2)^T\right\}\Psi_0 + \gamma^2\frac{2\mathrm{Var}(g(x_\star, \xi))}{\min\{\gamma\mu, p^2\}}, \tag{10}$$

*where the Lyapunov function is defined by $\Psi_t := \|x_t - x_\star\|^2 + \frac{\gamma^2}{p^2}\|h_t - h_\star\|^2$. If we choose $\gamma = \min\{1/A, \varepsilon\mu/2C\}$ and $p = \sqrt{\gamma\mu}$ then the communication and iteration complexities of ProxSkip-VR are $\#comms = \mathcal{O}\left(\left(\sqrt{\frac{A}{\mu}} + \sqrt{\frac{C}{\varepsilon\mu^2}}\right)\log\frac{1}{\varepsilon}\right), \#iters = \mathcal{O}\left(\left(\frac{A}{\mu} + \frac{C}{\varepsilon\mu^2}\right)\log\frac{1}{\varepsilon}\right)$, respectively.*

This recovers the result obtained by Mishchenko et al. [2022] for their stochastic variant of ProxSkip, which they call SProxSkip.

## 4 New FL Architecture: Regional Hubs Connecting the Clients to the Server

We now illustrate the versatility of our ProxSkip-VR framework by designing a new "FL architecture" and proposing an algorithm that can efficiently operate in this setting.

In particular, we consider the situation where the clients are clustered (e.g., based on region), and where *a hub* is placed in between each cluster and the central server. Clients communicate with their regional hub only, which can communicate with the central server (see Figure 1). There are $M$ hubs, hub $i$ handles $n_i$ clients, and client $j$ associated with hub $i$ owns loss function $\phi_{ij}$.

Mathematically, this can be modeled by problem (1). In this situation, we care about two sources of communication cost: the server and the hubs, and between the hubs and the clients. We propose to handle this via *local training (LT)* between the server and the hubs, and via *client sampling (CS)* and *compressed communication (CC)* between the hubs and the clients. Algorithmically, from the server-hubs perspective, we are applying a particular variant of ProxSkip-VR to (3)–(4), where $\phi_i$ is the aggregate loss handles by hub $i$. This takes care of communication efficiency between the server and the hub. Note also that we need not worry about partial

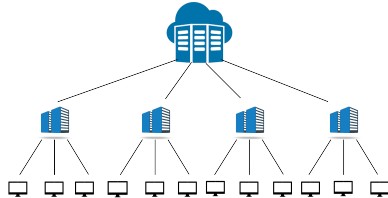

Figure 1: Server-hubs-clients FL architecture with 4 hubs and 12 clients.

participation of hubs, as these are designed to be always available. However, in this situation, it is costly for hub $i$ to compute the gradient of $\phi_i$ as this involves communication with all the clients it handles.

## 4.1 Handling of client sampling (CS) and compressed communication (CC)

In order to alleviate this burden, we propose a combination of CS and CC. However, we need to be very careful about how to do this. Indeed, both CS and CC, even when applied in isolation, and without ProxSkip in the mix, can lead to a substantial slowdown in convergence. For example, one will typically lose linear convergence in the strongly convex regime. However, techniques for preserving linear convergence in the presence of CS and CC exist: this is what variance reduction strategies are designed to do. For example, LSVRG [Hofmann et al., 2015, Kovalev et al., 2020a] is a VR technique for reducing the variance due to CS, and DIANA [Mishchenko et al., 2019] is a VR technique for reducing the variance due to CC. However, we are not aware of any VR method that combines CS (applied first) and CC (applied second).

We now propose such a technique. In iteration $t$, every hub $i \in \{1, 2, \ldots, M\}$ selects a random subset $\mathcal{S}_t^i \subseteq \{1, 2, \ldots, n_i\}$ of the clients it handles of cardinality $\tau_i$, chosen uniformly at random, and estimates the hub gradient via

$$\nabla \phi_i(x_t) \approx g_t^i := \frac{1}{|\mathcal{S}_t^i|} \sum_{j \in \mathcal{S}_t^i} \mathcal{Q}_t^{ij} \left( \nabla \phi_{ij}(x_t) - \nabla \phi_{ij}(y_t) \right) + \nabla \phi_i(y_t), \tag{11}$$

where $\mathcal{Q}_t^{ij} : \mathbb{R}^{d'} \to \mathbb{R}^{d'}$ is a randomized compression (e.g., sparsification or quantization) operator [Alistarh et al., 2017, Khirirat et al., 2018, Horváth et al., 2019b,a, Philippenko and Dieuleveut, 2020], i.e., a mapping satisfying

$$\mathbb{E}\left[ Q_t^{ij}(x) \right] = x, \quad \mathbb{E}\left[ \|Q_t^{ij}(x) - x\|^2 \right] \leq \omega \|x\|^2, \quad \forall x \in \mathbb{R}^{d'}, \tag{12}$$

and the control vector $y_t$ is updated probabilistically as follows:

$$y_{t+1} = \begin{cases} x_t & \text{with probability} \quad q \\ y_t & \text{with probability} \quad 1-q \end{cases}. \tag{13}$$

The global gradient estimator (a vector in $\mathbb{R}^{Md'}$), which we call HUB, is constructed as a concatenation of the above hub estimators:

$$\nabla f(x_t) := \left( \frac{n_i}{n} \nabla \phi_i(x_t) \right)_{i=1}^M \approx g_t := g(x_t, y_t, \xi_t) := \left( \frac{n_i}{n} g_t^i \right)_{i=1}^M, \tag{14}$$

where $\xi_t$ represents the combined randomness from the compressors $\{\mathcal{Q}_t^{ij}\}$ and random sets $\{\mathcal{S}_t^i\}$.

In order to analyze ProxSkip-VR in the consensus form, from now on we assume that $n_i = m = {}^n/_M$ and $\tau_i = \tau \in \{1, 2, \ldots, m\}$ for all $i$, and rely on a slightly different, more general reformulation:

$$\min_{x \in \mathbb{R}^d} \frac{1}{m} \sum_{j=1}^m \widetilde{\phi}_j(x) + r(x), \quad \widetilde{\phi}_j(x) := \frac{1}{M} \sum_{i=1}^M \phi_{ij}(x_i), \quad r(x) := \begin{cases} 0 & \text{if } x_1 = \cdots = x_M, \\ +\infty & \text{otherwise.} \end{cases}$$

Our proposed method ProxSkip-HUB is ProxSkip-VR combined with the novel HUB estimator (14), applied to the above reformulation; see Algorithm 2.

## 4.2 Theory for ProxSkip-HUB

In the following result we first claim that the above estimator satisfies Assumption 4 with $C = \tilde{C} = 0$ (i.e., it is variance-reduced), and the rest of the claim follows by application of our general theorem Theorem 5.

**Theorem 9.** *Assume that $\nabla \widetilde{\phi}_j$ is $L_j$-smooth for all $j$ and let Assumptions 2 and 3 hold. Then for the gradient estimator* (14), *Assumption 4 holds with the following constants:*

$$A = 4\left(L(\tau) + \frac{\omega}{\tau}L_{\max}\right), \quad B = 4\left(1 + \frac{\omega}{\tau}\right), \quad C = 0, \quad \tilde{A} = qL_{\max}, \quad \tilde{B} = 1 - q, \quad \tilde{C} = 0,$$

*and $\sigma_t := \sigma(y_t)$, $\sigma(y) := \frac{1}{m}\sum_{j=1}^m \|\nabla\widetilde{\phi}_j(y) - \nabla\widetilde{\phi}_j(x_\star)\|^2$, $L_{\max} := \max_j L_j$. Set $W = 2B/(1-\tilde{B})$ and $0 < \gamma \le \min\{1/\mu, 1/(A+W\tilde{A})\}$. Then the iterates of* ProxSkip-VR *for any $p \in (0,1]$ satisfy*

$$\mathbb{E}\left[\Psi_T\right] \le \max\left\{(1-\gamma\mu)^T, (1-p^2)^T, \left(1 - \frac{q}{2}\right)^T\right\}\Psi_0,$$

*where the Lyapunov function is defined by $\Psi_t := \|x_t - x_\star\|^2 + \frac{\gamma^2}{p^2}\|h_t - h_\star\|^2 + \gamma^2\frac{8}{q}\left(1 + \frac{\omega}{\tau}\right)\sigma_t$.*

We now consider two special cases. In the first, we specialize to the no compression regime, and in the second, to the full participation regime.

**Corollary 1** (No compression). *If we do not use compression (i.e., $\omega = 0$), then the communication and iteration complexities are $\#comms = \mathcal{O}\left(\sqrt{\frac{L_{\max}}{\mu}}\log\frac{1}{\varepsilon}\right), \#iters = \mathcal{O}\left(\frac{L_{\max}}{\mu}\log\frac{1}{\varepsilon}\right)$, respectively. However, if we use the estimator* (11) *in Theorem 5 directly, then the communication and iteration complexities are $\#comms = \mathcal{O}\left(\sqrt{\frac{L(\tau)}{\mu}}\log\frac{1}{\varepsilon}\right), \#iters = \mathcal{O}\left(\frac{L(\tau)}{\mu}\log\frac{1}{\varepsilon}\right)$, where $L(\tau) := \frac{m-\tau}{\tau(m-1)}L_{\max} + \frac{m(\tau-1)}{\tau(m-1)}L$.*

Notice that $L_{\max} \ge L$, and that $L(\tau) = L_{\max}$ for $\tau = 1$ and $L(\tau) = L$ for $\tau = m$. Moreover, $L(\tau)$ decreases as the minibatch size $\tau$ increases.

**Corollary 2** (No client sampling). *If we do not use client sampling (i.e., $\tau = m$), then the communication and iteration complexities are $\#comms = \mathcal{O}\left(\sqrt{\frac{L_{\max}}{\mu}\left(1 + \frac{\omega}{M}\right)}\log\frac{1}{\varepsilon}\right), \#iters = \mathcal{O}\left(\frac{L_{\max}}{\mu}\left(1 + \frac{\omega}{M}\right)\log\frac{1}{\varepsilon}\right)$, respectively.*

Assume $r(x) \equiv 0$. If $\mathcal{Q}_t^{ij}(x) \equiv x$, then $\omega = 0$, and we restore the well-known LSVRG method [Hofmann et al., 2015, Kovalev et al., 2020b], assuming that the functions $\phi_{ij}$ have the same smoothness constant. We can recover the same rate exactly as well, but with a slightly more refined analysis, one in which we do not need to work with compressors (LSVRG does not involve any), which makes for a tighter analysis. If $\tau = m$, we restore the rate of the well-known DIANA [Mishchenko et al., 2019, Horváth et al., 2019b] method and its sibling Rand-DIANA [Shulgin and Richtárik, 2021].

## 5 Experiments

To illustrate the predictive power of our theory, it suffices to consider $L_2$-regularized logistic regression in the distributed setting (1), with

$$\phi_i(x) = \frac{1}{n_i}\sum_{j=1}^{n_i}\log\left(1 + \exp\left(-b_{ij}a_{ij}^\top x\right)\right) + \frac{\lambda}{2}\|x\|^2,$$

where $n_i$ is the number of data points per worker $a_{ij} \in \mathbb{R}^{d'}$ and $b_{ij} \in \{-1, +1\}$ are the data samples and labels. We choose $n_i = m = n/M$ for all $i$. We set the regularization parameter $\lambda = 5 \cdot 10^{-4}L$ by default, where $L$ is the smoothness constant of $f$. We conduct several experiments on the w8a dataset from LibSVM library [Chang and Lin, 2011]. In Figure 2 (first row), we compare various LT baselines[4] for three choices of mini-batch sizes ($\tau = 16, 32, 64$) with our method ProxSkip-VR combined with the LSVRG estimator, which is a special case of the HUB estimator when

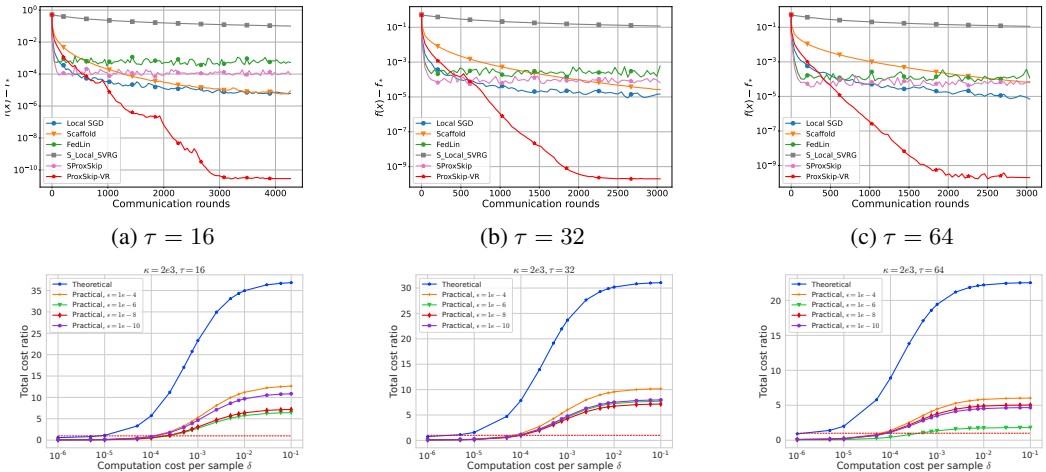

Figure 2: The top row shows the convergence results compared with baselines and the second row is the total cost ratio of ProxSkip over our ProxSkip-VR.

$\mathcal{Q}_t^{ij}(x) \equiv x$ for all $i, j$. We see that our method outperforms all other methods significantly due to its communication-acceleration properties.

Next, we derive the total cost, which includes communication cost (assumed to be 1, for normalization purposes), and computational cost (assumed to be $\delta$; and equal to the cost of performing one SGD step with a single data point). Let us consider the total cost of ProxSkip-VR in the case of the LSVRG estimator: in each iteration we compute 1 stochastic gradient, and with probability $q$ we compute the exact gradient. We do not need to compute a second stochastic gradient since we can use memory and the relation $y_{t+1} = x_t$. The total cost for ProxSkip-VR is equal to

$$\text{Cost}(\text{ProxSkip-VR}) \coloneqq T_{\text{comm.}}(\text{ProxSkip-VR}) + \delta \left(qm + (1-q)\tau + \tau\right) T_{\text{iter}}(\text{ProxSkip-VR}).$$

ProxSkip requires full/exact gradient computation at each iteration, so the total cost of ProxSkip is

$$\text{Cost}(\text{ProxSkip}) \coloneqq T_{\text{comm.}}(\text{ProxSkip}) + \delta m T_{\text{iter}}(\text{ProxSkip}).$$

Using Theorems 6 and 9 and the value $L(\tau) \coloneqq \frac{m-\tau}{\tau(m-1)} L_{\max} + \frac{m(\tau-1)}{\tau(m-1)} L$ of the expected smoothness constant for sampling with minibatch size $\tau$ uniformly at random, we get the following expression for the cost ratio, expressed as a function of $\delta$:

$$\text{Cost ratio}(\delta) \coloneqq \frac{\text{Cost}(\text{ProxSkip})}{\text{Cost}(\text{ProxSkip-VR})} = \frac{\sqrt{\mu L} + mL\delta}{\sqrt{\mu L(\tau)} + (2m\mu + (2L(\tau) - 2\mu)\tau)\,\delta}. \quad (15)$$

We can easily calculate the limits of this expression:

$$\text{Cost ratio}(\delta = 0) = \sqrt{\frac{L}{L(\tau)}}, \qquad \text{Cost ratio}(\delta \to \infty) = \frac{mL}{2(m\mu + (L(\tau) - \mu)\,\tau)}.$$

In Figure 2 (second row), we depict the theoretical cost ratio according to (15) and the corresponding experimental cost ratio obtained by an actual run of both methods to achieve $\varepsilon$-accuracy, with $\varepsilon = 10^{-6}$ and $\varepsilon = 10^{-8}$. Remarkably, the experimental results follows the same pattern as our theoretical prediction. The empirical curves appear lower because we use approximations for $L_{\max}$, $L$ and $\mu$. As we can see, starting from $\delta = 10^{-4}$, ProxSkip-VR starts to outperform ProxSkip. As $\delta$ grows, the advantage of variance reduction embedded in ProxSkip-VR over vanilla ProxSkip grows. These results suggest that variance reduction is of practical utility in terms of the total cost, even for small values of $\delta$, and its effectiveness grows with $\delta$.

## Acknowledgements

We would like to thank Eduard Gorbunov for useful discussions related to some aspects of the theory.

---

[4]With the exception of LocalSGD, all use client drift correction.

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
