# Appendix

## A   Supplementary Experiments

### A.1   Additional experiments with ProxSkip-VR

We conduct further experiments to validate the efficiency of our proposed method ProxSkip-VR. We instantiate our variance reduction design with LSVRG and compare our method with several baselines across various datasets (w8a/a9a), different number of workers (10/20), and different batch sizes (16/32/64). All results in Figures 3, 4, 5, 6 show that ProxSkip-VR achieves linear convergence and outperforms the baselines.

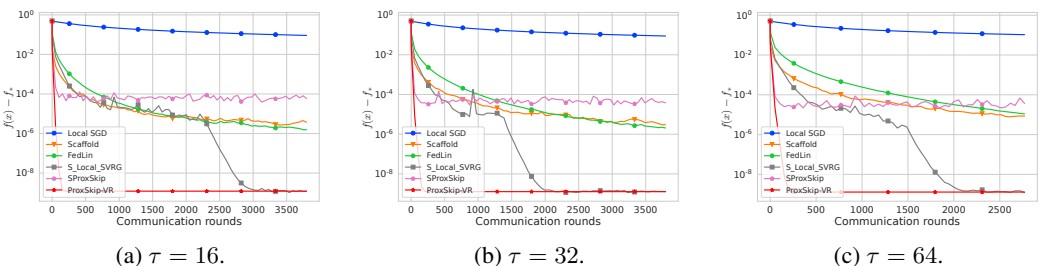

(a) $\tau = 16$.          (b) $\tau = 32$.          (c) $\tau = 64$.

Figure 3: Convergence results with 20 distributed workers on w8a dataset, $\kappa = 1e3$.

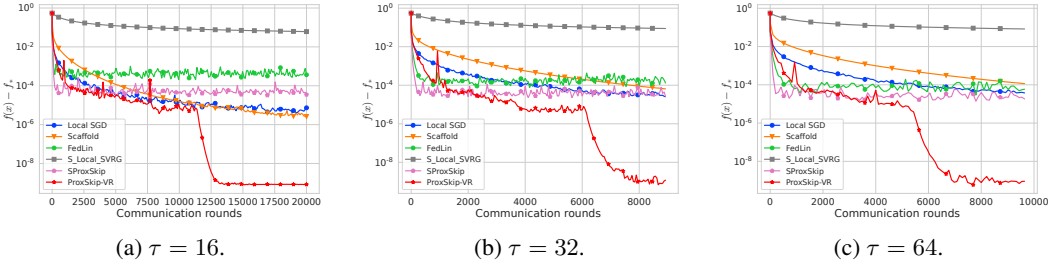

(a) $\tau = 16$.          (b) $\tau = 32$.          (c) $\tau = 64$.

Figure 4: Convergence results with 20 distributed workers on w8a dataset, $\kappa = 1e4$.

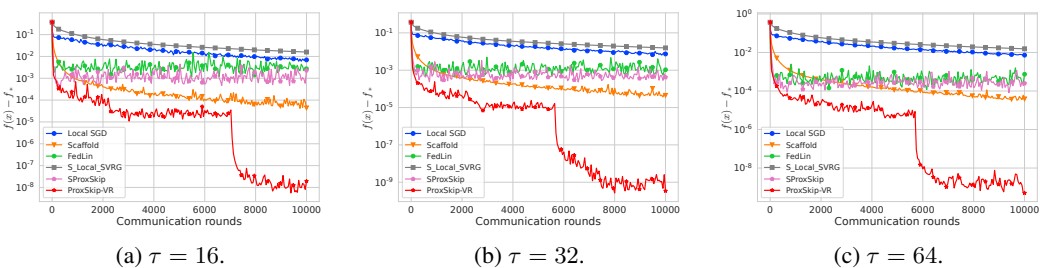

(a) $\tau = 16$.          (b) $\tau = 32$.          (c) $\tau = 64$.

Figure 5: Convergence results with 10 distributed workers on a9a dataset, $\kappa = 1e4$.

### A.2   Additional experiments of cost ratio of ProxSkip over ProxSkip-VR

Here we report on more experiments related to the total cost ratio of ProxSkip over ProxSkip-VR in Figures 7, 8, 9, 10. When GD is used as the subroutine, i.e., when ProxSkip-VR reduces to ProxSkip, the ratio is equal to one, and is depicted by the red horizontal dashed line. Any cost ratio above one means that ProxSkip-VR benefits over ProxSkip (e.g., a cost ratio of 10 means $10\times$ speedup in favor of our method). As seen in the plots, acceleration of our method over ProxSkip is clearly visible, and improves as the local computation cost $\delta$ per sample increases. For large values of $\delta$ (i.e., $\delta \approx 10^{-1}$),

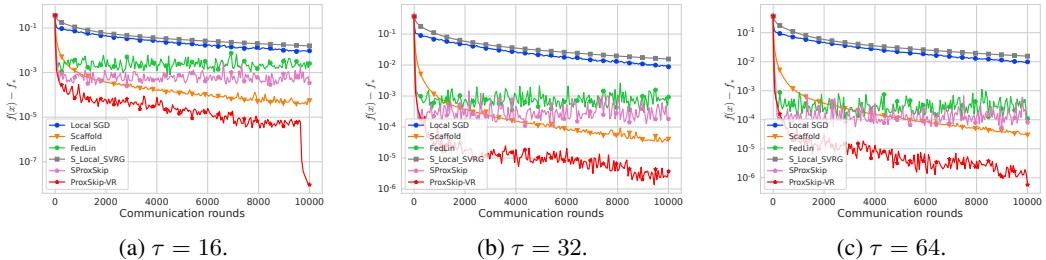

(a) $\tau = 16$.      (b) $\tau = 32$.      (c) $\tau = 64$.

Figure 6: Convergence results with 20 distributed workers on a9a dataset, $\kappa = 1e4$.

the acceleration can reach $20\times$ to $85\times$. Note also that acceleration is more significant for small minibatch sizes. That is, it is better for $\tau = 16$ than for $\tau = 32$, which is better than in the $\tau = 64$ case. This means that in terms of total cost, it is beneficial for the workers to use smaller minibatch sizes, i.e., it is beneficial to be as far from the full batch regime employed by ProxSkip as possible.

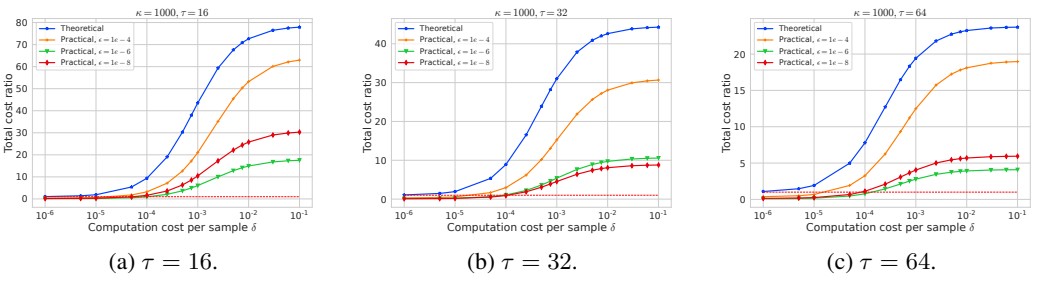

(a) $\tau = 16$.      (b) $\tau = 32$.      (c) $\tau = 64$.

Figure 7: Acceleration with 10 distributed workers on a9a dataset, $\kappa = 1e3$.

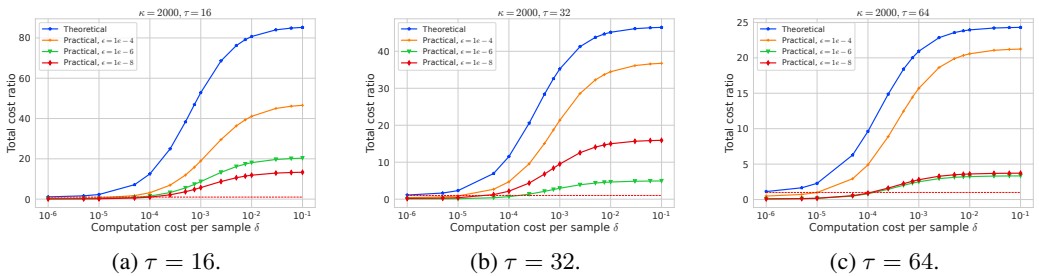

(a) $\tau = 16$.      (b) $\tau = 32$.      (c) $\tau = 64$.

Figure 8: Acceleration with 10 distributed workers on a9a dataset, $\kappa = 2e3$.

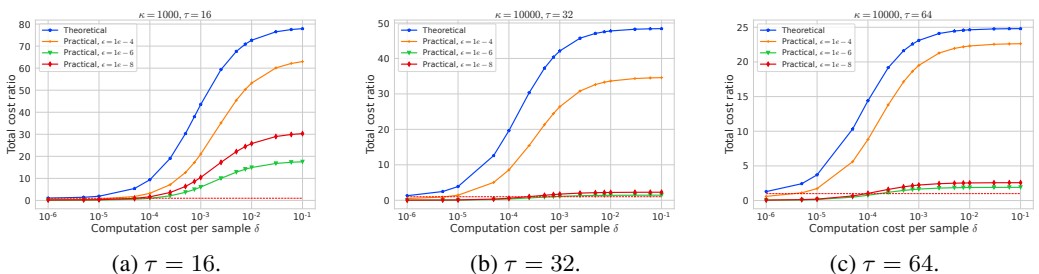

(a) $\tau = 16$.      (b) $\tau = 32$.      (c) $\tau = 64$.

Figure 9: Acceleration with 10 distributed workers on a9a dataset, $\kappa = 1e4$.

## A.3 Experiments with ProxSkip-HUB

In this work we introduced a new FL architecture: regional hubs connecting the clients to the server; see Section 4. For conceptual simplicity, and in order to facilitate fair comparison with ProxSkip-

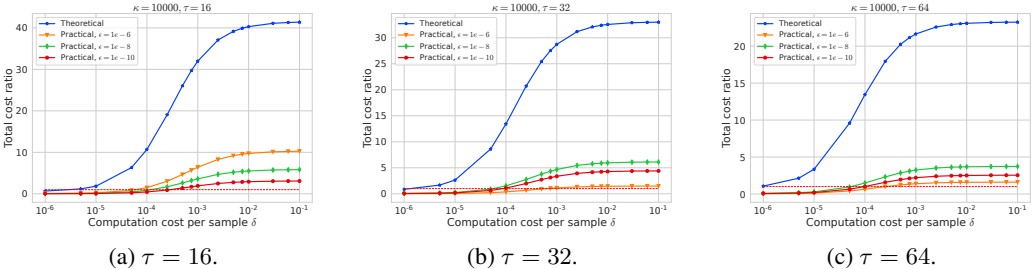

Figure 10: Acceleration with 10 distributed workers on `w8a` dataset, $\kappa = 1e4$.

LSVRG, we assume that the number of hubs equals to the number of clients, and that each client owns a single datapoint only. We compare ProxSkip-HUB with ProxSkip-LSVRG to check whether communication compression leads to any benefits in terms of total costs. Theoretically, and similarly to our analysis in Section 5, the total cost for ProxSkip-LSVRG is

$$\text{Cost}(\text{ProxSkip-LSVRG}) \coloneqq T_{\text{comm.}}(\text{ProxSkip-LSVRG}) + \delta \left( qm + (1-q)\tau + \tau \right) T_{\text{iter}}(\text{ProxSkip-LSVRG}).$$

Recall that we assume the communication cost from every worker/hub to the master is equal to 1, and the computation cost per sample is equal to $\delta$. Here we generalize to the multi-level structure. We assume that the communication cost from every client to hub is equal to $\delta'$. We choose the Rand-$k$ sparsification for ProxSkip-HUB; this compressor selects $k$-entries of the gradient vector, uniformly at random from the full $d$-dimensional gradient. The total cost of ProxSkip-HUB is

$$\begin{aligned}
\text{Cost}(\text{ProxSkip-HUB}) \coloneqq{} & T_{\text{comm.}}(\text{ProxSkip-HUB}) \\
& + \delta' \left( qm + \frac{k}{d} \left( (1-q)\tau + \tau \right) \right) T_{\text{iter}}(\text{ProxSkip-HUB}).
\end{aligned} \quad (16)$$

---

**Algorithm 2** ProxSkip-HUB

1: **Input**: stepsize $\gamma > 0$, probabilities $p > 0$, $q > 0$, initial iterate $x_0 \in \mathbb{R}^d$, initial shift $y_0 \in \mathbb{R}^d$, initial control variate $h_0 \in \mathbb{R}^d$, number of iterations $T \geq 1$
2: **for** $t = 0, 1, \dots, T-1$ **do**
3:      broadcast $x_t$ to all clients
4:      **for** $i \in S_t$ **do**
5:          $\hat{\Delta}_t^i = Q \left( \nabla f_i(x_t) - \nabla f_i(y_t) \right)$                           $\diamond$ Apply compression operator
6:      **end for**
7:      $\hat{\Delta}_t = \frac{1}{\tau} \sum_{i \in S_t} \hat{\Delta}_t^i$
8:      $\hat{g} = \hat{\Delta}_t + \nabla f(y_t)$
9:      $\hat{x}_{t+1} = x_t - \gamma(\hat{g}_t - h_t)$       $\diamond$ Take a gradient-type step adjusted via the control variate $h_t$
10:     Flip a coin $\theta_t \in \{0, 1\}$ where $\text{Prob}(\theta_t = 1) = p$ $\diamond$ To decide whether to skip the prox or not
11:     **if** $\theta_t = 1$ **then**
12:         $x_{t+1} = \text{prox}_{\frac{\gamma}{p}\psi} \left( \hat{x}_{t+1} - \frac{\gamma}{p} h_t \right)$     $\diamond$ Apply prox, but only very rarely! (with probability $p$)
13:     **else**
14:         $x_{t+1} = \hat{x}_{t+1}$                                                 $\diamond$ Skip the prox!
15:     **end if**
16:     $h_{t+1} = h_t + \frac{p}{\gamma}(x_{t+1} - \hat{x}_{t+1})$                         $\diamond$ Update the control variate $h_t$
17:     $y_{t+1} = \begin{cases} x_t & \text{with probability} & q \\ y_t & \text{with probability} & 1-q \end{cases}$            $\diamond$ Update the shift $y_t$
18: **end for**

---

Our experimental results are summarized Figure 11; we use the values $\delta = \delta' = 10^{-2}$. Clearly, and thanks to communication compression, ProxSkip-HUB has benefit in terms of the total cost compared to ProxSkip-LSVRG, and can reach up to three degrees of magnitude!

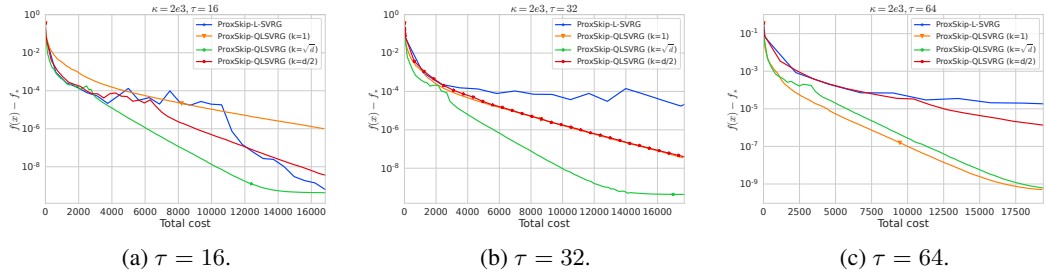

(a) $\tau = 16$.         (b) $\tau = 32$.         (c) $\tau = 64$.

Figure 11: Convergence results with different batch sizes and sparsification parameter $k$ on `a9a`, $\kappa = 2e3$.

# B  Basic Facts

## B.1  Bregman divergence, $L$-smoothness and $\mu$-strong convexity

The Bregman divergence of a differentiable function $f : \mathbb{R}^d \to \mathbb{R}$ is defined by

$$D_f(x, y) := f(x) - f(y) - \langle \nabla f(y), x - y \rangle. \tag{17}$$

It is easy to see that

$$\langle \nabla f(x) - \nabla f(y), x - y \rangle = D_f(x, y) + D_f(y, x), \quad \forall x, y \in \mathbb{R}^d. \tag{18}$$

For an $L$-smooth and $\mu$-strongly convex function $f : \mathbb{R}^d \to \mathbb{R}$, we have

$$\frac{\mu}{2}\|x - y\|^2 \le D_f(x, y) \le \frac{L}{2}\|x - y\|^2, \quad \forall x, y \in \mathbb{R}^d \tag{19}$$

and

$$\frac{1}{2L}\|\nabla f(x) - \nabla f(y)\|^2 \le D_f(x, y) \le \frac{1}{2\mu}\|\nabla f(x) - \nabla f(y)\|^2, \quad \forall x, y \in \mathbb{R}^d. \tag{20}$$

## B.2  Firm-nonexpansiveness of the proximity operator

Given $\psi : \mathbb{R}^d \to \mathbb{R}$, we define $\psi^*(y) := \sup_{x \in \mathbb{R}^d}\{\langle x, y \rangle - \psi(x)\}$ to be its Fenchel conjugate. The proximity operator of $\psi^*$ satisfies for any $\tau > 0$

$$\text{if } u = \operatorname{prox}_{\tau \psi^*}(y), \quad \text{then} \quad u \in y - \tau \partial \psi^*(u). \tag{21}$$

If Assumption 3 is satisfied, then firm nonexpansiveness of the proximity operator implies [Mishchenko et al., 2022] that

$$\left\|\operatorname{prox}_{\frac{\gamma}{p}\psi}(x) - \operatorname{prox}_{\frac{\gamma}{p}\psi}(y)\right\|^2 + \left\|\left(x - \operatorname{prox}_{\frac{\gamma}{p}\psi}(x)\right) - \left(y - \operatorname{prox}_{\frac{\gamma}{p}\psi}(y)\right)\right\|^2 \le \|x - y\|^2, \tag{22}$$

for all $x, y \in \mathbb{R}^d$ and any $\gamma, p > 0$.

## B.3  Young's inequality

For any two vectors $a, b \in \mathbb{R}^d$, we have

$$\|a + b\|^2 \le 2\|a\|^2 + 2\|b\|^2. \tag{23}$$

## B.4  Jensen's inequality

For a convex function $h : \mathbb{R}^d \leftarrow \mathbb{R}$ and any vectors $x_1, \ldots, x_n \in \mathbb{R}^d$, we have

$$h\left(\frac{1}{n}\sum_{i=1}^{n} x_i\right) \le \frac{1}{n}\sum_{i=1}^{n} h(x_i). \tag{24}$$

Applying this to the squared norm, $h(x) = \|x\|^2$, we get

$$\left\|\frac{1}{n}\sum_{i=1}^{n} y_i\right\|^2 \le \frac{1}{n}\sum_{i=1}^{n}\|y_i\|^2. \tag{25}$$

# C Analysis of ProxSkip-VR

In this section we provide the proof of Theorem 5.

## C.1 Main lemma of ProxSkip

We start from Lemma 1 initially introduced by Mishchenko et al. [2022]; for completeness we provide the whole proof. Let us define two additional sequences:

$$\hat{w}_t = x_t - \gamma \hat{g}_t(x_t), \qquad \hat{w}_\star = x_\star - \gamma \hat{g}_t(x_\star). \tag{26}$$

**Lemma 1.** *If Assumption 3 holds, $\gamma > 0$ and $0 < p \le 1$, then the iterates of ProxSkip-VR satisfy*

$$\mathbb{E}\left[\|x_{t+1} - x_\star\|^2 + \frac{\gamma^2}{p^2}\|h_{t+1} - h_\star\|^2\right] \le \|\hat{w}_t - w_\star\|^2 + \left(1 - p^2\right)\frac{\gamma^2}{p^2}\|h_t - h_\star\|^2. \tag{27}$$

*Proof.* In order to simplify, let us define two points:

$$x := \hat{x}_{t+1} - \frac{\gamma}{p}h_t, \quad y := x_\star - \frac{\gamma}{p}h_\star. \tag{28}$$

**STEP 1 (Optimality conditions).** Using the first-order optimality conditions for $f + \psi$ and using $h_\star := \nabla f(x_\star)$, we obtain the following fixed-point identity for $x_\star$:

$$x_\star = \mathrm{prox}_{\frac{\gamma}{p}\psi}\left(x_\star - \frac{\gamma}{p}h_\star\right) = \mathrm{prox}_{\frac{\gamma}{p}\psi}(y). \tag{29}$$

**STEP 2 (Recalling the steps of the method).** Recall that the vectors $x_t$ and $h_t$ are in Algorithm 1 updated as follows:

$$x_{t+1} = \begin{cases} \mathrm{prox}_{\frac{\gamma}{p}\psi}(x) & \text{with probability} \quad p \\ \hat{x}_{t+1} & \text{with probability} \quad 1 - p \end{cases} \tag{30}$$

and

$$h_{t+1} = h_t + \frac{p}{\gamma}(x_{t+1} - \hat{x}_{t+1}) = \begin{cases} h_t + \frac{p}{\gamma}\left(\mathrm{prox}_{\frac{\gamma}{p}\psi}(x) - \hat{x}_{t+1}\right) & \text{with probability } p \\ h_t & \text{with probability } 1 - p \end{cases}. \tag{31}$$

Let us consider the expected value $V_{t+1} := \mathbb{E}\left[\|x_{t+1} - x_\star\|^2 + \frac{\gamma^2}{p^2}\|h_{t+1} - h_\star\|^2\right]$:

$$
\begin{aligned}
V_{t+1} \overset{(30)+(31)}{=} \quad & p\left(\left\|\mathrm{prox}_{\frac{\gamma}{p}\psi}(x) - x_\star\right\|^2 + \frac{\gamma^2}{p^2}\left\|h_t + \frac{p}{\gamma}\left(\mathrm{prox}_{\frac{\gamma}{p}\psi}(x) - \hat{x}_{t+1}\right) - h_\star\right\|^2\right) \\
& + (1-p)\left(\|\hat{x}_{t+1} - x_\star\|^2 + \frac{\gamma^2}{p^2}\|h_t - h_\star\|^2\right) \\
\overset{(29)}{=} \quad & p\left(\left\|\mathrm{prox}_{\frac{\gamma}{p}\psi}(x) - \mathrm{prox}_{\frac{\gamma}{p}\psi}(y)\right\|^2 + \left\|\frac{\gamma}{p}h_t + \mathrm{prox}_{\frac{\gamma}{p}\psi}(x) - \hat{x}_{t+1} - \frac{\gamma}{p}h_\star\right\|^2\right) \\
& + (1-p)\left(\|\hat{x}_{t+1} - x_\star\|^2 + \frac{\gamma^2}{p^2}\|h_t - h_\star\|^2\right) \\
\overset{(28)+(29)}{=} \quad & p\left(\left\|\mathrm{prox}_{\frac{\gamma}{p}\psi}(x) - \mathrm{prox}_{\frac{\gamma}{p}\psi}(y)\right\|^2 + \left\|\mathrm{prox}_{\frac{\gamma}{p}\psi}(x) - x + y - \mathrm{prox}_{\frac{\gamma}{p}\psi}(y)\right\|^2\right) \\
& + (1-p)\left(\|\hat{x}_{t+1} - x_\star\|^2 + \frac{\gamma^2}{p^2}\|h_t - h_\star\|^2\right).
\end{aligned} \tag{32}
$$

**STEP 4 (Applying firm nonexpansiveness).** Applying firm nonexpansiveness of the proximal operator (22), this leads to the inequality

$$V_{t+1} \overset{(32)+(22)}{\leq} p\|x - y\|^2 + (1-p)\left(\|\hat{x}_{t+1} - x_\star\|^2 + \frac{\gamma^2}{p^2}\|h_t - h_\star\|^2\right)$$

$$\overset{(28)}{=} p\left\|\hat{x}_{t+1} - \frac{\gamma}{p}h_t - \left(x_\star - \frac{\gamma}{p}h_\star\right)\right\|^2 + (1-p)\left(\|\hat{x}_{t+1} - x_\star\|^2 + \frac{\gamma^2}{p^2}\|h_t - h_\star\|^2\right).$$

**STEP 5 (Simple algebra).** Next, we expand the squared norm and collect the terms, obtaining

$$
\begin{aligned}
V_{t+1} &\leq p\|\hat{x}_{t+1} - x_\star\|^2 + p\frac{\gamma^2}{p^2}\|h_t - h_\star\|^2 - 2\gamma\langle\hat{x}_{t+1} - x_\star, h_t - h_\star\rangle \\
&\quad + (1-p)\left(\|\hat{x}_{t+1} - x_\star\|^2 + \frac{\gamma^2}{p^2}\|h_t - h_\star\|^2\right) \\
&= \|\hat{x}_{t+1} - x_\star\|^2 - 2\gamma\langle\hat{x}_{t+1} - x_\star, h_t - h_\star\rangle + \frac{\gamma^2}{p^2}\|h_t - h_\star\|^2. \qquad (33)
\end{aligned}
$$

Finally, note that by our definition of $\hat{w}_t$, we have the identity $\hat{x}_{t+1} = \hat{w}_t + \gamma h_t$. Therefore, the first two terms above can be rewritten as

$$
\begin{aligned}
\|\hat{x}_{t+1} - x_\star\|^2 - 2\gamma\langle\hat{x}_{t+1} - x_\star, h_t - h_\star\rangle &= \|\hat{w}_t - w_\star + \gamma(h_t - h_\star)\|^2 \\
&\quad - 2\gamma\langle\hat{w}_t - w_\star + \gamma(h_t - h_\star), h_t - h_\star\rangle \\
&= \|\hat{w}_t - w_\star\|^2 + 2\gamma\langle\hat{w}_t - w_\star, h_t - h_\star\rangle \\
&\quad + \gamma^2\|h_t - h_\star\|^2 - 2\gamma\langle\hat{w}_t - w_\star, h_t - h_\star\rangle \\
&\quad - 2\gamma^2\|h_t - h_\star\|^2 \\
&= \|\hat{w}_t - w_\star\|^2 - \gamma^2\|h_t - h_\star\|^2. \qquad (34)
\end{aligned}
$$

Finally, plugging (34) into (33), we get:

$$V_{t+1} \leq \|\hat{w}_t - w_\star\|^2 + \left(1 - p^2\right)\frac{\gamma^2}{p^2}\|h_t - h_\star\|^2.$$

$\square$

## C.2 Main lemma

This lemma allows us to obtain a useful recursion for variance-reduced stochastic estimators used in our ProxSkip-VR algorithm.

**Lemma 2.** *Let Assumptions 2 and 4 hold. Then the iterates of* ProxSkip-VR *satisfy*

$$\mathbb{E}\left[\|\hat{w}_t - w_\star\|^2\right] \leq (1 - \gamma\mu)\|x_t - x_\star\|^2 - 2\gamma D_f(x_t, x_\star)(1 - \gamma A) + \gamma^2 B\sigma_t + \gamma^2 C.$$

*Proof.* We start from the definitions of the auxiliary sequence $\hat{w}_t$ (see (26)):

$$
\begin{aligned}
\|\hat{w}_t - w_\star\|^2 &\overset{(26)}{=} \|x_t - \gamma\hat{g}_t - (x_\star - \gamma\nabla f(x_\star))\|^2 \\
&= \|(x_t - x_\star) - \gamma(\hat{g}_t - \nabla f(x_\star))\|^2 \\
&= \|x_t - x_\star\|^2 - 2\gamma\langle x_t - x_\star, \hat{g}_t - \nabla f(x_\star)\rangle + \gamma^2\|\hat{g}_t - \nabla f(x_\star)\|^2. \quad (35)
\end{aligned}
$$

Taking expectation in (35) and using unbiasedness of $\hat{g}_t$ (see (5) in Assumption 4), we get

$$\mathbb{E}\left[\|\hat{w}_t - w_\star\|^2\right] \overset{(35)+(5)}{=} \|x_t - x_\star\|^2 - 2\gamma\langle x_t - x_\star, \nabla f(x_t) - \nabla f(x_\star)\rangle + \gamma^2\mathbb{E}\left[\|\hat{g}_t - \nabla f(x_\star)\|^2\right]. \quad (36)$$

Let us now consider the inner product in (36). Using (19) and (18), we obtain

$$\mathbb{E}\left[\|\hat{w}_t - w_\star\|^2\right] \leq (1 - \gamma\mu)\|x_t - x_\star\|^2 - 2\gamma D_f(x_t, x_\star) + \gamma^2\mathbb{E}\left[\|\hat{g}_t - \nabla f(x_\star)\|^2\right]. \quad (37)$$

To bound the last term in (37), we can apply Assumption 4:

$$\mathbb{E}\left[\|\hat{g}_t - \nabla f(x_\star)\|^2\right] \leq 2AD_f(x_t, x_\star) + B\sigma_t + C. \quad (38)$$

Plugging (38) into (37) gives us

$$
\begin{aligned}
\mathbb{E}\left[\|\hat{w}_t - w_\star\|^2\right] &\leq (1 - \gamma\mu)\|x_t - x_\star\|^2 - 2\gamma D_f(x_t, x_\star) + \gamma^2\left(2AD_f(x_t, x_\star) + B\sigma_t + C\right) \\
&\leq (1 - \gamma\mu)\|x_t - x_\star\|^2 - 2\gamma D_f(x_t, x_\star)(1 - \gamma A) + \gamma^2 B\sigma_t + \gamma^2 C, \quad (39)
\end{aligned}
$$

which is what we set out to prove. $\square$

### C.3 Proof of Theorem 5

*Proof.* Using definition of the Lyapunov function $\Psi_t$, and the tower property of conditional expectation, we obtain

$$
\begin{aligned}
\mathbb{E}\left[\Psi_{t+1}\right] &= \mathbb{E}\left[\|x_{t+1} - x_\star\|^2 + \frac{\gamma^2}{p^2}\|h_{t+1} - h_\star\|^2 + W\gamma^2\sigma_{t+1}\right] \\
&\overset{(27)}{\leq} \mathbb{E}\left[\|\hat{w}_t - w_\star\|^2\right] + (1-p^2)\frac{\gamma^2}{p^2}\|h_t - h_\star\|^2 \\
&\quad + W\gamma^2\left(2\tilde{A}D_f(x_t, x_\star) + \tilde{B}\sigma_t + \tilde{C}\right) \\
&\overset{(39)}{\leq} (1-\gamma\mu)\|x_t - x_\star\|^2 - 2\gamma D_f(x_t, x_\star)(1-\gamma A) + \gamma^2 B\sigma_t + \gamma^2 C \\
&\quad + (1-p^2)\frac{\gamma^2}{p^2}\|h_t - h_\star\|^2 + W\gamma^2\left(2\tilde{A}D_f(x_t, x_\star) + \tilde{B}\sigma_t + \tilde{C}\right) \\
&\leq (1-\gamma\mu)\|x_t - x_\star\|^2 - 2\gamma D_f(x_t, x_\star)\left(1 - \gamma(A + W\tilde{A})\right) \\
&\quad + \gamma^2 W\sigma_t\left(\frac{B + W\tilde{B}}{W}\right) + \gamma^2(C + W\tilde{C}) + (1-p^2)\frac{\gamma^2}{p^2}\|h_t - h_\star\|^2. \quad (40)
\end{aligned}
$$

Using the stepsize bound $\gamma \leq \frac{1}{A + W\tilde{A}}$, this leads to

$$
\mathbb{E}\left[\Psi_{t+1}\right] \leq (1-\gamma\mu)\|x_t - x_\star\|^2 + \gamma^2 W\sigma_t\left(\frac{B + W\tilde{B}}{W}\right) + \gamma^2(C + W\tilde{C}) + (1-p^2)\frac{\gamma^2}{p^2}\|h_t - h_\star\|^2.
$$

Let us denote $\beta := \frac{B + W\tilde{B}}{W}$. In order to obtain a contraction, we need to have $\beta < 1$, which is satisfied when $W > \frac{B}{1-\tilde{B}}$, and we get

$$
\begin{aligned}
\mathbb{E}\left[\Psi_{t+1}\right] &\leq (1-\gamma\mu)\|x_t - x_\star\|^2 + \gamma^2 W\sigma_t\beta + \gamma^2(C + W\tilde{C}) + (1-p^2)\frac{\gamma^2}{p^2}\|h_t - h_\star\|^2 \\
&\leq \max\left(1-p^2, \beta, 1-\gamma\mu\right)\Psi_t + \gamma^2(C + W\tilde{C}). \quad (41)
\end{aligned}
$$

Finally, using the tower property of expectation and unrolling recursion (41), we get

$$
\mathbb{E}\left[\Psi_T\right] \leq \max\left\{(1-\gamma\mu)^T, \beta^T, (1-p^2)^T\right\}\Psi_0 + \frac{\left(C + W\tilde{C}\right)\gamma^2}{\min\left\{\gamma\mu, p^2, 1-\beta\right\}}.
$$

$\square$

# D   Examples of Methods Without Variance Reduction

## D.1   Proof of Theorem 6 (GD estimator)

*Proof.* Let us show that GD estimator ($\hat{g}_t = \nabla f(x_t)$) satisfies Assumption 4

$$\mathbb{E}\left[\|\hat{g}_t - \nabla f(x_\star)\|^2\right] = \|\nabla f(x_t) - \nabla f(x_\star)\|^2 \overset{(19)}{\leq} 2LD_f(x_t, x_\star).$$

This means that Assumption 4 is satisfied with the following constant:

$$A = L, \quad B = 0, \quad C = 0, \quad \tilde{A} = 0, \quad \tilde{B} = 0, \quad \tilde{C} = 0, \quad \sigma_t \equiv 0.$$

Applying Theorem 5 leads to final recursion:

$$\mathbb{E}\left[\Psi_T\right] \leq \max\left\{(1-\gamma\mu)^T, (1-p^2)^T\right\}\Psi_0, \tag{42}$$

By inspecting (42) it is easy to see that

$$T \geq \max\left\{\frac{1}{\gamma\mu}, \frac{1}{p^2}\right\}\log\frac{1}{\varepsilon} \quad\implies\quad \mathbb{E}\left[\Psi_T\right] \leq \varepsilon\Psi_0. \tag{43}$$

Then the communication complexity is equal to

$$pT \geq \max\left\{\frac{p}{\gamma\mu}, \frac{1}{p}\right\}\log\frac{1}{\varepsilon}. \tag{44}$$

Setting $\gamma = \frac{1}{L}$ and solving $\frac{pL}{\mu} = \frac{1}{p}$ gives the optimal probability

$$p = \sqrt{\frac{\mu}{L}} = \frac{1}{\sqrt{\kappa}} \tag{45}$$

Finally, the iteration complexity and communication complexity have the following form:

$$T \geq \max\left\{\frac{1}{\gamma\mu}, \frac{1}{p^2}\right\}\log\frac{1}{\varepsilon} = \kappa\log\frac{1}{\varepsilon}, \tag{46}$$

$$pT \geq \max\left\{\frac{p}{\gamma\mu}, \frac{1}{p}\right\}\log\frac{1}{\varepsilon} = \sqrt{\kappa}\log\frac{1}{\varepsilon}. \tag{47}$$

$\square$

This recovers the result obtained by Mishchenko et al. [2022].

## D.2   Proof of Theorem 8 (SGD estimator)

*Proof.* Let us show that the SGD estimator $\hat{g}_t = g(x_t, \xi_t)$ satisfying Assumption 7 also satisfies Assumption 4. Using Young's inequality we get

$$
\begin{aligned}
\mathbb{E}\left[\|\hat{g}_t - \nabla f(x_\star)\|^2\right] &= \mathbb{E}\left[\|g(x_t, \xi_t) - \nabla f(x_\star)\|^2\right] \\
&= \mathbb{E}\left[\|g(x_t, \xi_t) - g(x_\star, \xi_t) + g(x_\star, \xi_t) - \nabla f(x_\star)\|^2\right] \\
&\overset{(23)}{\leq} 2\mathbb{E}\left[\|g(x_t, \xi_t) - g(x_\star, \xi_t)\|^2\right] + 2\mathbb{E}\left[g(x_\star, \xi_t) - \nabla f(x_\star)\|^2\right] \\
&\overset{(7)}{\leq} 4A''D_f(x_t, x_\star) + 2\mathrm{Var}(g(x_\star, \xi)).
\end{aligned}
\tag{48}
$$

This means that Assumption 4 is satisfied with the following constants:

$$A = 2A'', \quad B = 0, \quad C = 2\mathrm{Var}(g(x_\star, \xi)), \quad \tilde{A} = 0, \quad \tilde{B} = 0, \quad \tilde{C} = 0, \quad \sigma_t \equiv 0.$$

Applying Theorem 5 leads to the final bound:

$$\mathbb{E}\left[\Psi_T\right] \leq \max\left\{(1-\gamma\mu)^T, (1-p^2)^T\right\}\Psi_0 + \gamma^2\frac{2\mathrm{Var}(g(x_\star, \xi))}{\min\{\gamma\mu, p^2\}}. \tag{49}$$

In order to minimize the number of prox evaluations, whatever the choice of $\gamma$ will be, we choose the smallest probability $p$ which does not lead to any degradation of the rate $\min\{\gamma\mu, p^2\}$. That is, we choose $p = \sqrt{\gamma\mu}$. The first term on the right-hand side of (49) can be bounded as follows:

$$T \geq \frac{1}{\gamma\mu} \log\left(\frac{2\Psi_0}{\varepsilon}\right) \implies (1 - \gamma\mu)^T \Psi_0 \leq \frac{\varepsilon}{2}.$$

The second term on the right-hand side of (49) can be bounded as follows:

$$\gamma \leq \frac{\varepsilon\mu}{2C} \implies \frac{\gamma C}{\mu} \leq \frac{\varepsilon}{2}.$$

We choose the largest stepsize consistent with bounds $\gamma \leq \frac{\varepsilon\mu}{2C}$ and $\gamma \leq \frac{1}{A}$:

$$\gamma = \min\left\{\frac{1}{A}, \frac{\varepsilon\mu}{2C}\right\}.$$

Using this stepsizem we get the following iteration and (expected) communication complexities:

$$T \geq \max\left\{\frac{A}{\mu}, \frac{2C}{\varepsilon\mu^2}\right\} \log\left(\frac{2\Psi_0}{\varepsilon}\right), \qquad pT \geq \max\left\{\sqrt{\frac{A}{\mu}}, \sqrt{\frac{2C}{\varepsilon\mu^2}}\right\} \log\left(\frac{2\Psi_0}{\varepsilon}\right).$$

This recovers the result obtained by Mishchenko et al. [2022]. $\qquad\square$

# E  Analysis of ProxSkip-HUB

In this section we provide analysis of the new algorithm ProxSkip-HUB, which works for the new FL formulation described in Section 4. The pseudocode is presented in Algorithm 2.

## E.1  Lemma for minibatch sampling

Fix a minibatch size $\tau \in \{1, 2, \ldots, n\}$ and let $\mathcal{S}_t$ be a random subset of $\{1, 2, \ldots, n\}$ of size $\tau$, chosen uniformly at random. Define the gradient estimator via

$$g(x) := \frac{1}{\tau} \sum_{j \in \mathcal{S}_t} \nabla \widetilde{\phi}_j(x) \tag{50}$$

**Lemma 3.** *The gradient estimator $g(x)$ defined in (50) is unbiased. If we further assume that $n \geq 2$, $\widetilde{\phi}_j$ is convex and $L_j$-smooth for all $j$, and $f$ is $L$-smooth, then*

$$\mathbb{E}\left[\|g(x_t) - g(x_\star)\|^2\right] \leq 2L(\tau) D_f(x_t, x_\star),$$

*where*

$$L(\tau) := \frac{n - \tau}{\tau(n-1)} \max_j L_j + \frac{n(\tau - 1)}{\tau(n-1)} L.$$

*Proof.* Let $\chi_j$ be the random variable defined by

$$\chi_j = \begin{cases} 1 & j \in S \\ 0 & j \notin S \end{cases}.$$

It is easy to show that

$$\mathbb{E}\left[\chi_j\right] = \mathrm{Prob}(j \in S) = \frac{\tau}{n}. \tag{51}$$

Unbiasedness of $g(x)$ now follows via direct computation:

$$\mathbb{E}\left[g(x)\right] \overset{(50)}{=} \mathbb{E}\left[\frac{1}{\tau} \sum_{j \in S} \nabla \widetilde{\phi}_j(x)\right] = \mathbb{E}\left[\frac{1}{\tau} \sum_{j=1}^{n} \chi_i \nabla \widetilde{\phi}_j(x)\right] = \frac{1}{\tau} \sum_{j=1}^{n} \mathbb{E}\left[\chi_i\right] \nabla \widetilde{\phi}_j(x)$$

$$= \frac{1}{\tau} \sum_{j=1}^{n} \mathrm{Prob}(j \in S) \nabla \widetilde{\phi}_j(x) \overset{(51)}{=} \frac{1}{\tau} \sum_{j=1}^{n} \frac{\tau}{n} \nabla \widetilde{\phi}_j(x) = \nabla f(x).$$

Let us define

$$a_j := \nabla \widetilde{\phi}_j(x) - \nabla \widetilde{\phi}_j(x_\star). \tag{52}$$

Let $\chi_{j,k}$ be the random variable defined by

$$\chi_{j,k} = \begin{cases} 1 & j \in \mathcal{S}_t \text{ and } k \in \mathcal{S}_t \\ 0 & \text{otherwise} \end{cases}.$$

Note that

$$\chi_{j,k} = \chi_j \chi_k. \tag{53}$$

Further, it is easy to show that

$$\mathbb{E}\left[\chi_{j,k}\right] = \mathrm{Prob}(j \in \mathcal{S}_t, k \in \mathcal{S}_t) = \frac{\tau(\tau - 1)}{n(n-1)}. \tag{54}$$

Let us consider

$$
\begin{aligned}
\mathbb{E}\left[\|g(x_t) - g(x_\star)\|^2\right] &= \mathbb{E}\left[\left\|\frac{1}{\tau}\sum_{j\in\mathcal{S}_t}\nabla\widetilde{\phi}_j(x) - \frac{1}{\tau}\sum_{j\in\mathcal{S}_t}\nabla\widetilde{\phi}_j(x_\star)\right\|^2\right] \\
&= \mathbb{E}\left[\left\|\frac{1}{\tau}\sum_{j\in\mathcal{S}_t}\left(\nabla\widetilde{\phi}_j(x) - \nabla\widetilde{\phi}_j(x_\star)\right)\right\|^2\right] \\
&\overset{(52)}{=} \mathbb{E}\left[\left\|\frac{1}{\tau}\sum_{j\in\mathcal{S}_t}a_j\right\|^2\right] \\
&= \frac{1}{\tau^2}\mathbb{E}\left[\left\|\sum_{j=1}^n\chi_j a_j\right\|^2\right] \\
&= \frac{1}{\tau^2}\mathbb{E}\left[\sum_{j=1}^n\|\chi_j a_j\|^2 + \sum_{k\neq j}\langle\chi_j a_j, \chi_k a_k\rangle\right] \\
&\overset{(53)}{=} \frac{1}{\tau^2}\mathbb{E}\left[\sum_{j=1}^n\|\chi_j a_j\|^2 + \sum_{k\neq j}\chi_{j,k}\langle a_j, a_k\rangle\right]. \quad (55)
\end{aligned}
$$

Using the formulas (51) and (54) we can continue:

$$
\begin{aligned}
\mathbb{E}\left[\|g(x_t) - g(x_\star)\|^2\right] &= \frac{1}{\tau^2}\left(\frac{\tau}{n}\sum_{j=1}^n\|a_j\|^2 + \frac{\tau(\tau-1)}{n(n-1)}\sum_{j\neq k}\langle a_j, a_k\rangle\right) \\
&= \frac{1}{\tau n}\sum_{j=1}^n\|a_j\|^2 + \frac{\tau-1}{\tau n(n-1)}\sum_{j\neq k}\langle a_j, a_k\rangle \\
&= \frac{1}{\tau n}\sum_{j=1}^n\|a_j\|^2 + \frac{\tau-1}{\tau n(n-1)}\left(\left\|\sum_{j=1}^n a_j\right\|^2 - \sum_{j=1}^n\|a_j\|^2\right) \\
&= \frac{n-\tau}{\tau(n-1)}\frac{1}{n}\sum_{j=1}^n\|a_j\|^2 + \frac{n(\tau-1)}{\tau(n-1)}\left\|\frac{1}{n}\sum_{j=1}^n a_j\right\|^2. \quad (56)
\end{aligned}
$$

Since $\widetilde{\phi}_j$ is convex and $L_j$-smooth, we know that

$$
\|a_j\|^2 = \|\nabla f_j(x_t) - \nabla f_j(x_\star)\|^2 \leq 2L_j D_{\widetilde{\phi}_j}(x_t, x_\star). \quad (57)
$$

Since $f$ is convex and $L$-smooth, we know that

$$
\left\|\frac{1}{n}\sum_{i=1}^n a_i\right\|^2 = \|\nabla f(x_t) - \nabla f(x_\star)\|^2 \leq 2L D_f(x_t, x_\star). \quad (58)
$$

Let us apply the bound $L_j \leq \max_j L_j$ and use the identity $D_f(x_t, x_\star) = \frac{1}{n}\sum_{j=1}^{n} D_{\widetilde{\phi}_j}(x_t, x_\star)$ :

$$
\mathbb{E}\left[\left\|\frac{1}{\tau}\sum_{j\in\mathcal{S}_t}\nabla\widetilde{\phi}_j(x) - \frac{1}{\tau}\sum_{j\in\mathcal{S}_t}\nabla\widetilde{\phi}_j(x_\star)\right\|^2\right] \leq \frac{n-\tau}{\tau(n-1)}\frac{1}{n}\sum_{j=1}^{n} 2L_j D_{\widetilde{\phi}_j}(x_t, x_\star)
$$

$$
+ \frac{n(\tau-1)}{\tau(n-1)} 2LD_f(x_t, x_\star)
$$

$$
\leq 2\frac{n-\tau}{\tau(n-1)}\max_j L_j \frac{1}{n}\sum_{j=1}^{n} D_{f_j}(x_t, x_\star)
$$

$$
+ 2\frac{n(\tau-1)}{\tau(n-1)}LD_f(x, y)
$$

$$
= 2\frac{n-\tau}{\tau(n-1)}\max_j L_j D_f(x, y)
$$

$$
+ 2\frac{n(\tau-1)}{\tau(n-1)}LD_f(x_t, x_\star)
$$

$$
= 2\left(\frac{n-\tau}{\tau(n-1)}\max_j L_j + \frac{n(\tau-1)}{\tau(n-1)}L\right)D_f(x_t, x_\star).
$$

$\square$

## E.2  Proof of Theorem 9

As in previous analysis we need to show that Assumption 4 is satisfied for the ProxSkip-HUB method.

*Proof.* Let us consider the first inequality in Assumption 4 and show that it holds for the new gradient estimator $\hat{g}_t = \frac{1}{\tau}\sum_{j\in\mathcal{S}_t} Q(\nabla\widetilde{\phi}_j(x_t) - \nabla\widetilde{\phi}_j(y_t)) + \nabla f(y)$:

$$
\mathbb{E}\left[\|\hat{g}_t - \nabla f(x_\star)\|^2\right] = \mathbb{E}\left[\left\|\frac{1}{\tau}\sum_{j\in\mathcal{S}_t} Q(\nabla\widetilde{\phi}_j(x_t) - \nabla\widetilde{\phi}_j(y_t)) + \nabla f(y_t) - \nabla f(x_\star)\right\|^2\right]. \quad (59)
$$

Let $\Delta_t = \frac{1}{\tau}\sum_{j\in\mathcal{S}_t}(\nabla\widetilde{\phi}_j(x_t) - \nabla\widetilde{\phi}_j(y_t))$ and $\hat{\Delta}_t = \frac{1}{\tau}\sum_{j\in\mathcal{S}_t} Q(\nabla\widetilde{\phi}_j(x_t) - \nabla\widetilde{\phi}_j(y_t))$. Using smart zero $0 = \Delta_t - \Delta_t$ we have

$$
\mathbb{E}\left[\|\hat{g}_t - \nabla f(x_\star)\|^2\right] = \mathbb{E}\left[\left\|\hat{\Delta}_t - \Delta_t + \Delta_t + \nabla f(y) - \nabla f(x_\star)\right\|^2\right]
$$

$$
\overset{(23)}{\leq} 2\mathbb{E}\left[\|\hat{\Delta}_t - \Delta_t\|^2\right] + 2\mathbb{E}\left[\|\Delta_t + \nabla f(y_t) - \nabla f(x_\star)\|^2\right]
$$

$$
\leq 2\mathbb{E}\left[\left\|\frac{1}{\tau}\sum_{j\in\mathcal{S}_t} Q(\nabla\widetilde{\phi}_j(x_t) - \nabla\widetilde{\phi}_j(y_t)) - \frac{1}{\tau}\sum_{j\in\mathcal{S}_t}(\nabla\widetilde{\phi}_j(x_t) - \nabla\widetilde{\phi}_j(y_t))\right\|^2\right]
$$

$$
+ 2\mathbb{E}\left[\left\|\frac{1}{\tau}\sum_{j\in\mathcal{S}_t}(\nabla\widetilde{\phi}_j(x_t) - \nabla\widetilde{\phi}_j(y_t)) + \nabla f(y) - \nabla f(x_\star)\right\|^2\right]. \quad (60)
$$

Let us consider the first term (60), let us define $\theta_t^i = Q(\nabla\widetilde{\phi}_j(x_t) - \nabla\widetilde{\phi}_j(y_t)) - (\nabla\widetilde{\phi}_j(x_t) - \nabla\widetilde{\phi}_j(y_t))$:

$$
\begin{aligned}
\mathbb{E}\left[\|\hat{\Delta}_t - \Delta_t\|^2\right] &= \mathbb{E}\left[\left\|\frac{1}{\tau}\sum_{j\in\mathcal{S}_t} Q(\nabla\widetilde{\phi}_j(x_t) - \nabla\widetilde{\phi}_j(y_t)) - (\nabla\widetilde{\phi}_j(x_t) - \nabla\widetilde{\phi}_j(y_t))\right\|^2\right] \\
&= \mathbb{E}\left[\left\|\frac{1}{\tau}\sum_{j\in\mathcal{S}_t}\theta_t^i\right\|^2\right] \\
&= \mathbb{E}\left[\frac{1}{\tau^2}\left(\sum_{j\in\mathcal{S}_t}\|\theta_t^i\|^2 + \sum_{i\neq j}\left\langle\theta_t^i,\theta_t^j\right\rangle\right)\right] \\
&= \frac{1}{\tau^2}\left(\sum_{j\in\mathcal{S}_t}\mathbb{E}\left[\|\theta_t^i\|^2\right] + \sum_{i\neq j}\mathbb{E}\left[\left\langle\theta_t^i,\theta_t^j\right\rangle\right]\right).
\end{aligned}
$$

Using independence and unbiasedness of compressors we have

$$
\begin{aligned}
\mathbb{E}\left[\|\hat{\Delta}_t - \Delta_t\|^2\right] &= \frac{1}{\tau^2}\left(\sum_{j\in\mathcal{S}_t}\mathbb{E}\left[\|\theta_t^i\|^2\right] + \sum_{i\neq j}\mathbb{E}\left[\left\langle\theta_t^i,\theta_t^j\right\rangle\right]\right) \\
&= \frac{1}{\tau^2}\left(\sum_{j\in\mathcal{S}_t}\mathbb{E}\left[\|\theta_t^i\|^2\right] + \sum_{i\neq j}\left\langle\mathbb{E}\left[\theta_t^i\right],\mathbb{E}\left[\theta_t^j\right]\right\rangle\right) \\
&= \frac{1}{\tau^2}\sum_{j\in\mathcal{S}_t}\mathbb{E}\left[\|\theta_t^i\|^2\right] \\
&= \frac{1}{\tau^2}\sum_{j\in\mathcal{S}_t}\mathbb{E}\left[\|Q(\nabla\widetilde{\phi}_j(x_t) - \nabla\widetilde{\phi}_j(y_t)) - (\nabla\widetilde{\phi}_j(x_t) - \nabla\widetilde{\phi}_j(y_t))\|^2\right] \\
&\overset{(12)}{\leq} \frac{\omega}{\tau}\mathbb{E}\left[\frac{1}{\tau}\sum_{j\in\mathcal{S}_t}\|\nabla\widetilde{\phi}_j(x_t) - \nabla\widetilde{\phi}_j(y_t)\|^2\right]. 
\end{aligned}
\tag{61}
$$

Using Young's inequality and expectation of client sampling we get

$$
\begin{aligned}
\mathbb{E}\left[\|\hat{\Delta}_t - \Delta_t\|^2\right] &\overset{(23)}{\leq} \frac{2\omega}{\tau}\mathbb{E}\left[\frac{1}{\tau}\sum_{j\in\mathcal{S}_t}\|\nabla\widetilde{\phi}_j(x_t) - \nabla\widetilde{\phi}_j(x_\star)\|^2\right] \\
&\quad + \frac{2\omega}{\tau}\mathbb{E}\left[\frac{1}{\tau}\sum_{j\in\mathcal{S}_t}\|\nabla\widetilde{\phi}_j(y_t) - \nabla\widetilde{\phi}_j(x_\star)\|^2\right] \\
&\overset{(23)}{\leq} \frac{2\omega}{\tau}\frac{1}{n}\sum_{j=1}^n\|\nabla\widetilde{\phi}_j(x_t) - \nabla\widetilde{\phi}_j(x_\star)\|^2 \\
&\quad + \frac{2\omega}{\tau}\frac{1}{n}\sum_{j=1}^n\|\nabla\widetilde{\phi}_j(y_t) - \nabla\widetilde{\phi}_j(x_\star)\|^2 \\
&\overset{(20)}{\leq} \frac{4\omega}{\tau}L_{\max}D_f(x_t,x_\star) + \frac{2\omega}{\tau}\frac{1}{n}\sum_{j=1}^n\|\nabla\widetilde{\phi}_j(y_t) - \nabla\widetilde{\phi}_j(x_\star)\|^2. 
\end{aligned}
\tag{62}
$$

Let us consider the second term in (60):

$$
\mathbb{E}\left[\left\|\frac{1}{\tau}\sum_{j\in\mathcal{S}_t}(\nabla\widetilde{\phi}_j(x_t)-\nabla\widetilde{\phi}_j(y_t))+\nabla f(y)-\nabla f(x_\star)\right\|^2\right]
$$

$$
=\mathbb{E}\left[\left\|\frac{1}{\tau}\sum_{j\in\mathcal{S}_t}(\nabla\widetilde{\phi}_j(x_t)-\nabla\widetilde{\phi}_j(y_t)+\nabla\widetilde{\phi}_j(x_\star)-\nabla\widetilde{\phi}_j(x_\star))+\nabla f(y)-\nabla f(x_\star)\right\|^2\right]
$$

$$
\overset{(23)}{\leq}2\mathbb{E}\left[\left\|\frac{1}{\tau}\sum_{j\in\mathcal{S}_t}\nabla\widetilde{\phi}_j(x_t)-\frac{1}{\tau}\sum_{j\in\mathcal{S}_t}\nabla\widetilde{\phi}_j(x_\star)\right\|^2\right]
$$

$$
+2\mathbb{E}\left[\left\|\frac{1}{\tau}\sum_{j\in\mathcal{S}_t}\nabla\widetilde{\phi}_j(x_\star)-\frac{1}{\tau}\sum_{j\in\mathcal{S}_t}\nabla\widetilde{\phi}_j(y_t)-\left(\nabla f(x_\star)-\frac{1}{\tau}\sum_{j\in\mathcal{S}_t}\nabla\widetilde{\phi}_j(y_t)\right)\right\|^2\right]
$$

$$
\leq2\mathbb{E}\left[\left\|\frac{1}{\tau}\sum_{j\in\mathcal{S}_t}\nabla\widetilde{\phi}_j(x_t)-\frac{1}{\tau}\sum_{j\in\mathcal{S}_t}\nabla\widetilde{\phi}_j(x_\star)\right\|^2\right]+2\mathbb{E}\left[\left\|\frac{1}{\tau}\sum_{j\in\mathcal{S}_t}\nabla\widetilde{\phi}_j(y_t)-\frac{1}{\tau}\sum_{j\in\mathcal{S}_t}\nabla\widetilde{\phi}_j(x_\star)\right\|^2\right].
$$

$$(63)$$

Using Lemma 3 and Jensen's inequality (25) we have

$$
\mathbb{E}\left[\left\|\frac{1}{\tau}\sum_{j\in\mathcal{S}_t}(\nabla\widetilde{\phi}_j(x_t)-\nabla\widetilde{\phi}_j(y_t))+\nabla f(y)-\nabla f(x_\star)\right\|^2\right]
$$

$$
\leq2\mathbb{E}\left[\left\|\frac{1}{\tau}\sum_{j\in\mathcal{S}_t}\nabla\widetilde{\phi}_j(x_t)-\frac{1}{\tau}\sum_{j\in\mathcal{S}_t}\nabla\widetilde{\phi}_j(x_\star)\right\|^2\right]+2\mathbb{E}\left[\left\|\frac{1}{\tau}\sum_{j\in\mathcal{S}_t}\nabla\widetilde{\phi}_j(y_t)-\frac{1}{\tau}\sum_{j\in\mathcal{S}_t}\nabla\widetilde{\phi}_j(x_\star)\right\|^2\right]
$$

$$
\leq4L(\tau)D_f(x_t,x_\star)+\frac{2}{n}\sum_{j=1}^n\left\|\nabla\widetilde{\phi}_j(y_t)-\nabla\widetilde{\phi}_j(x_\star)\right\|^2. \tag{64}
$$

Combining two parts (62), (64) and plugging into (60) we get

$$
\mathbb{E}\left[\|\hat{g}_t-\nabla f(x_\star)\|^2\right]\leq2\mathbb{E}\left[\left\|\frac{1}{\tau}\sum_{j\in\mathcal{S}_t}Q(\nabla\widetilde{\phi}_j(x_t)-\nabla\widetilde{\phi}_j(y_t))-\frac{1}{\tau}\sum_{j\in\mathcal{S}_t}(\nabla\widetilde{\phi}_j(x_t)-\nabla\widetilde{\phi}_j(y_t))\right\|^2\right]
$$

$$
+2\mathbb{E}\left[\left\|\frac{1}{\tau}\sum_{j\in\mathcal{S}_t}(\nabla\widetilde{\phi}_j(x_t)-\nabla\widetilde{\phi}_j(y_t))+\nabla f(y)-\nabla f(x_\star)\right\|^2\right]
$$

$$
\leq8L(\tau)D_f(x_t,x_\star)+\frac{4}{n}\sum_{j=1}^n\left\|\nabla\widetilde{\phi}_j(y_t)-\nabla\widetilde{\phi}_j(x_\star)\right\|^2
$$

$$
+\frac{8\omega}{\tau}L_{\max}D_f(x_t,x_\star)+\frac{4\omega}{\tau}\frac{1}{n}\sum_{j=1}^n\|\nabla\widetilde{\phi}_j(y_t)-\nabla\widetilde{\phi}_j(x_\star)\|^2
$$

$$
\leq2\cdot4\left(L(\tau)+\frac{\omega}{\tau}L_{\max}\right)D_f(x_t,x_\star)+4\left(1+\frac{\omega}{\tau}\right)\sigma_t, \tag{65}
$$

where $\sigma_t=\frac{1}{n}\sum_{j=1}^n\|\nabla\widetilde{\phi}_j(y_t)-\nabla\widetilde{\phi}_j(x_\star)\|^2$. Let us consider update of control variable $y_t$:

$$
y_{t+1}=\begin{cases}x_t & \text{with probability} \quad q\\ y_t & \text{with probability} \quad 1-q\end{cases}. \tag{66}
$$

Let us show that second inequality in Assumption 4:

$$
\mathbb{E}\left[\sigma_{t+1}\right] = \mathbb{E}\left[\frac{1}{n}\sum_{j=1}^{n}\|\nabla\widetilde{\phi}_j(y_{t+1}) - \nabla\widetilde{\phi}_j(x_\star)\|^2\right]
$$

$$
= (1-q)\frac{1}{n}\sum_{j=1}^{n}\|\nabla\widetilde{\phi}_j(y_t) - \nabla\widetilde{\phi}_j(x_\star)\|^2 + q\frac{1}{n}\sum_{j=1}^{n}\|\nabla\widetilde{\phi}_j(x_t) - \nabla\widetilde{\phi}_j(x_\star)\|^2
$$

$$
= (1-q)\,\sigma_t + 2qL_{\max}D_f(x_t, x_\star). \tag{67}
$$

Using (65) and (72) bounds we can confirm that Assumption 4 is satisfied with the following constants:

$$
A = 4\left(L(\tau) + \frac{\omega}{\tau}L_{\max}\right), \quad B = 4\left(1 + \frac{\omega}{\tau}\right), \quad C = 0, \quad \tilde{A} = qL_{\max}, \quad \tilde{B} = 1 - q, \quad \tilde{C} = 0.
$$

Applying Theorem 5 leads to the final result

$$
\mathbb{E}\left[\Psi_T\right] \le \max\left\{(1-\gamma\mu)^T, (1-p^2)^T, (1-q/2)^T\right\}\Psi_0, \tag{68}
$$

where the Lyapunov function is defined by

$$
\Psi_t := \|x_t - x_\star\|^2 + \frac{\gamma^2}{p^2}\|h_t - h_\star\|^2 + \gamma^2\frac{8}{q}\left(1 + \frac{\omega}{\tau}\right)\sigma_t.
$$

Let us set $\gamma = \frac{1}{A + W\tilde{A}}$, $p = \sqrt{\gamma\mu}$ and $q = 2\gamma\mu$. Using the same proof as for ProxSkip in Section D.1 and $L(\tau) \le L_{\max}$ we get communication and iteration complexities:

$$
T_{\text{comm.}} = \mathcal{O}\left(\sqrt{\frac{L_{\max}}{\mu}\left(1 + \frac{\omega}{\tau}\right)}\log\frac{1}{\varepsilon}\right), \qquad T_{\text{iter.}} = \mathcal{O}\left(\frac{L_{\max}}{\mu}\left(1 + \frac{\omega}{\tau}\right)\log\frac{1}{\varepsilon}\right).
$$

$\square$

If we use full participation $\tau = n$ and $q = \frac{1}{\omega+1}$ and $r(x) \equiv 0$ then we get the same rate as for DIANA [Mishchenko et al., 2019, Horváth et al., 2019b] and RAND-DIANA [Shulgin and Richtárik, 2021].

# F    Analysis of ProxSkip-LSVRG

The analysis of ProxSkip-LSVRG is almost the same to the analysis of ProxSkip-HUB, with one exception. We use a different sigma component:

$$
\sigma_t = \mathbb{E}\left[\left\|\frac{1}{\tau}\sum_{j\in\mathcal{S}_t}(\nabla\widetilde{\phi}_j(y_t) - \nabla\widetilde{\phi}_j(x_\star))\right\|^2\right]. \tag{69}
$$

Let us consider $\mathbb{E}\left[\|\hat{g}_t - \nabla f(x_\star)\|^2\right]$:

$$
\mathbb{E}\left[\|\hat{g}_t - \nabla f(x_\star)\|^2\right] = \mathbb{E}\left[\left\|\frac{1}{\tau}\sum_{j\in S_t}\left(\nabla\widetilde{\phi}_j(x_t) - \nabla\widetilde{\phi}_j(y_t)\right) + \nabla f(y) - \nabla f(x_\star)\right\|^2\right] \tag{70}
$$

Using (63) we have

$$
\mathbb{E}\left[\|\hat{g}_t - \nabla f(x_\star)\|^2\right] \le 2\mathbb{E}\left[\left\|\frac{1}{\tau}\sum_{j\in\mathcal{S}_t}\nabla\widetilde{\phi}_j(x_t) - \frac{1}{\tau}\sum_{j\in\mathcal{S}_t}\nabla\widetilde{\phi}_j(x_\star)\right\|^2\right]
$$

$$
+ 2\mathbb{E}\left[\left\|\frac{1}{\tau}\sum_{j\in\mathcal{S}_t}\nabla\widetilde{\phi}_j(y_t) - \frac{1}{\tau}\sum_{j\in\mathcal{S}_t}\nabla\widetilde{\phi}_j(x_\star)\right\|^2\right]
$$

$$
\le 4L(\tau)D_f(x_t, x_\star) + 2\sigma_t. \tag{71}
$$

Let us show that second inequality in Assumption 4, using Lemma 3 we get

$$
\begin{aligned}
\mathbb{E}\left[\sigma_{t+1}\right] &= \mathbb{E}\left[\left\|\frac{1}{\tau}\sum_{j\in\mathcal{S}_t}(\nabla\widetilde{\phi}_j(y_{t+1})-\nabla\widetilde{\phi}_j(x_\star))\right\|^2\right] \\
&= (1-q)\,\mathbb{E}\left[\left\|\frac{1}{\tau}\sum_{j\in\mathcal{S}_t}(\nabla\widetilde{\phi}_j(y_t)-\nabla\widetilde{\phi}_j(x_\star))\right\|^2\right] + q\mathbb{E}\left[\left\|\frac{1}{\tau}\sum_{j\in\mathcal{S}_t}(\nabla\widetilde{\phi}_j(x_t)-\nabla\widetilde{\phi}_j(x_\star))\right\|^2\right] \\
&= (1-q)\,\sigma_t + 2qL(\tau)D_f(x_t,x_\star).
\end{aligned} \tag{72}
$$

We showed that Assumption 4 is satisfied with following constants:

$$
A = 2L(\tau), \quad B = 2, \quad C = 0, \quad \tilde{A} = qL(\tau), \quad \tilde{B} = 1-q, \quad \tilde{C} = 0. \tag{73}
$$

Applying Theorem 5 with $\gamma = \frac{1}{6L(\tau)}$ we get final bound:

$$
\mathbb{E}\left[\Psi_T\right] \le \max\left\{(1-\gamma\mu)^T, (1-p^2)^T, \left(1-\frac{q}{2}\right)^T\right\}\Psi_0,
$$

where the Lyapunov function is defined as

$$
\Psi_t := \|x_t - x_\star\|^2 + \frac{\gamma^2}{p^2}\|h_t - h_\star\|^2 + \gamma^2\frac{4}{q}\sigma_t.
$$

Using the same argument as for ProxSkip and setting $\frac{q}{2} = \gamma\mu$, we get

$$
T_{\text{comms}} = \mathcal{O}\left(\sqrt{\frac{L(\tau)}{\mu}}\log\frac{1}{\varepsilon}\right), \qquad T_{\text{iters}} = \mathcal{O}\left(\frac{L(\tau)}{\mu}\log\frac{1}{\varepsilon}\right). \tag{74}
$$

If $r(x) \equiv 0$, this recovers results of Kovalev et al. [2020a].