# OpenReview forum: "Variance Reduced ProxSkip: Algorithm, Theory and Application to Federated Learning"
_NeurIPS.cc/2022/Conference — NeurIPS 2022 Accept_

### Official Review · Reviewer_VKkX · 2022-07-10

**Rating:** 6
**Confidence:** 5
**Soundness:** 3 good
**Presentation:** 3 good
**Contribution:** 2 fair

**Summary:**

This paper investigates applications of the variance-reduced technique in Federated Learning and proposed a new algorithm based on **ProxSkip** in [Mishchenko et al. 2022]. The method constructed by the authors can be substantially faster in terms of the *total training time* than ProxSkip in theory and practice in the regime when local computation is sufficiently expensive. The authors also corroborate their theoretical results through experiments.

**Questions:**

1. Table 1 says the communication complexity of **FedLin** is worse than GD. But the communication complexity of FedLin is also reduced by a factor of $H$ compared with GD, where $H$ is the number of local updates (see the discussion below Eq. (9) of [Mitra et al., 2021]). Therefore, the communication complexity of FedLin can also attain $O(\sqrt{L/\mu}\log(1/\epsilon))$ by setting $H = \sqrt{L/\mu}$.

2. In [Mitra et al., 2021], the authors show that FedLin converges much faster than SCAFFOLD in experiments of the linear model. However, the performance of FedLin in Section 5 is worse than SCAFFOLD and Local SGD, which is unreasonable since FedLin has a *linear convergence rate*. And I don't think the difference between logistic regression and linear regression is the reason. So I doubt the results in Section 5, could you explain the reason?

**Limitations:**

This paper identifies 5 distinct generations of LT methods and classifies ProxSkip and ProxSkip-VR in Generation 5: Accelerated Age, but I don't think the full-gradient and variance-reduced methods are practical and popular in FL. Considering large-scale machine learning models, stochastic gradient descent (and its variants) with the *constant batch size* is a prevalent training scheme. So the summarization in this paper may be less appropriate.

Considering existing works [Mitra et al., 2021] and [Mishchenko et al. 2022], the contribution and novelty are limited. More importantly, the experiment results are not convincing.

**Strengths And Weaknesses:**

**Strengths**
1. The proposed method **ProxSkip-VR** enjoys a lower communication complexity than gradient descent.

2. This paper proposed a new FL architecture, where ProxSkip-VR can be substantially faster in terms of the *total training time* than ProxSkip.

3. The authors corroborate their theoretical results with carefully engineered proof-of-concept experiments.

**Weaknesses**

1. The algorithm can be impractical since we can not control the communication timestamp.

2.  Attributing to combine L-SVRG [Hofmann et al., 2015, Kovalev et al., 2020a] and DIANA [Mishchenko et al., 2019] techniques, the improvement over ProxSkip is natural. Hence the novelty is limitted.

---

> ### Author Response · Authors · 2022-07-29
> **On strengths**
>
> **On Strengths:** Thanks for saying that
>
> ---
>
> - "The proposed method ProxSkip-VR enjoys a lower communication complexity than gradient descent."
> - "This paper proposed a new FL architecture, where ProxSkip-VR can be substantially faster in terms of the total training time than ProxSkip."
> - "The authors corroborate their theoretical results with carefully engineered proof-of-concept experiments."
>
> ---
>
> - We wish to highlight that the fact that ProxSkip-VR has a (significantly) better communication complexity than GD makes our paper only the 2nd paper (after ProxSkip) which belongs to the 5th generation of local training methods, despite thousands of papers having been written on this topic on the last decade. Moreover, our results subsume those of Mishchenko et al as special cases (and suboptimal special cases in terms of total communication complexity!)
>
> - No comment was made on the innovation provided by our HUB model.
>
> - We also wish to stress that ProxSkip-LSVRG (which we tested in Section 5) applies to the standard FL formulation (1) as well. Moreover, ProxSkip-LSVRG beats ProxSkip in terms of the total complexity, which includes the communication and local computation complexity. This means that ProxSkip-LSVRG is currently the theoretical SOTA in terms of total complexity among all known LT methods. We believe this alone its very important for the FL community. Moreover, no prior work on LT methods included local computation complexity in their comparison because the bottleneck of all LT methods belonging to generation 4 or earlier generations had worse communication complexity that vanilla gradient descent. Including the local work would make the methods look even worse in theory. It is only now (in the 5th generation) that we can finally consider total complexity.
>
> - Yes, we agree our experiments are carefully engineered proof of concept experiments. We test several theoretical predictions of our theory, and successfully confirm that the predictions translate to genuine practical gains.

---

> ### Author Response · Authors · 2022-07-29
> **Weakness 1: communication timestamp**
>
> ---
>
> The algorithm can be impractical since we can not control the communication timestamp.
>
> ---
>
> - **This claim is both unsubstantiated (i.e., no evidence supporting this claim was provided) and more importantly, it is incorrect.**
>
> - **Moreover, the reviewer does not explain what he/she means by "communication timestamp"** (we never use this expression in the paper either, so it is not possible to be sure what the reviewer means). **Due to this situation, all we can do is to take an educated guess about what the reviewer wanted to say. If we are not guessing right, please explain in detail what you meant and we will be happy to respond.** Our interpretation of this criticism is: "ProxSkip-VR does not offer any mechanism for deciding how many local training steps are performed before communication takes place". We will respond to this interpretation below.
>
> - **This claim, as interpreted above, is incorrect.* Our method, just as the original ProxSkip method, involves the hyper-parameter $p$ which is equal to the probability of evaluating the proximity operator of $r$. For the specific regularizer $r$ defined in Eq (4) in the paper, evaluation of the proximity operator means communication by averaging (see Mishchenko et al, 2022). So, parameter $p$ indirectly controls communication frequency. Note that since in each iteration we communicate with probability $p$ (and otherwise take a local training step), this means the expected number of local training steps before communication takes place is equal to $1/p$. While virtually all standard LT methods in the FL literature rely on taking $K$ (a deterministic quantity) local training steps, our method takes a random number of local training steps, equal to $1/p$ in expectation. So, our method can control communication frequency by setting $1/p$ analogously to how standard FL methods control communication frequency by setting by setting $K$.
>
> We would very much appreciate a response. We believe the criticism is incorrect, and we would like you to either acknowledge that, or explain in detail what you meant if our guess was not right, and explain in detail why this is a problem. We will be most happy to reply.

---

> > ### Author Response · Authors · 2022-08-02
> > **New variant of ProxSkip-VR, and a new experiment**
> >
> > Dear reviewer,
> >
> > In order to probe the possible meaning of your question more deeply, we have performed a few experiments with a new variant of ProxSkip-VR, which we call ProxSkio-VR-D. This method takes $1/p$ local steps *deterministically*, whereas ProxSkip-VR takes $1/p$ local steps *in expectation*. We do not have an analysis for this variant; this is an open problem. Randomness is a key ingredient in our analysis (and in the analysis of the original ProxSkip algorithm) which makes it possible.
> >
> > We used the LSVRG estimator for our methods.
> >
> > Since we can not upload plots here, we have at least tried to convey the essential information they uncover by summarizing the results in a table; these are the same as those in our original submission, top row of Table 2, but we added the run of ProxSkio-VR-D:
> >
> > **batch size $\tau=16$:**
> >
> > | Comm. rounds  | 1000     | 3000     | 5000      | 7000       | 9000       |
> >
> > | ProxSkip-VR     | 7.53e-4 | 1.88e-6 | 3.92e-8 | 1.67e-10 | 3.19e-11 |
> >
> > | ProxSkip-VR-D | 6.31e-4 | 1.51e-6 | 1.09e-8 | 1.31e-10 | 3.17e-11 |
> >
> > **batch size $\tau=32$:**
> >
> > | Comm. rounds  | 1000     | 3000     | 5000        | 7000       | 9000       |
> >
> > | ProxSkip-VR     | 2.55e-4 | 8.03e-8 | 4.11e-10 | 2.96e-11 | 2.96e-11 |
> >
> > | ProxSkip-VR-D | 1.15e-4 | 2.50e-8 | 5.28e-11 | 2.97e-11 | 2.96e-11 |
> >
> > **batch size $\tau=64$:**
> >
> >  | Comm. rounds  | 1000     | 3000       | 5000        | 7000       | 9000       |
> >
> >  | ProxSkip-VR     | 1.77e-4 | 1.78e-5   | 4.70e-9   | 7.12e-11 | 2.98e-11 |
> >
> >  | ProxSkip-VR-D | 3.35e-5 | 4.01e-10 | 2.98e-11 | 2.96e-11 | 2.96e-11 |
> >
> > The numbers in the tables show the value of $f(x^t)-f(x^\star)$, i.e., the function suboptimality, after a certain number of communication rounds, as indicated by the number in the column. For each experiment, we obtained a tight estimate of the optimal value $f(x^{\star})$ by Nesterov's accelerated gradient descent method for a long time. We used the same stepsize for ProxSkip-VR-D as for ProxSkip-VR; i.e., we did not try to fine-tune the results, which means that both methods will likely improve their performance with further fine-tuning.
> >
> > **Our key finding is that ProxSkip-VR-D is slightly better, and can be even quite significantly better in the initial phases of the iterative process for larger minibatch sizes $\tau$.** For example, in the third experiment with $\tau=64$, ProxSkip-VR-D achieves accuracy $\varepsilon=4.01e-10$ after 3000 communication rounds, whereas ProxSkip-VR only achieves the accuracy $\varepsilon=1.78e-5$. This early advantage disappears later on. After 9000 communication rounds, both methods in all three experiments achieve comparable accuracy.
> >
> > We hope this gives an additional insight into your question about communication timestamp, and at the same time this **strengthens our contributions with a new heuristic variant of our method that appears to work better.** We will conduct more extensive testing and include the plots in the camera-ready version of the paper.

---

> > ### Comment · Reviewer_VKkX · 2022-08-05
> > **About the communication timestamp**
> >
> > The communication timestamp refers to the synchronization timestamp in FL. Like Local SGD or Scaffold, the synchronization schedule is deterministic, and we can choose one before the training starts. In your paper, the expectation of the number of skipped communication indeed equals $1/p$. However, the synchronization step is a Bernoulli random variable, which means that the synchronization schedule is also random and may be less flexible than popular FL methods.
> >
> > Also, there is another question about random synchronization in ``ProxSkip-VR``. Let's consider a case where $\mu$ is very close to $L$, which is more simple from the perspective of optimization. According to the choice of *communication probability* $p = \sqrt{\mu/L}$, if $\mu/L$ is close to 1, the number of skipped communication $K$ will also be close to $1$ in ``ProxSkip-VR`` . And if $\mu/L = 1$, then $K$ will be exactly 1! In ``FedLin`` or ``Scaffold``, we can still choose arbitrary large $K$. Could you address this issue?

---

> > > ### Author Response · Authors · 2022-08-05
> > > **Re: "the synchronization schedule is also random and may be less flexible than popular FL methods"**
> > >
> > > Thank you very much for engaging with us!
> > >
> > > We do not quite follow the argument the reviewer is trying to use here. Let us explain. Our approach, which uses a *random* number of local training steps, is claimed to be *less flexible* when compared to existing approaches, such as FedAvg/FedLin, which use a *deterministic* number of local training steps. **However, no explanation was given to support this claim, and it was not explained what notion of flexibility the reviewer has in mind.** Please can you elaborate? In what sense can any of these approaches be described as less flexible than the other? They are clearly different, but flexibility does not seem to be the right word to describe the difference. We offer some more thoughts about this below:
> > >
> > > 1.  **If flexibility is interpreted as the ability to control the # of local steps using a single parameter, then both approaches are equally flexible:** From the perspective of controlling the # of local training steps, both approaches depend on a single parameter: the classical ones use parameter $K$ (= # of local training steps), and our approach (which we borrowed from the ProxSkip work of Mishchenko et al, 2022) uses parameter $p$ (= the probability with which local training stops at all clients, and communication happens). As you acknowledged above, $1/p$ has about the same meaning as $K$ in the classical approach. That is, our method can be seen as working with a random # of local steps, where this random # is equal to $1/p$ in expectation.
> > >
> > > 2. **In some sense, one approach is more "flexible" than the other, and vice versa:** It is true that in our approach, at least in theory, we are not able to set the # of local steps to a deterministic quantity. In this sense, our approach is less flexible than the classical approach. On the other hand, the classical approach, at least in theory, is unable to set the # of local steps to a random quantity. In this sense, our approach is more flexible than the classical approach.
> > >
> > > 3. **Both have an easy implementation, so both are "flexible":** Let us now interpret "flexibility" as "implementation simplicity" (we do not know what the reviewer mean, so we continue making some guesses). Please notice that, in practice, the random communication schedule of ProxSkip-VR can be implemented easily: a) The server generates the sequence of $K_r = 1/p_r$ for all communication rounds $r=1, 2, \dots$ a-priori, before the algorithm is run, and communicates the sequence with the workers either at the start, or just-in-time, each $K_r$ just before the local training belonging to communication round $r$ is commenced. Alternatively, b) the server sends the same seed to all clients, at the start of the iterative process, for the generation of the numbers $K_1,K_2, \dots$. This is all very easy to execute on, and is extremely cheap in terms of communication overhead. Both of these two implementations are very simple, and make the random choice of local steps virtually as easy to implement as is the case with the deterministic number of local steps $K$ in classical algorithms.
> > >
> > > 4. **ProxSkip-VR-D: a deterministic communication-timestamp variant of ProxSkio-VR** Please note that we have now experimented with a new variant of ProxSkip-VR, which we call ProxSkip-VR-D. We did so precisely because in your original review you challenged the fact that our method employs a random number of local training steps as opposed to a deterministic number of local steps. See our response entitled "New variant of ProxSkip-VR, and a new experiment". This new variant uses a deterministic number of local steps, and in this sense has the exact same "flexibility" as the classical methods. The change from $1/p$ steps in expectations to $1/p$ deterministic steps does not seem to have done any harm in our experiment, an din fact led to some improvements, especially in the early stages of the process, and for larger minibatch sizes (we experimented with the LSVRG estimator). We do not have a theory for ProxSkip-VR-D, however. We believe that this alone should settle the issue you are pointing, if indeed it is an issue (which we do not agree with).
> > >
> > > 5. **Communication complexity is more important than "flexibility":** A key point we wish to make is that the notion of flexibility (which was not defined by the reviewer; and all our guesses as to what it might mean show that in terms of flexibility, the two approaches seem to be about equivalent) is certainly less important than the notion of communication efficiency. Indeed, the key reason behind the introduction of the local training paradigm, employed in FedAvg and subsequently in hundreds or thousands of papers (including FedLin), was its impressive practical utility in terms of how it improved communication efficiency! However, prior to ProxSkip, it was not shown theoretically by any paper that local training leads to a reduced # of communication rounds. We build and improve upon ProxSkip in this work!

---

> > > ### Author Response · Authors · 2022-08-05
> > > **Re: "another question about random synchronization in ProxSkip-VR"**
> > >
> > > Sure, we are most happy to explain what happens here. It's very simple, and does not point to any issue in our work.
> > >
> > > First, a recap:
> > >
> > > - **Recap:** Please note that the communication complexity of Generation 4 LT methods, such as FedLin and Scaffold, is ${\cal O}( L/\mu \log 1/\epsilon)$. This is the same as the communication complexity of gradient descent, which only employs a single local step. On the other hand, the communication complexity of Generation 5 LT methods is much better; it is *accelerated*. For example, the communication complexity of ProxSkip is ${\cal O}( \sqrt{L/\mu} \log 1/\epsilon)$. See Table 2 for the complexity of other variants. For example, in the case of ProxSkip-LSVRG it is ${\cal O}( \sqrt{L_\tau/\mu} \log 1/\epsilon)$, where $L_\tau = \frac{m-\tau}{\tau(m-1)}\max_i L_i + \frac{m(\tau-1)}{\tau (m-1)} L$, where $m$ is the number of data points on each client, and $\tau \in \{1,2,\dots,m\}$ is the minibatch size. In the context of what the reviewer is asking here, it makes most sense to compare methods that use the same batch size. So, below we will compare FedLin/Scaffold with ProxSkip. A similar comparison can be done for stochastic versions of all methods, but the benefit of our approach will be even larger as we employ variance reduction.
> > >
> > > We now answer your question:
> > >
> > > -  **Problems with $L/\mu\approx 1$ are simple; local training is not needed then:** The key to answering your question is the simple observation that if the condition number $\kappa =L/\mu$ is small, stay close to $1$, the optimization problem we are trying to solve is very easy! Indeed, in this case already vanilla gradient descent solves the problem in $\approx {\cal O}(1 \log 1/\epsilon)$ communication rounds, and hence **local training steps are not needed to begin with!** Please recall that we do not use local training steps just because we want to. The entire purpose of local training is to improve communication complexity. There is nothing to improve in this case, and hence we do not have any need for local training steps! In other words, one local step is already enough! ProxSkip theory predicts this. Indeed, we explain in Theorem 6 that $p=1/\sqrt{\kappa}$ is optimal. If $\kappa \approx 1$, then $1/p\approx 1$, and ProxSkip takes just 1 local step in expectation.
> > >
> > > - **Flexibility of choosing $p$:** Please note that if we really *want* to, we can choose $p$ to be different from $1/\sqrt{\kappa}$ in the same way classical methods can *choose* the constant $K$ defining the number of local steps arbitrarily. While the choice $p=1/\sqrt{\kappa}$ is optimal, i.e., it minimizes the number of communication rounds, all choices of $p$ in the interval $(,1]$ lead to a convergent method. However, with arbitrary choice of $p$, the communication complexity of ProxSkip becomes ${\cal O}( (p \kappa + 1/p) \log 1/\epsilon )$ (this can be deduced from Theorem 6; and was already mentioned in Mishchenko et al, 2022). Note that this is better than GD, and that any Generation 4 LT method, including Scaffold and FedLin, for any choice of $p$ in the interval $(1/\kappa,1)$. So, not only does the ProxSkip approach (which we improve on in our paper) beat GD, FedLin and Scaffold for the optimal choice $p=1/\sqrt{\kappa}$, it improves it whenever $p$ is smaller than 1, and not too small, i.e., not smaller than $1/\kappa$. So, we offer a very broad range of parameters $p$ for which ProxSkip is better than previous methods, suggesting the robustness and increased flexibility of our approach.
> > >
> > > - **A new modification?** Having said that, it may perhaps be possible to modify our method so that it takes a larger number of local steps in expectation than $1/p=\sqrt{\kappa}$, while still retaining the ${\cal O}(\sqrt{\kappa} \log 1/\epsilon)$ communication complexity. [ It is not clear what utility such a change may serve; we merely try too follow the thought process suggested/implied by the reviewer who perhaps believes that there might be some utility from more local steps even if they are no needed to reduce the number of communication rounds. ] Indeed, perhaps we can do what Generation 4 method do: use the stepsize $1/(L K)$ instead of $1/L$, and then choose $p/K$ instead of $p$, where $K> 1$. This means reducing the stepsize by the factor $K$, and at the same increasing the number of local steps (in expectation) from $1/p$ to $K/p$. Perhaps, and we are merely speculating here, such a method might retain the accelerated communication complexity ${\cal O}( \sqrt{L/\mu} \log 1/\epsilon)$ of ProxSkip. But as we said above, it is not clear at all why this would be preferable to taking fewer local steps.

---

> ### Author Response · Authors · 2022-07-29
> **Weakness 2: the improvement is natural**
>
> ---
>
> Attributing to combine L-SVRG [Hofmann et al., 2015, Kovalev et al., 2020a] and DIANA [Mishchenko et al., 2019] techniques, the improvement over ProxSkip is natural. Hence the novelty is limited.
>
> ---
>
> - Please note that we did not claim any "non-natural" or "super-natural" improvements. In hindsight, yes, the improvements may look "natural", especially since we believe we explained where they come from well. Most advances look natural in hindsight. This does not mean they were easy to obtain.
>
> - Note that our method (in particular, ProxSkip-LSVRG, which we consider in Sections 4 and 5) *is* the current theoretical SOTA among all local training methods developed in the last decade in terms of the total complexity, which includes communication complexity and local computation complexity. How can an advance like this be judged as borderline reject?
>
> - We have several novelties here:
>   - i) Our paper is the only the 2nd paper that belongs to the 5th generation of local training (LT) methods. This immediately means our approach is very novel when compared to almost all previous work on LT methods (with the sole exception of the original ProxSkip method).
>   - ii) Our approach generalized that of ProxSkip - we recover their methods ProxSkip-GD and ProxSkip-SGD as special cases (see Table 2). We generalize to any gradient estimator satisfying Assumption 4 - and there are tens of such estimators in the literature by now (The work of Gorbunov alone lists 15). DIANA and LSVRG are only two specific variance reduced estimators - our approach is much more general and hence much more flexible. **This generalization is of course novel.**
>   - iii) We develop a new optimization formulation for FL: the server-hubs-clients model, described in Section 4. **This alone is a new and innovative contributor to the FL literature, independently of our algorithmic contributions.**
>   - iv) We design a new variance-reduced gradient estimator that can operate in the server-hubs-clients FL model. This is the HUB estimator, described in Section 4. This is an entirely new variance reduced gradient estimator not considered in the SGD literature before. This alone, in our view, is a contribution worth publishing. The literature and research on variance reduced estimators is still growing and thriving. This estimator allows us to benefit both from ProxSkip-like communication acceleration due to hub-level local training, and at the same time from  DIANA-like acceleration of the communication between the clients and the hubs via variance-reduced communication compression. (Our method does not reduce to DIANA, however) **This is all very novel - we are not aware of any previous work that can manage to do this!**
>   - v) Moreover, we show that a specific special version of ProxSkip-HUB, which we call ProxSkip-LSVRG (see Section 4 and Table 2) is a new method for solving the standard FL formulation (1). As such, it is currently the theoretical SOTA among all LT training methods in terms of the total cost, which includes communication cost and also local computation cost (this is all explained in Section 5). **This is very novel - not only we get the current best theoretical result, we are also the first to consider total cost theoretically in the cost of FL in the first place!**
>   - vi) Our work does not only provide general and SOTA theory, we also **conduct experiments that confirm that our method ProxSkip-LSVRG can indeed be much better in terms of total complexity (by an order of magnitude - see top row of Figure 2, for example)**.
>   - vii) Finally, our general framework opens the doors to new developments in the FL literature. We believe it has the potential to be very impactful in the FL community's future work towards advancing the 5th generation of LT methods. This aspect of our work, we believe, is also important.
>
> **We are convinced that the criticism "novelty is limited" does not do justice to our paper at all.**
>
> We would appreciate if the reviewer could reflect on our response and let us know if we managed to convince him/her that our work has a lot of novelty in it.

---

> ### Author Response · Authors · 2022-07-29
> **Question 1: Complexity of FedLin (part 1)**
>
> ---
>
> Table 1 says the communication complexity of FedLin is worse than GD. But the communication complexity of FedLin is also reduced by a factor of $H$ compared with GD, where $H$ is the number of local updates (see the discussion below Eq. (9) of [Mitra et al., 2021]). Therefore, the communication complexity of FedLin can also attain $O(\sqrt{L/\mu} \log(1/\epsilon))$ by setting $H = \sqrt{L/\mu}$.
>
> ---
>
> - **We are quite surprised by this statement, which is of course incorrect.** Please retract it. While it is well known that FedLin can only attain the GD-like *non-accelerated* communication complexity $O((L/\mu) \log(1/\epsilon))$ at best, and this is *very simple to check* by inspection their paper and results, we will explain below, in several alternative ways, why what the reviewer claims is incorrect.
>
> - **Questioning the logic used by the reviewer:** By the same logic as used by the reviewer, one can set $H = L/\mu$ and "prove" that FedLin converges in $O((L / \mu) (1/H) \log (1/\epsilon)) = O(1 \log (1/\epsilon))$ communication rounds, which contradicts existing lower bounds (https://jmlr.org/papers/volume20/19-543/19-543.pdf). The logic is incorrect.
>
> - **A direct quote from [Mitra et al., 2021]:** “Thus, relative to a centralized baseline, FedLin incurs the same computational cost in terms of gradient queries, and **reduces communication by a factor of $H$, at the expense of a convergence rate that is slower by a factor of $H$**. We emphasize here that just as with FedLin, $H$ does not show up in the convergence rate (exponent) of algorithms like FedSplit [10] and SCAFFOLD [11] either.” The improvement by a factor of $H$ and slowdown by the same factor cancel out, and the resulting communication complexity is the same as that of GD.
>
> - **Another quote from [Mitra et al., 2021]:** “The primary reason for the slower convergence rate (relative to a centralized baseline) stems from the need to set $\eta \propto  1/H$ to mitigate client-drift under objective heterogeneity. At this stage, **one may conjecture that the above requirement is simply an artifact of a conservative analysis of Algorithm 1, and that a more refined analysis will reveal the utility of performing more local steps even in the heterogeneous setting.** Our next result suggests otherwise; for a proof, see Appendix E.” FedLin analysis needs stepsizes that are $H$ times smaller than standard GD stepsizes (ProxSkip and our methods do not). In the end, this means that FedLin cannot beat GD in terms of communication complexity in the heterogeneous data regime. Methods belonging to Generation 5 of LT methods (= ProxSkip and our methods) do.
>
> - **Quote from the arXiv version of FedLin:** “From Theorems 1, 2, 3, and 4, we note that **FedLin matches the guarantees of centralized gradient descent** (up to constants) for the strongly convex, convex, and non-convex settings, respectively, despite arbitrary heterogeneity in the client objectives, and arbitrary device heterogeneity.”
>
> - **Another quote from arXiv version of FedLin:** “Unfortunately, Theorem 1 is unable to capture any such advantage for the heterogeneous setting. The primary reason for the slower convergence rate (relative to a centralized baseline) stems from the need to set $\eta \propto 1/H$ to mitigate client-drift under objective heterogeneity. Loosely speaking, **any progress made during the course of the $H$ local iterations gets “washed away” when the learning rate $\eta$ is scaled down by $H$."**

---

> ### Author Response · Authors · 2022-07-29
> **Question 1: Complexity of FedLin (part 2)**
>
> ---
>
> Table 1 says the communication complexity of FedLin is worse than GD. But the communication complexity of FedLin is also reduced by a factor of $H$ compared with GD, where $H$ is the number of local updates (see the discussion below Eq. (9) of [Mitra et al., 2021]). Therefore, the communication complexity of FedLin can also attain $O(\sqrt{L/\mu} \log(1/\epsilon))$ by setting $H = \sqrt{L/\mu}$.
>
> ---
>
> **In this part 2 response to Question 1, we will explain why the reviewer is incorrect by directly inspecting the relevant theorem from the FedLin paper (https://proceedings.neurips.cc/paper/2021/file/7a6bda9ad6ffdac035c752743b7e9d0e-Paper.pdf).**
>
>
> - Theorem 1 from the FedLin paper provides rates for full-gradient case in the strongly convex regime:
>
> **Theorem 1. (Strongly convex case)** Suppose each $f_{i}(x)$ is $L$-smooth and $\mu$-strongly convex. Moreover, suppose $\tau_{i} \geq 1, \forall i \in \mathcal{S}$, and $\delta_{c}=\delta_{s}=1$. Then, with stepsizes $\eta_{i}=\frac{1}{6 L \tau_{i}}, \forall i \in \mathcal{S}$, FedLin guarantees:
>
> $$ \Delta_T \leq (1- 1/ (6 \kappa) )^{T}  \Delta_0, $$
>
> where $\Delta_T = f( \bar{x}_{T+1})-f(x^{*} )$,
>
> $\kappa = L/\mu$ is the condition number, and $x^{*}$ is the minimizer of $f$.
>
> **Note that the rate does not depend on the number of local steps $\tau_i$. The stepsize does, but the rate does not. The corresponding communication complexity guarantee resulting from the above rate is $O(\kappa \log \frac{1}{\varepsilon})$ -- same as GD. By setting $\tau_i=\sqrt{\kappa}$, the communication complexity does not change. What the reviewer write is incorrect.**
>
> - Further, Theorem 4 provides rates for stochastic case in the strongly convex regime:
>
> **Theorem 4. (Strongly convex case with noise)** Consider the above stochastic oracle model. Suppose each $f_{i}(x)$ is $L$-smooth and $\mu$-strongly convex. Moreover, suppose $\tau_{i} \geq 1, \forall i \in \mathcal{S}$, and $\delta_{c}=\delta_{s}=1$. For each $i \in \mathcal{S}$, let $\eta_{i}=\frac{\bar{\eta}}{\tau_{i}}$, where $\bar{\eta} \in(0,1)$ satisfies $\bar{\eta}<\frac{1}{6 L}$. Then, $\forall t \in[T]$, FedLin guarantees:
>
> $$
> \mathbb{E} [ d_{t+1} ] \leq\left(1-\frac{\bar{\eta} \mu}{2}\right) \mathbb{E}\left[ d_t \right]+25 \bar{\eta}^{2} \sigma^{2} .
> $$
>
> where $d_{t+1} = || \bar{x}_{t+1}-x^{*} ||^{2}$.
>
> Note that if we set $\sigma = 0$ then we recover the rate of deterministic FedLin (up to numerical constants).
>
> **Note again that the rate does not depend on the number of local steps $\tau_i$.**
>
> In this theorem, $T$ is the number of communications according to description of Algorithm 1. **This is clearly the rate of GD method. The above rate can be used to derive a bound on communication complexity $T$ using standard arguments:  FedLin has $O(\kappa \log \frac{1}{\varepsilon})$ communication complexity in the deterministic (i.e., full gradient) case**, and non-accelerated sublinear communication complexity  in the stochastic case.

---

> > ### Comment · Reviewer_VKkX · 2022-08-05
> > **Response to "Complexity of FedLin"**
> >
> > Sorry for the incorrect communication complexity of FedLin. I have checked that FedLin does have the same communication complexity as GD.

---

> > > ### Author Response · Authors · 2022-08-05
> > > **Re: Response to "Complexity of FedLin"**
> > >
> > > Thanks for checking and acknowledging this; this is very much appreciated! We were most happy to explain.

---

> ### Author Response · Authors · 2022-07-29
> **Question 2: FedLin vs Scaffold in experiments**
>
> ---
>
> In [Mitra et al., 2021], the authors show that FedLin converges much faster than SCAFFOLD in experiments of the linear model. However, the performance of FedLin in Section 5 is worse than SCAFFOLD and Local SGD, which is unreasonable since FedLin has a linear convergence rate. And I don't think the difference between logistic regression and linear regression is the reason. So I doubt the results in Section 5, could you explain the reason?
>
> ---
>
> **Our results are correct. The thought process you provided is not evidence that our results are not correct. We do not want to dodge the question, just wanted to have this on record. We will try to give some plausible explanations.**
>
> - **First, some clarifications:** FedLin has a  linear convergence rate  in the deterministic setting only (their Theorem 1). In the stochastic case, Scaffold and FedLin do *not* have linear rate (to the exact solution) because of the non-diminishing variance of the gradient estimators (their Theorem 4). In this case, Scaffold and FedLin converge linearly, but only up to some neighborhood of the solution whose size depends on the variance (https://arxiv.org/abs/1910.06378, https://arxiv.org/abs/2102.07053). Local SGD converges up to some neighborhood of the solution whose size depends on the variance and heterogeneity [https://arxiv.org/abs/1909.04746].
>
> - **There might be three reasons for such a behavior:**
>
> 1) Mitra et al. (2021) used **tuned stepsizes**: “and use a step-size of $10^{-3}$ for both algorithms (the step-size was tuned to get best results).” Such stepsizes are not supported by theory. It is know (see the experiments in Mishchenko et al, 2022) that both FedLin and Scaffold perform much better with tuned stepsizes than with theoretical stepsizes. In our work we used only theoretical stepsizes. That is, we examine all methods **as analyzed**. We do so since we wish to give insights into what the theory predicts, not what can be achieved with clever heuristics not supported by theory.
>
> 2) In the Scaffold paper and in the theory of Local SGD, the variance of the gradient estimators is divided by factor of $M$, where $M$ is the number of clients. In FedLin, we do not have such division (check Theorem 4 (Strongly convex case with noise)).
>
> 3) We used all constants that are stated in theorems. Scaffold has large numerical constants in stepsizes, but Local SGD does not have such large constants. So, in the beginning of the training, Local SGD is faster, but after some number of communication rounds, Scaffold starts to be better. Unfortunately, it is not always the case for FedLin since it is more sensitive to the variance of the gradient estimator. However, with increasing the batch-size, we can see that the gap between Local SGD and FedLin is decreasing. With large enough batch size, FedLin is better than local SGD since it can control the client drift but Local SGD cannot.

---

> > ### Comment · Reviewer_VKkX · 2022-08-05
> > **Response to "FedLin vs Scaffold in experiments"**
> >
> > Thanks for your response!
> >
> > 1. About the stochastic case. The experiment in your paper and FedLin are both **finite-sum** problems, which are deterministic settings if you use the full gradient to construct the variance-reduced term. And you did not mention that in your paper.
> >
> > 2. About the tuned stepsizes in Mitra et al. (2021). I think the experiments are used to support the superiority of the proposed method over other baselines in a fair and practical setting. It is fine to verify the theory predicts, but it is better to report the additional results for *all methods* with tuned stepsizes since we barely use untuned parameters in real applications. In addition, you should clarify the choice of hyper-parameter in the paper to avoid misunderstandings.

---

> > > ### Author Response · Authors · 2022-08-05
> > > **Re: Response to "FedLin vs Scaffold in experiments"**
> > >
> > > Thanks for engaging with us, much appreciated!
> > >
> > > 1. **We are quite unsure what the reviewer is trying to say here.**
> > >
> > >  - Yes, we work with finite-sum problems in the paper, as clearly indicated in (1) and also (4). In our paper, $M$ equals the number of clients, or the number of hubs (in Section 4). This is natural, as in federated learning the number of clients is always finite. There is a finite number of phones, companies, or any kind of devices on the planet. Our experiments are also in that setting, and this is clear: indeed, the first sentence of the experimental section directly refers to the finite-sum problem (1): "To illustrate the predictive power of our theory, it suffices to consider L2-regularized logistic regression in the distributed setting (1), ...
> > >
> > > - Next, **all variants of ProxSkip-VR in this paper, with the exception of a single variant, ProxSkip (= ProxSkip-GD in our terminology; see Theorem 6 and the first line of Table 2), are stochastic. In our experiments, all methods we compare to are stochastic, including ProxSkip-LSVRG, which is the method we work with in Figure 2.** We use Local SGD, Scaffold and FedLin in the stochastic setting, i.e., the $M$ clients use stochastic rather than full gradients. SProxSkip is the terminology used in Mishchenko et al (2022) and refers to the method we call ProxSkip-SGD in our paper. It is stochastic.  The only place where a deterministic method cane be visible in Fig 2 is in the bottom row, where the ratio compares ProxSkip-LSVRG to ProxSkip. That is, we should the improvement our approach offers.
> > >
> > > - In the light of the above, we are afraid we do not know what you want to say when you say "The experiment in your paper and FedLin are both finite-sum problems, which are deterministic settings if you use the full gradient to construct the variance-reduced term. And you did not mention that in your paper." **What you say is correct, but there is a problem with the "if" in the sentence we just quoted. We do *not* use full gradients in any of the plots in top row of Fig 2 (and we use it in the bottom row of Fig 2 for a good reason), and hence the "if" does not apply.** We did not mention what you wanted us to mention since we did *not* use full gradients. We hope this addresses your concern -- **it seems you perhaps misunderstood what we do in the experiments? Since we are still confused about what exactly you mean, would it be possible for you to elaborate? We are always happy to answer any questions.**
> > >
> > > 2. **In all our experiments we used all hyper parameters for all methods as indicated by the theory; we will make this clear; thanks for pointing this out!** This is *fully fair* if what we want to test is the reflection of the theory in an experiment. Our experiments underscore the vast superiority of ProxSkip over Generation 4 and 3 baselines, and the additional improvement offered by ProxSkip-VR. **It is true that we did not conduct experiments with fine-tuned parameters. We will add such experiments in the supplementary material, and we agree these kinds of experiments would be also interesting.** However, please notice that fine-tuning can be very expensive, especially in federated learning. It does not come for free, and in some sense, amounts to "cheating". That is, if one performs, say, 10 runs of a method (with 10 settings for the stepsize, for exmaple), only to report the best run, this means that 10 times as much work was needed to obtain the results. Our theory provides excellent baseline settings for the parameters, and this goes a very long way in mitigating the need for tuning. Surely, just like Generation 4 methods, our methods will do even better with additional tuning.
> > >
> > > Hope this addresses your concerns. If not, please do elaborate, and we will be very happy to respond!

---

> ### Author Response · Authors · 2022-07-30
> **Limitations (part 1)**
>
> ---
>
> This paper identifies 5 distinct generations of LT methods and classifies ProxSkip and ProxSkip-VR in Generation 5: Accelerated Age, but I don't think the full-gradient and variance-reduced methods are practical and popular in FL. Considering large-scale machine learning models, stochastic gradient descent (and its variants) with the constant batch size is a prevalent training scheme. So the summarization in this paper may be less appropriate.
>
> ---
>
> Your belief that "full-gradient and variance-reduced methods are not practical and popular in FL", whether it is true or not, has little to do with the theoretical advances we make in this paper. We do not see how this is a criticism of our work. We will elaborate:
>
> - **On "popularity":** For the sake of the argument, let us *assume* that it is the case that "full-gradient and variance-reduced" methods are not popular. Why would this be an issue? Breakthroughs and paradigm changes necessarily happen before something before popular; popularity comes later as the new ideas get more traction and more work is done in the area. Should research only happen for methods that are popular? Of course not, this is not how science should work. It makes sense to study popular methods, yes, and it also makes sense to be creating pathways into new territories. We do not see how popularity is a necessary property of a method or class of methods worthy of study, or if this is not what you are trying to say, why this would be a serious limitation of our (or any) work.
>
> - **On "popularity" and how it relates to or not to what we do:** Having said that, please recall that we study new variants of LT (local training) methods - and LT methods are immensely popular in FL; so much that the term "federated optimization" is often used to refer to methods based on the "local training" paradigm. There are thousands of papers on LT methods since LT became popular in around 2015 (Povey et al, 2015; Moritz et al, 2016, McMahan et al, 2017), and none of them except one (ProxSkip by Mishchenko et al, 2022) show theoretical advantage in terms of communication complexity over vanilla GD in the heterogeneous data regime. If a breakthrough is to happen, new ideas are needed, and this is what Mishchenko et al (2022) did with their ProxSkip method. This work initiated Generation 5 of LT methods, and opens the door to new possibilities. We investigated one such possibility: further enhancing ProxSkip with a very flexible variance reduction framework, and succeeded. This enables the FL community to ask, for the first time, deeper questions such as: how to design methods that minimize the total cost, which includes communication and computation cost. The coarser theoretical approaches characterizing Generation 4 and earlier generations of LT methods were not in the position to ask these deeper questions because they were still struggling with explaining why, when and how LT leads to provable communication acceleration. Even if variance reduction was not popular in earlier generations of LT methods (and we do not believe this claim; own this later), it does not mean it will not become popular in Generation 5 LT methods. In fact, we hope that our work shows how variance reduction is helpful to improve LT methods.
>
> - **Variance reduction is popular in FL:** Having said that, we challenge the reviewer's claim that VR methods were *not* popular in FL. For example, all Generation 4 methods, including Scaffold, FedLin and S-Local-GD (see Table 1) gained their Generation 4 status precisely because of a specific type of variance reduction they employ: "client drift reduction". Client drift reduction reduces the variance of the local models obtained via local training due to them drifting apart because of data heterogeneity. While this type of VR is not the same as variance reduction of the variance coming from subsampling/minibatching (such as SVRG, SAGA, SAG, SDCA, MISO, LSVRG, ...) nor variance reduction of the variance coming from communication/gradient compression (such as DIANA), the ideas are very similar. Indeed, Scaffold is explicitly based on the SAGA mechanism. Scaffold has more than 600 citations (https://scholar.google.com/citations?view_op=view_citation&hl=en&user=8l-mDfQAAAAJ&citation_for_view=8l-mDfQAAAAJ:hC7cP41nSMkC). As you can see, arguably, variance reduction *is* very popular popular!
>
> - **Variance reduction and deep learning:** Next, it is true that variance reduction is generally not popular when training deep learning models (see "On the ineffectiveness of variance reduced optimization for deep learning" by Defazio and Bottou). However, our paper is *not* about the nonconvex regime, let alone about deep learning, which poses a very specific type of nonconvex regime. Our method can't be criticized just because it may not work well in regimes it was not designed for in the first place (if that is the case).

---

> ### Author Response · Authors · 2022-07-30
> **Limitations (part 2)**
>
> ---
>
> This paper identifies 5 distinct generations of LT methods and classifies ProxSkip and ProxSkip-VR in Generation 5: Accelerated Age, but I don't think the full-gradient and variance-reduced methods are practical and popular in FL. Considering large-scale machine learning models, stochastic gradient descent (and its variants) with the constant batch size is a prevalent training scheme. So the summarization in this paper may be less appropriate.
>
> ---
>
> - **Re. the claim that "full gradient methods are not popular":** Please note that none of the new methods we develop in this paper use the full gradient. On the other hand, this is exactly what ProxSkip (in our notation, ProxSkip-GD) does. All nontrivial variants of our general method ProxSkip-VR, such as ProxSkip-SGD (already considered by Mishchenko et al, 2022), ProxSkip-HUB and its special cases ProxSkip-LSVRG and ProxSkip-Q (see Table 2), use some sort of a stochastic gradient. In the case of ProxSkip-LSVRG, stochasticity comes from client subsampling (or, equivalently, data subsampling, if we think of server-hubs-clients as server-clients-data instead), and in case of ProxSkip-Q, stochasticity comes from gradient compression. Our framework goes further, and captures a more general estimator than HUB in the server-hubs-clients model in which each client computes a stochastic gradient instead of the full gradient. Assumption 4 is satisfied, and such a method can be analyzed as a corollary of Theorem 5. We will attempt to add this method and its analysis to the supplementary material. In summary, as you can see, restriction to the "full gradient" setting, whether the setting is popular or not, which is irrelevant, is not an issue with our paper. Indeed, we offer the most flexible general and non-full-gradient setting to study ProxSkip-VR!
>
> - **Re. large scale ML models and SGD:** We believe that the limitation "Considering large-scale machine learning models, stochastic gradient descent (and its variants) with the constant batch size is a prevalent training scheme. So the summarization in this paper may be less appropriate" claimed by the reviewer is misguided. Notice that in this paper we specifically set out to study LT methods. SGD is *not* an LT method since it only takes one SGD step in between communications. On the other hand, Local SGD *is* an LT method, and is much faster in practice than SGD! Indeed, on lines 79-83 we cite McMahan et al (2017), who say, among other things, that "Communication costs are the principal constraint, and we show a reduction in required communication rounds by 10$\times$--100$\times$ as compared to synchronized stochastic gradient descent." In summary, we do not include SGD in Table 1, even though the reviewer seems to think this is an omission, which it is not, because i) SGD is not a LT method and our paper is about LT methods, and because ii) Local SGD, which is an LT method, is much faster in practice (again see the quotes from lines 79-83)! In fact, Local SGD  belongs to Generation 3 of LT methods (see Table 1), just as its full-batch sibling, Local GD. ProxSkip is vasty superior to GD and Local GD, and ProxSkip-SGD is vastly superior to Local SGD and to all LT methods from Generation 4 (Scaffold, S-Local-GD, FedLin). This was shown buy Mishchenko et al (2022) already. We automatically beat Local SGD since ProxSkip-LSVRG beats both ProxSkip-GD and ProxSkip-SGD. What we are trying to say is: there is no omission, Table 1 does not need to include SGD.

---

> ### Author Response · Authors · 2022-07-30
> **Limitations (part 3)**
>
> ---
>
> Considering existing works [Mitra et al., 2021] and [Mishchenko et al. 2022], the contribution and novelty are limited. More importantly, the experiment results are not convincing.
>
> ---
>
> - **Contribution and novelty are limited:** We already responded to the "contribution and novelty are limited" claim in air post "Weakness 2: the improvement is natural". In summary, we believe that this claim is whole unjustified and misguided.
>
> - **Re. experiments:**
>
>   - You already said when enumerating the strengths of our paper that **"The authors corroborate their theoretical results with carefully engineered proof-of-concept experiments."** How can this be true, and the experiments be unconvincing at the same time? What exactly is not convincing?  We do not see any evidence nor specific criticism in your review pointing to issues with some specific experiments we conduced.
>
>   - **We believe the experiments in Section 5 are **very convincing**. We have theory predicting the speedup (see Equation (13) and the blue line in each of the three plots in Fig 2, bottom row) of ProxSkip-LSVRG over ProxSkip, and algorithm runs (Fig 2) confirm that we can obtain 10$\times$ speedup in terms of total cost! This is 1000% improvement! How can a combination of strong theory and corroborating experiments not be convincing?**

---

> ### Comment · Reviewer_VKkX · 2022-08-07
> **Response to authors**
>
> Thanks for your detailed response and discussion, and I have raised my score.

---

### Official Review · Reviewer_RRdw · 2022-07-14

**Rating:** 7
**Confidence:** 4
**Soundness:** 3 good
**Presentation:** 3 good
**Contribution:** 3 good

**Summary:**

The paper proposes a new variance reduced variant to ProxSkip, called ProxSkip-VR, for federated learning. It possesses a combination of preferable techniques in federated learning including data sampling, client sampling and local training. The variance reduced estimator is general and covers numerous existing estimators. Linear convergence is proved for the strongly convex objective. More importantly, the paper take a closer look on the cost of training FL methods by incorporating local training cost and communication cost whereas most existing FL methods only consider communication cost in convergence results. Numerical experiments on regularized logistic regression is presented to illustrate the performance of ProxSkip-VR compared to existing methods.

**Questions:**

- ProxSkip-VR assumes that the stochastic gradient $g_t$ is unbiased. Can the theory be generalized to biased estimators to cover this type of gradient estimators?
- Does the proof of ProxSkip-VR in Appendix C hold for any choice of $y_t$ or it has to be a certain choice so that $g_t$ satisfies Assumption  4?

**Limitations:**

With the definition of $r$ in (4), I think assumption 3 is not needed.
I hope to see more discussion on how to select $y_t$ in the main text.

**Strengths And Weaknesses:**

Strengths:
- The paper is well-written. I can see the history of FL methods evolving over different generations.
- ProxSkip-VR possesses several important properties for an FL method: client sampling, data sampling, and variance reduced local update step.
- The paper proposes a new architecture for FL which contains hubs between server and workers. This architecture is reasonable and it represents many real-world applications.
- Convergence results are provided which cover different type of gradient estimators.
- The evaluation considers the total cost which consists of communication cost and local training cost which has not been considered in existing works. This also help show the advantage of using variance reduced estimator in local training.

Cocerns:
- Table 1 summarizes results of ProxSkip-VR w.r.t. existing works. It shows that communication complexity of ProxSkip-VR is better than GD but it does not explicitly discuss how it is better and the reader needs to go back and check the other paper to learn about this comparison.
- I do not see how the control vector $y_t$ is chosen in the proof of ProxSkip-VR (Theorem 5). I wonder if $y_t$ is chosen the same as in (11).

Minor comments:
- The experiments has a non-zero value for the penalty parameter $\lambda$ for $\{x\}^2$ regularizer but the code appears to set it to 0. I am not sure if the code precisely reflect the setting of the experiments described in the paper.

---

> ### Author Response · Authors · 2022-07-29
> **Concern 1: "Table 1"**
>
> ---
>
> *Table 1 summarizes results of ProxSkip-VR w.r.t. existing works. It shows that communication complexity of ProxSkip-VR is better than GD but it does not explicitly discuss how it is better and the reader needs to go back and check the other paper to learn about this comparison.*
>
> ---
>
> - **We agree we did not specify in Section 2 nor Table 1 what *exactly/precisely* we meant by "better than GD" or "accelerated", and that mentioning it at this point is better than either relying on the patience of the reader to get the answer in Table 2 (more on this below), which is located further in the paper, or expecting the reader to first read the ProxSkip paper. Som this point is well taken. We will make this clear in two ways: a) by modifying the table with an extra note, and b) by adding a bit more detail to our description of generation 4 and 5 in Section 2.**
>
> - Please note though that it is Table 2 whose purpose is to compare the communication and iteration complexity of our results compared to  the previous state of the art of Mishchenko et al (ICML 2022). Our results are highlighted in light green color, while previous theoretical SOTA is contained in the first two rows of the table (= ProxSkip-VR using the GD and SGD estimators, respectively). We did *not* compare to any previous generation method as this is not necessary. Indeed, Section 2 and Table 1 explain that previous generation methods do not even achieve better-than-GD communication complexity (for arbitrarily heterogeneous data, which is the regime we consider). In other words, we build and improve upon ProxSkip, which is the current state of the art in terms of theoretical communication complexity, and hence it is enough to compare to ProxSkip.
>
> - The main purpose of Table 1 is to provide a breakdown of theoretical progress in the area of LT (local training) methods (in the smooth strongly convex regime) into 5 generations and *not* to specify the complexities. Our goal here to set our work in the context of prior advances, and to make it clear that our contributions belong to the latest generation of LT methods, which are characterized by "accelerated" communication complexity. We believe the table serves this purpose well. As such, Table 1 provides an overview from a bird's eye perspective over the entirety of research on the topic of LT methods, and as such necessarily needs to leave out details. What separates generation 4 from generation 5 methods was made clearly enough, we believe: while generation 4 methods provide (at best) GD-type communication complexity rates, generation 5 offers "accelerated" (= better than GD) rates. We believe this is made reasonably clear in the table, and in the text from Section 2.
>
> Please let us know if this reply is satisfactory to you.

---

> ### Author Response · Authors · 2022-07-29
> **Concern 2: the choice of the control vector**
>
> ---
>
> *"I do not see how the control vector $y_t$ is chosen in the proof of ProxSkip-VR (Theorem 5). I wonder if $y_t$ is chosen the same as in (11)."*
>
> *"Does the proof of ProxSkip-VR in Appendix C hold for any choice of $y_t$ or it has to be a certain choice so that $g_t$ satisfies Assumption 4?"*
>
> ---
>
> - In Theorem 5 and its proof we require the the gradient estimator $g_t$ (and the control vector $y_t$ which gives rise to the estimator) to satisfy Assumption 4. That is all. This is a general assumption that includes a large array of known (and many yet to be discovered) gradient estimators. Just before Theorem 5 we say: "Assumption 4 covers a very large collection of gradient estimators, including an infinite variety of subsampling/minibatch estimators, gradient sparsification and quantization estimators, and their combinations; see [Gorbunov et al., 2020b] for examples." We give two simple examples of gradient estimators in Section 3.2: i) GD estimator (see Theorem 7) and ii) SGD estimator satisfying an expected smoothness assumptions (see Theorem 8). These estimators are simple (i.e., not variance reduced), and as such do not depend on any control vectors. Technically, we can set $y_t\equiv 0$ for these estimators. These details can be found in [Gorbunov et al., 2020b].
>
> - In Section 4 we define an entirely new variance-reduced gradient estimator, one that was not considered in the literature before. We believe that this alone could have been written up as a stand-alone paper, thus contributing to the literature of variance reduced SGD methods (such as SVRG, SAGA, DIANA and so on), without combining the estimator with ProxSkip. This estimator is defined in (12), with the control vector being defined in (11); see also (10) which is a part of the definition as (12) relies on it. We call this the HUB estimator. Theorem 9 is an application of Theorem 5. In order to apply it, we need to check that this new estimator (12) satisfies Assumption 4. Please note we say this just before stating Theorem 9: "Our proposed method is thus ProxSkip-VR combined with the novel estimator (12). The following result first claims that the above estimator satisfies Assumption 4 with $C = \tilde{C} = 0$ (i.e., it is variance-reduced), and the rest of the claim follows by application of our general theorem Theorem 5.".
>
> - So, in summary, (11) is how the control vector is defined when our general method (Algorithm 1) is used with the HUB estimator defined in (12) (which depends on (10) and (11)). However, ProxSkip-VR is a much more general method, and Theorem 5 and its proof rely in Assumption 4 only to characterize the estimator and the control vector giving rise to it. Table 2 lists 5 special cases of ProxSkip-VR; the HUB estimator leads to one of these.
>
> We hope this settles your question. Please do not hesitate to ask if you have any further questions!

---

> ### Author Response · Authors · 2022-07-29
> **Concern 3 (minor comment): experiments and the penalty parameter**
>
> ---
>
> The experiments has a non-zero value for the penalty parameter $\lambda$ for $||x||^2$ regularizer but the code appears to set it to 0. I am not sure if the code precisely reflect the setting of the experiments described in the paper.
>
> ---
>
> - We did *not* use $\lambda = 0$ in our experiments. Explanation is below:
>
> - Similarly to the experimental setup of the original ProxSkip paper, we set the regularization parameter as $\lambda = ratio * L$ for some constant $ratio$. In order to do so, we first need to compute $L$ (e.g., see L95 in the file main_0004.py); that is, we first build $f(x)$ without any regularizer and get $L$ (L89). After that, we set $\lambda = ratio * L$ (L98). We then add the regularizer terms to each function $f_i(x)$ (L134), and subsequently denote the whole integrated object as ``losses" (L136), which is used for further computations on the function value and the function's gradient.
>
> - Looking at the code, we now noticed that in the uploaded files main_0002.py and main_0007.py we set $\lambda = ratio$; whereas it should have been set to $\lambda = ratio * L$, as explained above. This is because for some reason we uploaded old versions of these two files. The experiments were conduced in the correct way, however.

---

> > ### Comment · Reviewer_RRdw · 2022-08-06
> > **Re: cocern 3**
> >
> > Thanks for clarifying this. My mistake for not carefully check this. Now when I read the code again, you first initialize the loss with $\ell2=0$ then change it later. I only skim through the code the first time so I did not catch this.

---

> > > ### Author Response · Authors · 2022-08-06
> > > **Re: Re: cocern 3**
> > >
> > > Thanks a lot for your time and efforts and for checking this again!
> > >
> > > authors

---

> ### Author Response · Authors · 2022-07-29
> **Concern 4: possible extension to biased gradients?**
>
> ---
>
> ProxSkip-VR assumes that the stochastic gradient $g_t$ is unbiased. Can the theory be generalized to biased estimators to cover this type of gradient estimators?
>
> ---
>
> - This is an interesting question, but beyond the scope of the current work since biased estimators require very different analysis tools.
> However, we are not aware of any theoretical benefits of biased estimators over unbiased ones in the strongly convex regime we consider in this paper. So, from this perspective, such a study would not be well-motivated. In fact, we already thought about this a bit after submitting our paper (as a possible future research direction), but do not have any interesting results to report yet.
>
> - Having said that, to the best of our knowledge, well designed biased estimators are superior (in theory and practice) to the best unbiased ones in the smooth *nonconvex* regime. Examples: SARAH/SPIDER (NeurIPS 2018: https://proceedings.neurips.cc/paper/2018/hash/1543843a4723ed2ab08e18053ae6dc5b-Abstract.html), MVR/STORM (NeurIPS 2019: https://proceedings.neurips.cc/paper/2019/hash/b8002139cdde66b87638f7f91d169d96-Abstract.html), PAGE (ICML 2021: https://proceedings.mlr.press/v139/li21a.html), MARINA (ICML 2021: http://proceedings.mlr.press/v139/gorbunov21a.html). However, it is not at clear whether ProxSkip would bring any theoretical benefits in the nonconvex regime. Indeed, acceleration (whether of Polyak or Nesterov or ProxSkip type) seems to lead to benefits in the convex and strongly convex regimes. A study of communication acceleration via LT in the smooth *nonconvex* regime would also require very different tools; currently this is an important open problem.

---

> ### Author Response · Authors · 2022-07-29
> **Concern 5 (limitations): Assumption 3 and choice of $y_t$**
>
> ---
>
> *"With the definition of $r$ in (4), I think Assumption 3 is not needed."*
>
> *"I hope to see more discussion on how to select $y_t$ in the main text."*
>
> ---
>
> - Our main theorem (Theorem 5) is about solving the general problem (3) under the following assumptions: Assumption 2 (strong convexity of $f$), Assumption 3 ($r$ needs to be proper, closed and convex) and Assumption 4 (VR gradient estimator). So, it holds for *any* regularizer satisfying Assumption 3. In Theorems 6 and 8, which are corollaries of Theorem 5, we also do not need $f$ nor $r$ to have any further special structure, i..e, we solve the general problem (3). **In summary, we can not leave our Assumption 3 from Theorem 5, 6 and 8.**
>
> - In Section 4 and Theorem 9, however, we consider a special-structure problem (see the displayed equation below (12)), where both $f$ and $r$ are defined in some particular ways. The regularizer $r$ defined here (and also in (4)) happens to satisfy Assumption 3, and we indeed do *not* need to invoke Assumption 3 in Theorem 9. **We will remove it from here.**
>
> - We will add a bit more discussion on how to select $y_t$ in the text. In brief, this is discussed in detail in Gorbunov et al. [2020b] (which we say in the paper) -- they give 15 special cases of SGD estimators (see their Table 1 and 2) satisfying this assumption and in each case define what $y_t$ is. For non-variance-reduced methods $y_t$ is not needed; so one can technically set $y_t\equiv 0$. For each variance-reduced method, $y_t$ is defined in some very specific way, which is not possible to describe without describing the full algorithm. ProxSkip-VR can be used with any of these 15 estimators; and also any that satisfy Assumption 4 that were discovered later. Our contribution to the design of *new* variance-reduced estimators (the HUB estimator) is fully described in Section 4, and $y_t$ is defined there.

---

> ### Comment · Reviewer_RRdw · 2022-08-06
> **Response to authors**
>
> I want to thank the authors for the detailed responses about my review. It help improve the clarity of the paper. I suggest the authors to incorporate their responses to the paper as well. After reading the responses, I believe the authors have addressed my concerns. I decide to keep my score.

---

### Official Review · Reviewer_2cyy · 2022-07-18

**Rating:** 7
**Confidence:** 4
**Soundness:** 4 excellent
**Presentation:** 2 fair
**Contribution:** 3 good

**Summary:**

This work theoretically studies the heterogeneous cross-silo federated optimization problem. Previously,  Mishchenko et al. [2022] had shown that SCAFFOLD with randomized number of local steps (which they call Prox-Skip / ScaffNew) enjoyed an accelerated communication complexity, proving improvement due to local steps.

This paper extends their framework to employ variance reduction and compressed communication at the client level, while maintaining the accelerated communication complexity. They also propose an hierarchical federated model where clients are attached to always-available hubs for which their algorithms are particularly well suited.

**Questions:**

# Suggestions for improvement

1. Section 4 is way too terse and hard to comprehend. The setting of the hub model and the algorithm description should be separated (perhaps with pseudo-code). I had to work hard to reconstruct what was being conveyed. E.g. of confusions: why is the global gradient a concatenation of estimators in eq (12)? control vector y_t is used before it is defined (in fact it is never properly defined or initialized) etc.
2. It is also unclear how Theorem 9 relates to Theorem 5?  Why can we not use VR and CC directly in Section 3 with Thm. 5? Thms 6 and 8 are definitely direct corollaries of Thm 5 and it would be clearer if stated as such.
3. More time should be spent explaining the results in Corollaries 1 and 2 - what is $\omega$ and how should we think about it? why does it scale as $\omega / M$ and is the optimal scaling? The latter relies on the independence of the compression randomness, which is not stated. Overall, I think the paper would greatly benefit from moving some of the exposition in Section 2 to the Appendix and instead expanding on the results in Sections 3 and 4, with greater discussion about the implications of the theoretical rates.
3. The result for the setting with C > 0 is not optimal since there is no speed-up wrt to number of clients. The optimal rate likely scales as C/M (as SCAFFOLD obtains).
4. Also, while client and data sampling are discussed in detail in Sec 1.2, their effect on ProxSkip-VR is never discussed.
5. It should be made clear that all the discussion in the paper is for cross-silo FL and does not apply to cross-device FL.
6. Nitpick (feel free to ignore): the colors used in the text (e.g. in math symbols like W and E in Thm 5, and algorithm names) made it visually busy, confusing and harder to read for me. I would suggest using such colors sparsely in text - more as a shorthand (e.g. red is always error terms) rather than for emphasis or drawing attention.

**Limitations:**

The limitations of the method are not adequately discussed. The questions section above goes into detail.

Note: I am willing to raise my score on an assurance from the authors that the writing and presentation will be improved. The results are otherwise technically sound and interesting.

**Strengths And Weaknesses:**

# Strengths

1. The approach is well motivated and forms a natural line of inquiry - combining communication compression + variance reduction with ProxSkip.
2. The results obtained by the authors, though not surprising in the light of ProxSkip, are significant and technically involved. It is a result of synthesizing several state of the art analysis techniques, and recover most of them as state of the art.
3. I also enjoyed the motivation of the local variance reduction via the hub model and can see it being useful.

# Weaknesses

Most of the weaknesses stem from poor writing (see questions section for detailed feedback) - i) section 4 is very terse and not elaborated well , and ii) the limitations of the method and comparison with prior work is not well carried out.
The most impactful aspect of this paper might be the new hierarchical FL model proposed. But unfortunately, there is almost no discussion on its unique aspects, optimality etc.

---

> ### Author Response · Authors · 2022-07-29
> **Suggestion 1: Section 4 is too terse**
>
> ---
>
> Section 4 is way too terse and hard to comprehend. The setting of the hub model and the algorithm description should be separated (perhaps with pseudo-code). I had to work hard to reconstruct what was being conveyed. E.g. of confusions: why is the global gradient a concatenation of estimators in eq (12)? control vector $y_t$ is used before it is defined (in fact it is never properly defined or initialized) etc.
>
> ---
>
> - **We indeed had to pack a lot of material into little space provided by the page limit; we also feel this section is a bit terse in some parts. So, this criticism is warranted, and we will implement a remedy (which is very easy to do).**
>
> -  In particular, you are right that we skipped some details related to the conversion between formulation (1) which we use to describe the gradient estimator, and the *equivalent* formulation $\min \frac{1}{m} \sum_{j=1}^m \tilde{\phi}_j(x) + r(x)$ which we use to *analyze* method ProxSkip-HUB in Theorem 9. This is indeed confusing, and we will fix this. We believe this is the source of most of the confusion you describe. We hope to manage to do it soon so that we can upload the modified pdf file to OpenReview for you to check.
>
> - In Section 4 describing ProxSkip-VR using the HUB gradient estimator, the control variate $y_t$ is fist mentioned in (10), and defined a bit later, yet still *in the same sentence* in (11). We believe this is standard.
>
> - It is true we did not say in Section 4 how $y_t$ is initialized: we can choose $y_0$ arbitrarily (we will add a comment about this). However, we did say this in the appendix, where we provided pseudocode for the method as Algorithm 2.

---

> ### Author Response · Authors · 2022-07-29
> **Suggestion 2: Theorem 9 vs Theorem 5**
>
> ---
>
> It is also unclear how Theorem 9 relates to Theorem 5? Why can we not use VR and CC directly in Section 3 with Thm. 5? Thms 6 and 8 are definitely direct corollaries of Thm 5 and it would be clearer if stated as such.
>
> ---
>
> -  **Theorem 9 can be seen as an application of Theorem 5** to the optimization formulation
>
> $$\min \frac{1}{m} \sum_{j=1}^m \tilde{\phi}_j(x) + r(x)$$
>
> (which is equivalent to (1) in the case when $n_i=n_j$ for all $i,j$) and the HUB gradient estimator described in Section 4. Note that in the sentence immediately preceding Theorem 9 we say: "The following result first claims that the above estimator satisfies Assumption 4 with $C = \tilde{C} = 0$ (i.e., it is variance reduced), and the rest of the claim follows by application of our general theorem Theorem 5." If you look at the proof of Theorem 9, you will see that at the end we invoke Theorem 5. Most of the proof of Theorem 9 is checking that Assumption 4 holds for the HUB gradient estimator.
>
> - **We can't apply Theorem 5 directly since Theorem 5 deals with the more general problem (3)**, i.e., with
>
> $$ \min f(x) + r(x),$$
>
> where we work with a general smooth and strongly convex function $f$ (see Assumption 2), and a general proper, closed and convex regularizer $r$ (see Assumption 3), and a generic VR estimator (see Assumption 4). In Section 4, on the other hand, we specify $f(x) = \frac{1}{m} \sum_{j=1}^m \tilde{\phi}_j(x) $, where $x=(x_1,\dots,x_M)$, and choose a very specific regularizer satisfying Assumption 3: $r(x) = 0$ if $x_1=\dots=x_M$ and $r(x)=+\infty$ otherwise. We then define a particular VR estimator that we call HUB. In order to apply Theorem 5, we need to prove that it satisfies Assumption 4. This is what most of the proof of Theorem 9 is devoted to.
>
> - **Re corollaries vs theorems.** We make it clear in the paper that Theorem 5 is the main resut (e.g., Section 3.1 is called "Framework for expressing VR estimators and our main result"; and just before Theorem 5 we say: "Now we are ready to formulate our main result.") Theorems 6 and 7 are direct corollaries of Theorem 5 for the two estimators considered therein (all that needs to be checked is that Assumption 4 holds). We will state this also explicitly by adding a sentence. We make it clear that Theorem 9 is a corollary of Theorem 5 as well: just before the result we say "the rest of the claim follows by application of our general theorem Theorem 5." The reason why we did *not* call Theorems 6, 7 and 9 corollaries is because i) we think they are important on their own (Theorems 6 and 7 received the results of Mishcheno et al and Theorem 9 is the key example of a new variance-reduced version of ProxSkip we propose in the paper, ii)  we already have 2 consequences of Theorem 9 which we call Corollary 1 and Corollary 2. It would be confusing to call Theorem 9 a corollary, and then have 2 corollaries of a corollary. Hope this sheds light on how we thought about this. However, we appreciate that this still may be a bit confusing, and will clarify things a bit more. Thanks for the suggestion.

---

> ### Author Response · Authors · 2022-07-29
> **Suggestion 3: Explain Corollaries 1 and 2 in more detail...**
>
> ---
>
> More time should be spent explaining the results in Corollaries 1 and 2 - what is $\omega$ and how should we think about it? Why does it scale as $\omega/M$ and is the optimal scaling? The latter relies on the independence of the compression randomness, which is not stated. Overall, I think the paper would greatly benefit from moving some of the exposition in Section 2 to the Appendix and instead expanding on the results in Sections 3 and 4, with greater discussion about the implications of the theoretical rates.
>
> ---
>
> - OK, we will add more clarifying text here! If there were no page limits, we would certainly have done so right from the start. page limitations force authors to be unduly terse in some places...
>
> - $omega$ was defined between (10) and (11). It is a characteristic of the compression operators. The larger the value, the larger the compression variance (which is bad), but this allows for more compression, which means that less data is communicated (which is good). The class of unbiased compression operators $Q$ satisfying the inequality $\mathbb{E} || Q(x) - x ||^2 \leq \omega ||x||^2$  for all $x$ is widely studied. We will point to some further papers (besides those we already mention when defining $\omega$) so that the readers unfamiliar with this topic can read about this.
>
> - Yes, the compressors $\{Q_t^{ij}\}$ should be independent and we forgot to state this. Will fix!
>
> - Yes, the $\omega/M$ scaling (i.e., the variance decreases as the number of hubs increases) is indeed a consequence of independence. We do not have any lower bounds to prove optimality of this scaling (in fact, the literature on communication compression for distributed learning is void of any lower bounds).
>
> - While Section 2 is of an introductory nature so that the readers can better appreciate our technical contributions (e.g., Theorems 5 and 9, and the Total Complexity results and test in the Experiments section), it also serves a dual purpose of a reflection on the progress of the field. We think that some people might refer to our work precisely of Section 2 and the overview it provides. Initially this section was about 2 pages longer, and we shortened it before submission. We will think of a way to shorten this further, or move this material to the Appendix.
>
> - Yes, we will expand on Sections 3 and 4, and provide more discussion about the implications of the theoretical rates (in particular, about Theorem 9). We will include some of this material in the main paper, and what we can't include here due to space limitations, we will include in the appendix.

---

> ### Author Response · Authors · 2022-07-29
> **Suggestion 4: Setting $C>0$**
>
> ---
>
> The result for the setting with $C > 0$ is not optimal since there is no speed-up wrt to number of clients. The optimal rate likely scales as $C/M$ (as SCAFFOLD obtains).
>
> ---
>
> Are you referring to $C$ from Equation (5) and Theorem 8? In our response we assume that this is the case.
>
> - Indeed, $C$ in Theorem 8 is *not* divided by the number of clients. We spent several weeks thinking about this before submission, but could not prove a better result. This might be a very difficult problem, or even impossible. But we found a different resolution - please keep reading ....
>
> - One reason behind the difficulty could be: it is known that Nesterov's acceleration does not work with noisy gradients (see  Devolder, Glineur and Nesterov: https://link.springer.com/article/10.1007/s10107-013-0677-5); unless their their noise is removed via variance reduction such as SVRG (see the Katyushua paper by Z Allen Zhu). The noise model considered in Theorem 8 works with a fixed noise/variance level, and this does not work well with acceleration. While our acceleration technique via local training (i.e., via ProxSkip) is not the same as Nesterov's acceleration, **we suspect the same phenomenon may be at play here.**
>
> - Please note that in contrast to Scaffold, whose rate is worse (or at best the same) than the rate of gradient descent, **ProxSkip-SGD (covered by Theorem 8) has an accelerated communication complexity:**
> $O( ( \sqrt{ A / \mu } + \sqrt{ 2C / \epsilon \mu^2 }) \log 1/\epsilon )$. Focusing solely on the term $\sqrt{ 2C / \epsilon \mu^2 } $ which depends on the variance $C$, this is to be contrasted with Scaffold's $C/\epsilon \mu M$. **Due to the presence of the square root, our term can be substantially smaller despite the fact that we do not have an explicit boost from $M$.**
>
> - Our resolution of the question you ask is *not* to merely improve dependence from $C$ to $C/M$ in Theorem 8, which relies in simple (= non variance reduced) SGD estimators characterized in Assumption 7, but to remove dependence on $C$ completely, via variance reduction! This is one of the ways our main result can be interpreted.  Notice, for example, that ProxSkip-LSVRG (see Table 2) has communication complexity $O( \sqrt{ L_\tau / \mu} \log 1/\epsilon)$, and there is no effect of $C$ present here whatsoever! **So, ProxSkip-LSVRG can be seen as a method which resolves the issue you are pointing to far beyond the limited $C/M$ scaling - we have no dependence on $C$ here whatsoever!**
>
> We hope this answers your question. If we did not understand what you meant, please clarity and we will be most happy to respond.

---

> ### Author Response · Authors · 2022-07-29
> **Suggestion 5: Client and data sampling (part 1)**
>
> ---
>
> Also, while client and data sampling are discussed in detail in Sec 1.2, their effect on ProxSkip-VR is never discussed.
>
> ---
>
> - **On client sampling:** Section 4 is dedicated to client sampling (and additionally, client communication compression) in the HUB model. Please note that in this model the hubs are always available, and hence not sampled. Theorem 9 covers this case; and the communication and iteration complexity results are also summarized in the "HUB row" of Table 2. We do comment on these results in the paragraph following Theorem 9. In particular, we show that in the client sampling regime without any compression (i.e., when $\omega=0$; see also the "LSVRG row" of Table 2 which summarizes this case), the communication complexity of our method becomes $O( \sqrt{L_{\tau} / \mu} \log 1/\epsilon )$ (see Corollary 1). Note that $L_\tau$ was defined in Table 2 (and recalled in Section 5 as well). Also note that as $\tau$ increases to $m$ (where $m=n/M$ is the number of clients belonging to each hub), then $L_\tau\to L$, and we recover the $O( \sqrt{L/ \mu} \log 1/\epsilon )$ communication complexity result of ProxSkip in the limit. On the other hand, if $\tau=1$, i.e., if only a single client is sampled by each of the $M$ hubs, then $L_\tau=\max_i L_i$, which is larger than $L$ and can be much larger. This is the price we pay for client sampling. In other words, of only a single client is sampled by each hub, then the number of communications increases from $O( \sqrt{L  / \mu} \log 1/\epsilon )$ to $O( \sqrt{\max_i L_{i} / \mu} \log 1/\epsilon )$. Note that this is still an accelerated communication complexity and the price we pay for client sampling is very small. In summary, it is not true that the effect of client sampling on ProxSkip is not discussed. However, it is true that the discussion can be improved, say, by adding something along the lines of what we just wrote to explain Corollary 1.
>
> - **On data sampling in the HUB model:** We did not consider data sampling of data owned by the clients in the HUB model as we know this would lead to overly complicated expressions and estimators. However,  this *can be done* and we can add a comment on this. We can also add a theorem if you request that (in a new section in the appendix); let us know! Our approach via Theorem 5 will still apply (since we can show that Assumption 4 will still apply!), which underlines the usefulness of our framework.

---

> ### Author Response · Authors · 2022-07-29
> **Suggestion 5: Client and data sampling (part 2)**
>
> ---
>
> Also, while client and data sampling are discussed in detail in Sec 1.2, their effect on ProxSkip-VR is never discussed.
>
> ---
>
>   **On data sampling the standard FL model (1):**
>
> - Our result (Theorem 9) from Section 4 also covers data sampling for the standard FL model (1); i.e., not for the HUB mode. Let us explain. To see this, we need to *re-interpret* the HUB model and the meaning of Theorem 9 differently, so that the interpretation makes sense from the viewpoint of (1). However, nothing needs to be changed in theory! Here is the new interpretation: think of the $M$ hubs as clients (so, there are $M$ clients instead of $M$ hubs), and think of the clients as data (so, each client has $m = n/M$ data points as opposed to each hub being associated with $m$ clients). With this new interpretation, what was client sampling in the HUB model turns into data sampling in the standard FL model (1) described in Sections 1.1 and 1.2. So, what we referred to by "data sampling" in Section 1.2 for solving the standard FL problem (1) is covered by Theorem 9 as well. The same conclusions apply as we described above; the communication complexity increases from $O( \sqrt{L  / \mu} \log 1/\epsilon )$ to $O( \sqrt{\max_i L_{i} / \mu} \log 1/\epsilon )$.
>
> - In fact, the setup and experiments in Section 5 consider precisely this case (i.e., we do no consider compression, and we interpret $M$ as number of clients and $m$ as number of data points per client), and we go more in depth here both analytically and empirically. For example, (13) provides the speedup factor we obtain through data sampling with ProxSkip-LSVRG (which is, as you will recall, special case of ProxSkip-HUB in the case when $\omega=0$) when compared to ProxSkip-GD (which samples all $m$ datapoints always) in terms of the **total cost** = communication cost + computation cost. **This theoretical speedup factor can be positive even for small values of $\delta$ (local computation cost), and increases with $\delta$. Our theoretical prediction (see blue line in Fig 2, bottom row) can be observed in practice (see the same plots). The theoretical speedups (35x in the first column) are somewhat larger than the observed practical speedups (10x in the same plot), and both are very highly significant. This is all an effect of data sampling!** This justifies the last two sentences of our abstract: "While all previous theoretical results for LT methods ignore the cost of local work altogether, and are framed purely in terms of the number of communication rounds, we show that our methods can be substantially faster in terms of the total training cost than the state-of-the-art method ProxSkip in theory and practice in the regime when local computation is sufficiently expensive. We characterize this threshold theoretically, and confirm our theoretical predictions with empirical results." We will clarify this more in the paper; indeed, we can see that this may not be entirely clear!
>
>
> **In summary, it is not true that we do not have results on client and data sampling in the paper. We do! However, it is true that our explanation is a bit terse, and we will improve the writing along the above explanations to make the results more clear! Thanks for raising this point, this will improve the paper.**

---

> ### Author Response · Authors · 2022-07-29
> **Suggestion 6: cross-silo vs cross-device**
>
> ---
>
> It should be made clear that all the discussion in the paper is for cross-silo FL and does not apply to cross-device FL.
>
> ---
>
> - This is clear already (implicitly) from the fact that in (1) we have a sum over finely many clients. We will also add a footnote mentioning this explicitly.
>
> - Having said that, please note that the novel HUB model is neither cross silo, nor cross device. It's a new model closely related to the cross-silo model, but not equivalent to it. We will mention this also.

---

> ### Author Response · Authors · 2022-07-29
> **Suggestion 7: the use of colors**
>
> ---
>
> Nitpick (feel free to ignore): the colors used in the text (e.g. in math symbols like $W$ and $\mathbb{E}$ in Thm 5, and algorithm names) made it visually busy, confusing and harder to read for me. I would suggest using such colors sparsely in text - more as a shorthand (e.g. red is always error terms) rather than for emphasis or drawing attention.
>
> ---
>
> - You are absolutely right. While this is indeed a small detail, we did *not* intend for $W$ and $\mathbb{E}$ to be colored. We uses the color in "production" to ensure consistency of macro use (i.e., consistency of notation). But we forgot to delete \color{red} from these two commands before submission! We will delete the colors for these two macros.
> - We did intend, however, to keep the algorithm names in red color. We believe this is helpful.

---

> ### Author Response · Authors · 2022-07-29
> **Possible score increase**
>
>
> ---
>
> Note: I am willing to raise my score on an assurance from the authors that the writing and presentation will be improved. The results are otherwise technically sound and interesting.
>
> ---
>
> **Thank you very much for this encouraging remark!**
>
> We do indeed assure you that we will improve the paper following your suggestions for improvements. We found them very helpful. We hope our detailed reply contained in our 7 posts names "Concern 1" - "Concern 7" give you enough information to see that we are indeed engaging with your questions and suggestions.
>
> **We hope this is enough for you to increase the score.** We are working on updating the paper. However, we are not sure if we manage to do it all by the time author feedback is over. We will certainly try! In any case, we will certainly do this all for the camera ready version of the paper at the very latest.
>
> We would very much appreciate if you could let us know what you think!

---

> ### Author Response · Authors · 2022-08-02
> **We managed to prove a new theorem covering HUB framework also with data sampling on the clients!**
>
> > Also, while client and data sampling are discussed in detail in Sec 1.2, their effect on ProxSkip-VR is never discussed.
>
> Dear Reviewer 2cyy,
>
> In addition to previous answer we will add two examples for HUB framework: ProxSkip-DIANA with unbiased stochastic estimator $g_t$ with bounded variance and ProxSkip-DIANA-VR with additional variance reduction for data sampling. We did not it previously since the notation becomes heavy and hard for understanding. However, these theorems are okay to be formulated in supplementary materials.
>
> **First theorem:**
>
> Assume that the stochastic estimator satisfies unbiasedness and bounded variance assumptions:
>
> $E \left[g_j(x_t)\right] = \tilde{\phi}_j(x_t),$
>
> $E || g_j(x_t) - \nabla \tilde{\phi}_j (x_t) ||^{2} \leq \tilde{\sigma}_j^2$
>
> for all $t \geq 0, j=1, \ldots, m$
>
> for constants
>
> $\tilde{\sigma}_j \leq \infty$,
>
> $\tilde{\sigma}^2 := \frac{1}{m} \sum_{j=1}^{m} \tilde{\sigma}_j^{2}$.
>
> Also assume that each $\tilde{\phi}j$ is $L_j$-smooth and $\mu$-strongly convex.
>
> Then using
>
> $c \geq \frac{4 \omega}{\alpha n}$, $\gamma \leq \frac{2}{(\mu+L)\left(1+\frac{2 \omega}{n}+c \alpha\right)}$ and $\gamma \leq \frac{\alpha}{2 \mu}$, we have
>
> $$
> E V_{t+1} \leq (1-\gamma \mu) || x_t - x_* ||^2 + X_t  + Y_t + Z,
> $$
>
> where
>
> $X_t = \left(1-\frac{\alpha}{2}\right)c\gamma^2 \frac{1}{n}\sum_{i=1}^{n}\Vert y^j_t - \nabla f(x_*) \Vert^2$
>
> $Y_t = (1-p^2)\frac{\gamma^2}{p^2} ||h_t - h_*||^2$
>
> $Z = \gamma^2\left(c\alpha+\frac{\omega+1}{n}\right)\sigma^2$
>
> $V_t = \Vert x_{t} -x_* \Vert^2 + c\gamma^2 \frac{1}{n}\sum_{j=1}^{n}\Vert y^j_{t} - \nabla f_i(x_*) \Vert^2 + \frac{\gamma^2}{p^2}\Vert h_{t} - h_{*} \Vert^2.$
>
> To make ProxSkip-DIANA-VR, we need to assume finite sum structure for each client in the hub:
>
> $\tilde{\phi}j(x) = \frac{1}{K} \sum_{k=1}^K \tilde{\phi}_{k,j}(x)$,
>
> then we can formulate another theorem using L-SVRG mechanism for client sampling for each HUB:
>
> **Theorem:**
>
> Assume that $\nabla \widetilde{\phi}_j$ is $L_j$-smooth for all $j$ and let Assumptions 2 and 3. Then for the gradient estimator of VR-DIANA, Assumption 4 holds with the following constants:
>
> $	A = L_{\max}\left(1+\frac{4\omega+2}{m}\right)$
>
> $ B =  \frac{2(\omega+1)}{m} $
>
> $C = 0$
>
> $\tilde{A} = L_{\max}\left(\frac{1}{K}+4 \alpha\right)$
>
> $\tilde{B} = 1-\alpha$
>
> $\tilde{C} = 0$
>
> and
>
> $\sigma_{t} =  O + P$,
>
> where
>
> $O = \frac{1}{m} \sum_{j=1}^m || c_t^i - \nabla \tilde{\phi}_j (x_\star) ||^2$
>
> and
>
> $P = \frac{1}{Km} \sum_{k=1}^{K} \sum_{j=1}^{m} || \nabla f_{k j} (w_{k j}^t )-\nabla f_{k j} (x_*) ||^2$.
>
> Let $L_{\max}=  \max_j L_j$ and $H = \frac{2B}{(1-\tilde{B})}$
>
> $0<\gamma \leq \min ( \frac{1}{\mu}, \frac{1}{(A+H \tilde{A}) } ).$
>
> Then the iterates of ProxSkip-VR for any $p\in (0,1]$ satisfy
>
> $$ E \Psi_{T}  \leq \max [ (1-\gamma \mu)^T, (1-p^2)^T, (1+\frac{2(\omega+1)}{mK M}-\alpha)^T ] \Psi_0, $$
>
> where the Lyapunov function is defined by
>
> $$\Psi_t = ||x_t - x_*||^2 + \frac{\gamma^2}{p^2} ||h_t - h_{\star}||^2 + \gamma^{2} H \sigma_t.$$

---

> ### Comment · Reviewer_2cyy · 2022-08-05
> **Thank you for the detailed clarifications!**
>
> I really appreciate the effort the authors took in responding to my concerns. I have raised my score and believe this paper represents substantial technical novelty and should be accepted. However, I strongly recommend the authors to incorporate all the clarifications made in the responses here directly into the main paper as well.

---

> > ### Author Response · Authors · 2022-08-05
> > **Re: Thank you for the detailed clarifications!**
> >
> > Thank you very much for this: both for the extra effort in looking through our rebuttal, and for increasing your score!
> >
> > We will incorporate as much of this as possible into the main body of the paper (subject to the page limits), and we will also incorporate the rest in a suitable place in the Appendix (and links in the main paper to the Appendix for this to be more reader-friendly). Thanks for the suggestions, the paper will be more accessible after the changes.

---

### Author Response · Authors · 2022-07-29
**A message to all 3 reviewers**

Dear reviewers,

Thanks for your reviews and the time you have put into reading and assessing our work. You have said many positive things about our paper, including:

- "The approach is well motivated and forms a natural line of inquiry - combining communication compression + variance reduction with ProxSkip" (Reviewer 2cyy)
- "The results obtained by the authors, though not surprising in the light of ProxSkip, are significant and technically involved. It is a result of synthesizing several state of the art analysis techniques, and recover most of them as state of the art." (Reviewer 2cyy)
- "I also enjoyed the motivation of the local variance reduction via the hub model and can see it being useful." (Reviewer 2cyy)
- "The results are otherwise technically sound and interesting." (Reviewer 2cyy)
- "The paper is well-written. I can see the history of FL methods evolving over different generations." (Reviewer RRdw)
- "ProxSkip-VR possesses several important properties for an FL method: client sampling, data sampling, and variance reduced local update step." (Reviewer RRdw)
- "The paper proposes a new architecture for FL which contains hubs between server and workers. This architecture is reasonable and it represents many real-world applications." (Reviewer RRdw)
- "Convergence results are provided which cover different type of gradient estimators." (Reviewer RRdw)
- "The evaluation considers the total cost which consists of communication cost and local training cost which has not been considered in existing works. This also help show the advantage of using variance reduced estimator in local training." (Reviewer RRdw)
- "The method constructed by the authors can be substantially faster in terms of the total training time than ProxSkip in theory and practice in the regime when local computation is sufficiently expensive." (Reviewer VKkX)
- "The proposed method ProxSkip-VR enjoys a lower communication complexity than gradient descent." (Reviewer VKkX)
- "This paper proposed a new FL architecture, where ProxSkip-VR can be substantially faster in terms of the total training time than ProxSkip." (Reviewer VKkX)
- "The authors corroborate their theoretical results with carefully engineered proof-of-concept experiments." (Reviewer VKkX)

The initial scores we received are: 7 - accept (Reviewer RRdw), 5 - borderline accept (Reviewer 2cyy) and 4 - borderline reject (Reviewer VKkX).

- Reviewer RRdw (score 7): Thanks for your very positive evaluation of our work! We will respond to all your (we believe minor) concerns in detail soon.

- Reviewer 2cyy said (score 5): "**I am willing to raise my score** on an assurance from the authors that the writing and presentation will be improved. The results are otherwise technically sound and interesting." **We do assure you that we will do as you ask!** Soon we will respond to all your concerns, and we will also improve the explanations, as per your request. We will try to do this as soon as possible. We hope you will be satisfied with this and raise your score accordingly.

- Reviewer VKkX (score 4) is "absolutely certain" (confidence 5) about their evaluation, yet makes some i) assertions when evaluating our work that happen to be factually incorrect (we will explain this in detail), and ii) unsubstantiated generic comments. **We believe that this review is (unfortunately, grossly) misleading and does not do justice to our work.** We will attempt to persuade this reviewer in our response that the key issues raised are non-issues and stem from a misunderstanding by the reviewer. We hope this will work - we would be very glad if this reviewer could engage with us! We are most happy to clarify.

Kind regards,

the authors

---

> ### Author Response · Authors · 2022-07-30
> **We have now thoroughly addressed all issues pointed out by all reviewers!**
>
> Dear reviewers,
>
> We have now (we believe thoroughly) addressed all issues you have raised. Please read our responses and engage with us. We will be most delighted to hear what you have to say! If any misunderstandings or issues remain, we will try to explain and resolve!
>
>
> - We believe we addressed all questions of **Reviewer RRdw (score 7)**. Please let us know if we addressed everything satisfactorily, and especially please let us know whether there is anything else we can do for you to be able to raise your score for our paper, if this is a possibility. Please may you consider championing our paper in reviewer discussion phase? We would be delighted if you could do that! Thanks in advance!
>
> - We believe we addressed all (very useful!) suggestions for improvement by **Reviewer 2cyy (score 5)**. This reviewer said " I am willing to raise my score on an assurance from the authors that the writing and presentation will be improved" - we gladly offer these assurances. At the same time, we offer evidence that we understood what the reviewer meant in our detailed response, which we hope goes much beyond just verbal "assurances". We are also working on improving the actual paper, and hope to be done by the author feedback deadline. However, we are not sure if we can do this all that fast as we have other reviewing and editorial duties as well at the same time, and also some of us are taking up vacations, which does not help. But we will try our best. We will certainly have it all done by the camera ready deadline at the latest. We would kindly ask this reviewer to let us know if our response is satisfactory. Thanks in advance!
>
> - We believe that essentially all criticism raised by **Reviewer VKkX (score 4)** is either factually invalid (and we provide evidence), or grossly misleading (and we explain why), or unjustified and invalid (and we address this as well). We hope this reviewer will engage with our author response. However, we wish to flag this reviewer the Area Chair since this review is simply not addressing any real issues in our paper, and is simply just adding noise and confusion to the evaluation process. It is very clear to us from the review comments that this reviewer is not an expert in the area, yet the reviewer self-evaluated as "absolutely certain". This is a big red flag as well. With the same breath, we wish to thank the reviewer for their effort! We truly appreciate it despite what we say above. We know that sometimes one is asked to review a paper outside of one's domain, and it is hard to evaluate such papers. We needed to defend our paper - we hope you understand this. It's just our professional duty. We have absolutely nothing against the reviewer as a person. Thanks!
>
>
>
>
> Knd regards,
>
> Authors

---

### Author Response · Authors · 2022-08-02
**ProxSkip-VR-D: a new variant of ProxSkip-VR, and encouraging preliminary experiments**

Dear reviewers,

We were inspired by one question of Reviewer VKkX to propose and test numerically a new variant of our new method ProxSkip-VR, which we call ProxSkio-VR-D. This method takes $1/p$ local steps *deterministically*, whereas ProxSkip-VR takes $1/p$ local steps *in expectation*. We do not have an analysis for this variant; this is an open problem. Indeed, randomness is a key ingredient in our analysis (and in the analysis of the original ProxSkip algorithm) which makes it possible. So, the analysis of this new variant might be difficult.

We used the LSVRG estimator for our methods in the below experiments.

Since we can not upload plots here, we have at least tried to convey the essential information they uncover by summarizing the results in a table; these are the same as those in our original submission, top row of Table 2, but we added the run of ProxSkip-VR-D:

**batch size $\tau=16$:**

| Comm. rounds  | 1000     | 3000     | 5000      | 7000       | 9000       |

| ProxSkip-VR     | 7.53e-4 | 1.88e-6 | 3.92e-8 | 1.67e-10 | 3.19e-11 |

| ProxSkip-VR-D | 6.31e-4 | 1.51e-6 | 1.09e-8 | 1.31e-10 | 3.17e-11 |

**batch size $\tau=32$:**

| Comm. rounds  | 1000     | 3000     | 5000        | 7000       | 9000       |

| ProxSkip-VR     | 2.55e-4 | 8.03e-8 | 4.11e-10 | 2.96e-11 | 2.96e-11 |

| ProxSkip-VR-D | 1.15e-4 | 2.50e-8 | 5.28e-11 | 2.97e-11 | 2.96e-11 |

**batch size $\tau=64$:**

 | Comm. rounds  | 1000     | 3000       | 5000        | 7000       | 9000       |

 | ProxSkip-VR     | 1.77e-4 | 1.78e-5   | 4.70e-9   | 7.12e-11 | 2.98e-11 |

 | ProxSkip-VR-D | 3.35e-5 | 4.01e-10 | 2.98e-11 | 2.96e-11 | 2.96e-11 |

The numbers in the tables show the value of $f(x^t)-f(x^\star)$, i.e., the function suboptimality, after a certain number of communication rounds, as indicated by the number in the column. For each experiment, we obtained a tight estimate of the optimal value $f(x^{\star})$ by Nesterov's accelerated gradient descent method for a long time. We used the same stepsize for ProxSkip-VR-D as for ProxSkip-VR; i.e., we did not try to fine-tune the results, which means that both methods will likely improve their performance with further fine-tuning.

**Our key finding is that ProxSkip-VR-D is slightly better, and can be even quite significantly better in the initial phases of the iterative process for larger minibatch sizes $\tau$.** For example, in the third experiment with $\tau=64$, ProxSkip-VR-D achieves accuracy $\varepsilon=4.01e-10$ after 3000 communication rounds, whereas ProxSkip-VR only achieves the accuracy $\varepsilon=1.78e-5$. This early advantage disappears later on. After 9000 communication rounds, both methods in all three experiments achieve comparable accuracy.

We believe this **strengthens our contributions with a new heuristic variant of our method that appears to work better.** We will conduct more extensive testing and include the plots in the camera-ready version of the paper.

---

### Meta-Review · Area_Chair_Zn72 · 2022-08-26

**Recommendation:** Accept
**Confidence:** Certain

**Metareview:**

The paper proposes a new optimization algorithm for distributed learning with applications to federated learning. The results are interesting and I recommend acceptance. This paper benefited considerably during the author rebuttal phase and I strongly urge the authors to incorporate all the reviewer suggestions and the author clarifications in the final version of the paper.

**Award:**

No

---

### Decision · Program_Chairs · 2022-09-14

Accept